# PARAMETER IDENTIFIABILITY AND PRESERVED DYNAMICS IN DATA-CONSTRAINED RECURRENT NEURAL NETWORKS

## ABSTRACT

Researchers routinely study the neural algorithms of the brain by training data-constrained recurrent neural networks (dRNNs) to reproduce observed neural activity. However, whether the biological insights gained from these overparameterized dRNNs are actionable remains underexplored. In particular, it is unclear which dRNN parameters are constrained by a given training set of neural trajectories. To bridge this gap, we focus on a simplified but experimentally relevant setting of dRNN training, characterize the identifiable parameter subspaces there, and report five key findings: (i) dRNNs contain vast unconstrained parameter regions due to intrinsically low-dimensional training data; (ii) existing training methods can mistakenly attribute importance to non-identifiable parameters; (iii) a generalized blueprint explains the ability of practical estimators to operate exclusively within identifiable parameter subspaces; (iv) despite parameter non-identifiability, activity subspaces with preserved dynamics exist across all trained dRNNs; and (v) targeted intervention experiments can optimally expand the identifiable parameter subspaces. Our results establish practical guidelines to overcome parameter non-identifiability issues when training dRNN models in systems neuroscience.

## 1 INTRODUCTION

Recent advances in large-scale neural recording allow researchers to measure brain-wide activity in animals (Kim & Schnitzer, 2022; Manley et al., 2024; Bounds & Adesnik, 2024; Stringer et al., 2019). Computational neuroscientists have developed methods to analyze these high-dimensional recordings and gain mechanistic insights into neural computation (Schneider et al., 2023; Gardner et al., 2022; Mante et al., 2013; Sussillo & Barak, 2013; Pandarinath et al., 2018). A key conceptual advance is that the brain represents information at the level of neural populations rather than individual neurons (Saxena & Cunningham, 2019; Pouget et al., 2000; Kira et al., 2023; Churchland et al., 2012; Averbeck et al., 2006). With this view, researchers analyze *dynamical properties* of population-activity patterns to understand how the brain solves tasks (Liu et al., 2024; Nair et al., 2023; Langdon et al., 2023; Vyas et al., 2020; Khona & Fiete, 2022), revealing, for instance, how a line attractor in the hypothalamus might encode aggression in male mice (Vinograd et al., 2024).

A prominent approach in systems neuroscience fits neural network models to reproduce recorded neural activity. The trained models then serve as in silico analogues of the biological circuits (Perich & Rajan, 2020). These models have been used to analyze the structure of population dynamics, including latent variables underlying neural activity (Valente et al., 2022; Nair et al., 2023), flow fields governing responses to perturbations (Kim et al., 2023; Linderman et al., 2017), and communication patterns across brain regions (Perich et al., 2021; Perich & Rajan, 2020). Critically, their predictions are increasingly used to guide causal experiments (Walker et al., 2019; Liu et al., 2024; Vinograd et al., 2024), though their internal structure is not guaranteed to reflect ground truth mechanisms (Das & Fiete, 2020; Qian et al., 2024; Brinkman et al., 2018; Göring et al., 2024). A key failure mode arises when some model parameters are not constrained by the data distribution; this is the problem of parameter identifiability, the focus of this work.

**Models of neural activity.** Models trained to reproduce neural activity have long promised insight into biological and computational mechanisms, though their interpretability remains debated. Current approaches in this space can be broadly placed into one of the three categories. First, latent variable models are used to extract low-dimensional variables and their dynamics from high-dimensional neural activities (Langdon & Engel, 2025; Dinc et al., 2025; Schneider et al., 2023;

Dubreuil et al., 2022). The experimental relevance of the extracted latent structures, *e.g.*, fixed-point attractors, can be tested with perturbation experiments (Vinograd et al., 2024). But, the exact causal link between neural activities and latent variables is not always obvious.

Second, comparably smaller (100-1000s of neurons) recursive models are used to explicitly account for unobserved influences and make the most accurate predictions possible (Durstewitz et al., 2023; Schmidt et al., 2019). Here, prediction is paramount. Third, simplified data-constrained RNN (dRNN) models (Perich & Rajan, 2020) are used to explain observed neural dynamics (Dinc et al., 2023), infer experimentally relevant quantities such as inter-area communication patterns (Perich et al., 2021), and test the plausibility of theoretical hypotheses directly on empirical datasets (Valente et al., 2022; Finkelstein et al., 2021). These studies use simple RNN models, which are less powerful but more interpretable and nonetheless possess universal approximation properties (Beiran et al., 2021). Our work focuses on models of this third type.

**Challenges of dRNNs.** Recovering synaptic connectivity from observed dynamics is generally ill-posed (Das & Fiete, 2020; Brinkman et al., 2018), and functional properties, such as presumed underlying attractors, can be unreliable when inferred from data alone (Qian et al., 2024; Göring et al., 2024). Even dRNNs with a one-to-one mapping between recorded and modeled neurons (Perich & Rajan, 2020; Perich et al., 2021; Dinc et al., 2023) remain poorly understood in terms of identifiability. Nevertheless, RNNs and other predictive models have been used to uncover putative mechanistic features, including population-level gating mechanisms (Finkelstein et al., 2021), inter-area communication motifs (Perich & Rajan, 2020), and low-dimensional attractor dynamics (Valente et al., 2022). In general, dRNNs trained on neural data can yield either genuine mechanistic insights or misleading interpretations; and sometimes both.

**Identifiability of dRNN parameters.** Parameter symmetries have been characterized in both recurrent (Al-Falou & Trummer, 2003; Biswas & Fitzgerald, 2022) and feedforward architectures (Bui Thi Mai & Lampert, 2020; Bona-Pellissier et al., 2023). In these cases, however, the primary concern is to characterize the properties of parameters that support the input-output map or steady-state responses, rather than constrain the network to reproduce the continuous neural-activity dynamics. In contrast, linear dynamical systems enjoy remarkably clean identifiability properties: under conditions of controllability (Kalman et al., 1960) and observability (Kalman, 1963), system parameters can be uniquely recovered from input-output trajectories up to well-understood equivalences (Grewal & Glover, 2003), which can guide explorations in nonlinear RNNs. An extended review, including additional background on broader identifiability literature, is provided in Appendix S1.

**Contributions.** We examine when and how dRNN parameters are constrained by their neural datasets. We then address estimation from finite, noisy data, suggesting how estimation can be engineered to confine parameters to their identifiable components. Finally, we derive two experimental insights: (i) variation in some directions in parameter space yields the same predictions, but not in others, and (ii) data collected with targeted experimental interventions can expand identifiable parameter subspaces. Understanding when and why such divergence occurs is essential, as finding mechanistic insight in unconstrained parameters can mislead analysis and waste experimental effort.

## 2 RESULTS

### 2.1 PARAMETER IDENTIFIABILITY IN DRNNS TRAINED AS DIGITAL TWINS

We consider a biologically motivated and interpretable class of RNNs characterized by:

$$\tau \dot{r}(t) = -r(t) + \phi(W^{\text{rec}} r(t) + W^{\text{in}} u(t) + \epsilon_{\text{in}}(t)) + \epsilon_{\text{conv}}(t), \tag{1}$$

where $\tau \in \mathbb{R}$ is the time constant, $r(t) \in \mathbb{R}^N$ the neural activities and $\dot{r}(t) \in \mathbb{R}^N$ their time derivatives, $u(t) \in \mathbb{R}^{N_{\text{in}}}$ the inputs, $W^{\text{rec}} \in \mathbb{R}^{N \times N}$ the recurrent weights, $W^{\text{in}} \in \mathbb{R}^{N \times N_{\text{in}}}$ the input weights, $\epsilon_{\text{in}}(t)$ and $\epsilon_{\text{conv}}(t) \in \mathbb{R}^N$ some unknown input and conversion noise terms, and $\phi(\cdot)$ a monotonic nonlinearity. When necessary, one can absorb bias terms into $W^{\text{in}}$ by fixing one input to unity. The noise terms $\epsilon_{\text{conv}}$ and $\epsilon_{\text{in}}$ also model observation errors jointly (see **Methods**). For analysis, we define neural-input states as $x(t) = [r(t), u(t)] \in \mathbb{R}^{N_{\text{tot}}}$ with corresponding parameter matrix $\theta = [W^{\text{rec}}, W^{\text{in}}] \in \mathbb{R}^{N \times N_{\text{tot}}}$ and discretize dynamics with step size $\alpha = \Delta t / \tau$.

This architecture follows a simplified abstraction where neurons compute a weighted sum of their inputs and then apply a threshold function to determine their output (McCulloch & Pitts, 1943).

Despite its simplicity, Eq. 1 can approximate arbitrary, smooth low-dimensional dynamical systems (Dinc et al., 2025; Beiran et al., 2021) and continuous input-output mappings (Schäfer & Zimmermann, 2006) in the limit of an infinite number of neurons: $N \to \infty$. To build dRNNs, we record neural activity from $N$ neurons while model animals, *i.e.*, "generators", perform $M$ behavioral trials. Each trial $m \in \{1, \ldots, M\}$ includes task inputs (like visual cues, conceptualized as $u(t)$ in Eq. 1) and measured activities $\tilde{r}_i(t)$ from neurons $i = 1, \ldots, N$ over time $t = 1, \ldots, T^{(m)}$. In total, this yields $T = \sum_{m=1}^{M} T^{(m)}$ many samples of neural activities. We then estimate the parameters $\theta$ such that the dRNN's activities $r_i(t)$ match the recorded activities $\tilde{r}_i(t)$.

While these models make simplifying assumptions (ignoring unobserved neurons, imposing specific forms on neuronal dynamics), they can successfully reproduce neural activity and generate hypotheses about population computations (Perich et al., 2025). However, dRNNs are overparameterized and thus can make predictions using parameters unconstrained by data, which can produce incorrect insights, potentially leading to predictions that waste experimental resources (Qian et al., 2024; Das & Fiete, 2020). To formally study this issue and identify its potential remedies, we first define parameter identifiability in dRNNs constrained to reproduce a set of neural activities $\mathcal{Y} = \{r(1), r(2), \ldots, r(T)\}$ given neural inputs[1] $\mathcal{X} = \{x(0), \ldots, x(T-1)\}$:

**Definition 1** (Parameter identifiability in dRNNs). *Given a set of samples $\{X_i\} \subset \mathcal{X}$ of observed quantities for $i = 1, \ldots, T$, let $\mathcal{P}_{\mathcal{X}} = \{P(Y_i|X_i; \theta), \theta \in \Theta\}$ be the family of probability distributions describing the predictions $\{Y_i\} \subset \mathcal{Y}$ made by an dRNN model with parameter space $\Theta$. Then, dRNN is identifiable if and only if, for all observed $X_i$, the mapping $\theta \mapsto P(Y_i|X_i; \theta)$ is injective.*

In practice, for a sample $(x_i, y_i) \in (\mathcal{X}, \mathcal{Y})$, the relationship $y_i \sim P(Y|x_i; \theta)$ is established using Eq. 1. In the noiseless case, each input $x$ uniquely determines the output $y$ following a Dirac distribution centered at $y = -r + \phi(\theta x)$ for $x = [r, u]$. In this case, a dRNN parameterized by $\theta$ is identifiable if and only if one unique $\theta$ makes correct one-step predictions on all observed neural activities. Finally, since this definition is concerned with the reconstruction process of entire neural trajectories, it places a more stringent condition on the RNN parameters compared to earlier works studying low-dimensional input-output mappings (Al-Falou & Trummer, 2003) or steady-state neural responses (Biswas & Fitzgerald, 2022) in RNNs.

## 2.2 NEURAL-INPUT SUBSPACES CONSTRAIN LINEAR COMBINATIONS OF PARAMETERS

Our focus on single-step predictions renders dRNNs equivalent to a generalized linear model, whose parameters can be divided into identifiable and non-identifiable components:

**Theorem 1** (Identifiability in dRNNs). *Consider an RNN defined by Eq. 1 with parameters $\theta^* \in \mathbb{R}^{N \times N_{\mathrm{tot}}}$, where the noise random variables $\epsilon_{\mathrm{in}}$ and $\epsilon_{\mathrm{conv}}$ are independent, with the latter having a non-vanishing characteristic function. Consider an observation matrix $X \in \mathbb{R}^{T \times N_{\mathrm{tot}}}$ defining the conditioning domain $\mathcal{X}$ and denote $P_{\mathcal{X}} \in \mathbb{R}^{N_{\mathrm{tot}} \times N_{\mathrm{tot}}}$ the projection matrix onto its row space. Then, any discretized RNN parametrized by $\theta$ such that:*

$$\theta P_{\mathcal{X}} = \theta^* P_{\mathcal{X}}, \tag{2}$$

*gives the same conditional probability distribution on single-step predictions as the ground-truth RNN. Then, $\theta^* P_{\mathcal{X}}$ (out of all RNN parameters) is identifiable if and only if the parameter space is restricted to $\Theta_{\mathcal{X}} = \{\theta \in \mathbb{R}^{N \times N_{\mathrm{tot}}} : \theta = \theta P_{\mathcal{X}}\}$ (identification condition). In particular, an unrestricted $\theta^* \in \mathbb{R}^{N \times N_{\mathrm{tot}}}$ is identifiable if and only if $P_{\mathcal{X}} = I$. (Proof in Appendix S2.1.3.)*

The proof follows from the multiplicative relationship between $X$ and $\theta$ in Eq. 1, which leads to an equivalence class spanned by the projection matrix $P_{\mathcal{X}}$. Theorem 1 reflects the broader principle that parameters are identifiable only up to the information content in the observed data (Rothenberg, 1971). The specific condition quantifying this content, *i.e.*, $P_{\mathcal{X}}$, is well-studied in the context of generalized linear models, regression, and dynamical systems (Rao et al., 1973; Ljung & Glad, 1994). Our key insight here is recognizing that data-constrained RNNs in neuroscience, despite their nonlinear global dynamics, can be studied as part of this classical identifiability framework. Here, as illustrated in Fig. 1**A-B**, linear subspaces spanned by the observed neural activities constrain linear combinations of RNN parameters. In what follows, we refer to the subspace $\Theta_{\mathcal{X}} = \{\theta \in \mathbb{R}^{N \times N_{\mathrm{tot}}} :$

---

[1]Here, to prevent cluttered symbols, we abuse the notation and use $\mathcal{X}$ to refer to both the training samples and the conditioning domain they represent. These two are related, but not exactly the same, notions.

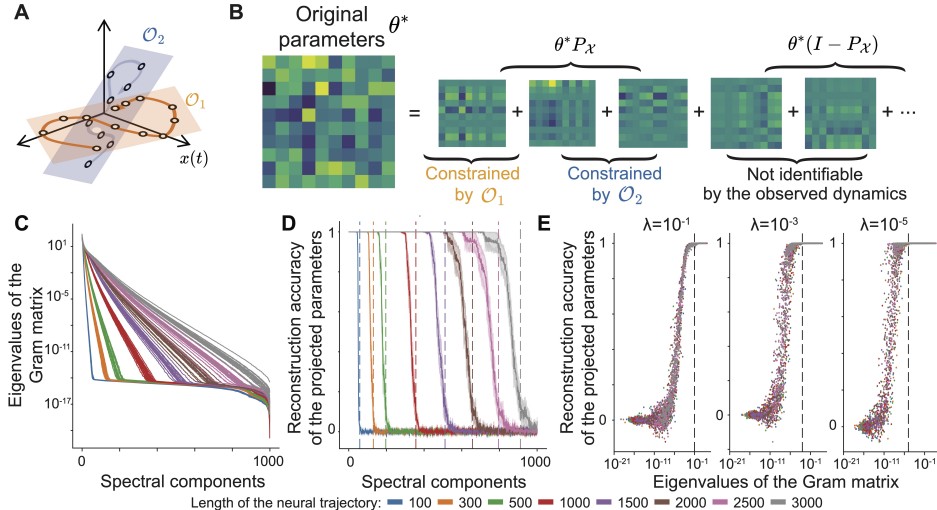

Figure 1: **Neural trajectory subspaces constrain linear combinations of RNN parameters. A** Observed neural trajectories $\mathcal{O}_1$ and $\mathcal{O}_2$ are confined to linear subspaces and constrain linear combinations of the parameters. **B** Parameters can be decomposed into components constrained by the observed data and an unconstrained, non-identifiable remainder. **C-E** We train dRNNs to replicate trajectories of varying lengths, continuously sampled from noiseless, chaotic generator RNNs. **C** Eigenvalues of the Gram matrix for each reconstruction instance. Each solid line corresponds to a distinct seed and/or trajectory length. **D** To compute reconstruction accuracy, for each spectral component of the Gram matrix, we first projected both the ground truth and estimated parameters onto that component, and then computed the correlation between these projections. Solid lines: mean; shaded regions: s.e.m. over 20 randomly initialized RNNs. Dashed lines: the spectral threshold corresponding to eigenvalues $< 10^{-14}$ averaged over 20 seeds. **E** Reconstruction accuracy of the projected parameters versus the corresponding eigenvalues of the Gram matrix for varying regularization strengths ($\lambda$). Parameters: see Appendix S3.

$\theta = \theta P_\mathcal{X}\}$ as the identifiable parameter subspace, and $\theta^* P_\mathcal{X}$ as the identifiable component of the ground truth parameters $\theta^*$. While Theorem 1 proves that an identifiable component exists, it does not guarantee its practical estimation from finite samples, rather that the non-identifiable component $\theta^*(I - P_\mathcal{X})$ is unconstrained and cannot be recovered.

## 2.3 NEURAL-INPUT SUBSPACES CONSTRAIN LIMITED COMBINATIONS OF PARAMETERS EVEN IN THE ABSENCE OF NOISE

We now illustrate Theorem 1 empirically using noiseless chaotic RNNs with ground-truth parameters $\theta^*$. To do so, we train dRNNs by minimizing single-step prediction errors:

$$\hat{\theta} = \mathrm{argmin}_\theta \mathcal{L}(\theta), \quad \text{where:} \quad \mathcal{L}(\theta) = \frac{1}{T}\sum_{i=1}^{T} \mathcal{L}_{\text{single}}(y_i, \text{dRNN}(\theta, x_i)). \quad (3)$$

Here, $\text{dRNN}(\theta, x_i)$ is a single-step prediction. With an appropriate choice of loss function $\mathcal{L}_{\text{single}}$ (*e.g.*, a convex loss (Dinc et al., 2023) for Figs. 1 and 2 ), we expect $\hat{\theta}P_\mathcal{X} = \theta^* P_\mathcal{X}$, where the projection matrix is $P_\mathcal{X} = X^T(XX^T)^+X$. Then, Gram matrix defined as:

$$G_\mathcal{X} = \frac{1}{T}X^T X \quad \in \mathbb{R}^{N_{\text{tot}} \times N_{\text{tot}}} \quad (4)$$

has the same rank as $P_\mathcal{X}$. Hence, its non-zero eigenvalue count gives $\dim \Theta_\mathcal{X}$, the dimensionality of the restricted parameter subspace, and in the case of mean-centered observation matrix $X$, its spectral decomposition corresponds to the principal component analysis regularly performed in neural datasets. We computed the Gram matrix eigenvalues for trajectories from the noiseless chaotic RNNs (Fig. 1**C**). Short trajectories show rapid eigenvalue decay to machine precision ($\sim 10^{-14}$) within tens of spectral components, while longer trajectories sustain more non-zero eigenvalues. Chaotic RNNs explore broader state space as training sample count $T$ increases, which allows us

to test Theorem 1 across diverse conditions. Components of the RNN parameters projected onto spectral components with non-zero eigenvalues (but not others) were accurately reconstructed (Fig. 1**D**), *i.e.*, noiseless estimation process recovers only the identifiable components.

Without Theorem 1, one could attempt to explain why dRNNs fail to recover certain parameter components with three hypotheses. First, one might argue that the training procedure is inadequate. However, all dRNN models in Fig. 1**C-D**, despite having parameters $\theta$ distinct from the ground truth parameters $\theta^*$, achieved near-perfect training accuracy (single-step root mean square error of $\leq 10^{-7} \pm O(10^{-8})$, see Fig. S1). Second, the estimation may lack proper regularization, causing overfitting. However, enforcing weight regularization does not improve but actually worsens the dimensions of parameters that can be estimated (Fig. 1**E**). Finally, one might suspect that these results are specific to these hyperparameters (*e.g.*, network size). Yet these results generalize to networks of varying sizes (Fig. S2). Theorem 1 explains why all three hypotheses fail: the issue is not the quality of the estimation but fundamental limits in parameter identifiability. Only by collecting more diverse samples can we raise $\text{rank}(G_\mathcal{X})$ and expand the identifiable subspace $\Theta_\mathcal{X}$.

### 2.4 PARAMETER ESTIMATION ACCURACY DROPS WITH THE GRAM MATRIX SPECTRUM UNDER NOISY DYNAMICS

Without noise, an appropriate estimator recovers the identifiable part of the ground-truth parameters; every sample can be matched exactly, and each one imposes a linear constraint on the parameters. As a result, the loss in Eq. 3 has at least one global minimum with value zero, and any minimizer $\hat{\theta}$ must satisfy $\hat{\theta}P_\mathcal{X} = \theta^* P_\mathcal{X}$. In Fig. 1, the estimator was convex, so this was the unique global minimum, and the identifiable component was fully recovered. Next, we show that Theorem 1 still provides useful guarantees under noisy dynamics.

*Usefulness of the Gram matrix in quantifying the sample size:* Without knowing the noise distribution or the specific estimator used to minimize Eq. 3, we rely on sample-size intuition: estimating parameters becomes easier with more samples. Theorem 1 gives the key idea. Each sample $x \in \mathbb{R}^{N_{\text{tot}}}$ constrains a linear combination of parameters, $\theta^* P_x$ where $P_x = \frac{xx^T}{x^T x}$. The strength of this constraint depends on how often $x$'s direction appears in the training set. The spectrum of the Gram matrix in Eq. 4 shows which directions are supported by many samples (Fig. 2**A**) and, therefore, which linear combinations of parameters are best constrained under noisy dynamics.

To test this, we repeated the experiments from Fig. 1**C-E** with i.i.d. noise $\epsilon_{\text{in}} \sim \mathcal{N}(0, \sigma_{\text{in}}^2)$ added to each neuron at each time point (see Eq. 1). Unlike the noiseless case (Fig. 1**C**), all eigenvalues of the Gram matrix were non-zero (Fig. 2**B**). However, this did not translate to successful parameter estimation. The dRNNs reconstructed parameters along directions corresponding to the highest eigenvalues, but not the lower ones (Fig. 2**C-D**), where the estimations exhibited inflated norms (Fig. 2**E**). Relatedly, using $\hat{\theta}P_K$ ($P_K$ is the projection operator to the top $K$ Gram-matrix spectral components) for different values of $K$ to estimate $\theta^*$ revealed that the optimal choice was $K < N$ across all tested noise levels (Fig. 2**F**). In other words, the sampled directions at the bottom of the spectrum, *i.e.*, directions underrepresented in the training data, corrupted parameter estimation.

*Regularization can mitigate spurious estimation of non-identifiable components:* Linear combinations of parameters aligned with the lower spectrum of the Gram matrix were less constrained, and their norms overestimated (Fig. 2**E**). Weight regularization could in principle mitigate this. In fact, Dinc et al. (2023) has shown that Eq. 3 approximates regularized least-squares between $X_i$ and $\phi^{-1}(Y_i)$ near its global minimum when the loss is a weighted convex loss (see Eq. S60). In this case, results from seminal works such as (Tikhonov & Arsenin, 1977) suggest that the estimated parameters can be written as a summation over the spectrum of the Gram matrix following $\hat{\theta} = \sum_{i=1}^{R} S_\lambda(\sigma_i^2)C_i$, where $C_i \in \mathbb{R}^{N \times N_{\text{tot}}}$ is some data-dependent rank-one contribution to the estimated parameters $\hat{\theta} \in \mathbb{R}^{N \times N_{\text{tot}}}$, $\sigma_i^2$ is the $i$th eigenvalue of the Gram matrix in Eq. 4, and $S_\lambda(x) = \frac{x}{x+\lambda}$ is a smooth thresholding function with properties $S_\lambda(x/\lambda \to \infty) \to 1$ and $S_\lambda(x = 0) = 0$. In words, more regularization suppresses the reconstruction of the linear combination of parameters aligned with the lower Gram matrix spectrum (Figs. 1**F** and S3). Consequently, regularization suppressed the noise-induced parameter inflation (Fig. S4**A-B**). Finally, Fig. S4**C-D** illustrates an example dRNN training, in which the optimal regularization parameter can be chosen via cross-validation.

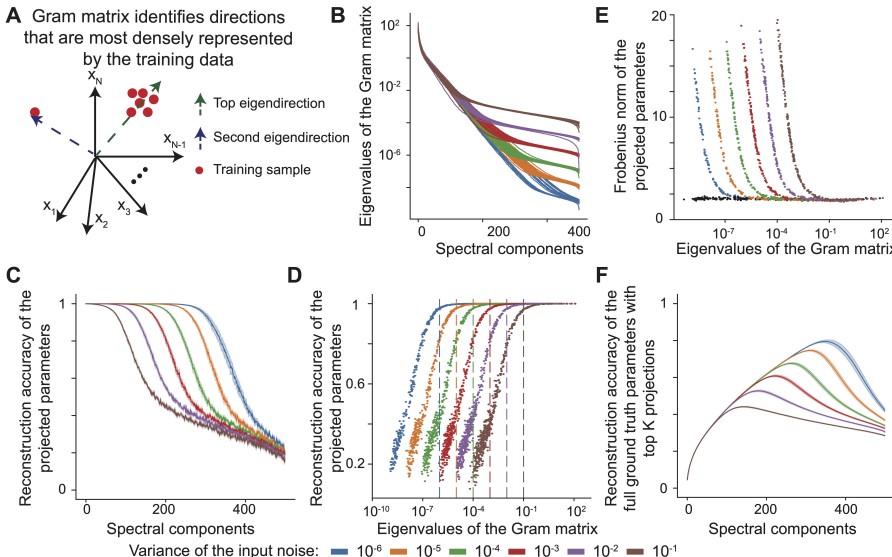

Figure 2: **Noisy dynamics contracts the estimable subspaces residing on the Gram matrix spectrum.** **A** The Gram matrix identifies directions most densely represented by the training data, with the top eigendirection aligned with the direction with the highest sample density. **B-F** Parameter reconstruction experiments as in Fig. 1**C-E**, but with input noise $\epsilon_{\text{in}} \sim \mathcal{N}(0, \sigma_{\text{in}}^2)$. For varying noise levels, we show: **B** Eigenvalues of the Gram matrix. **C** Reconstruction accuracy of the projected parameters versus spectral components. **D** Scatter plot comparing reconstruction accuracy against corresponding eigenvalues. **E** Frobenius norm of the projected parameters along spectral components. Here, black dots correspond to the norm of the projected ground truth parameters. **F** Accuracy of reconstructing $\theta^*$ with $\hat{\theta}P_K$, in which $P_K$ is the projection matrix constructed using the top K spectral components. Parameters: see Appendix S3.

## 2.5 TRAINING METHODS CAN SPURIOUSLY ESTIMATE NON-IDENTIFIABLE PARAMETERS

In practice, dRNNs are trained using various algorithms. The dominant approach, FORCE, uses modified recursive least-squares to update parameters (Sussillo & Abbott, 2009; Perich et al., 2021), which we now study under the lens of Theorem 1.

*Regularization does not eliminate non-identifiable components in FORCE learning:* FORCE starts by randomly initializing dRNN weights, often at the edge of chaos (Perich et al., 2021). Then, as dRNN dynamics are inferred forward in time, weights are simultaneously updated using the prediction error and an estimate of the least-squares Hessian, which is initialized as $H := \lambda^{-1}I$ and updated in an online manner with incoming data streams. $\lambda$ corresponds to a weight regularization in the limit of large samples (Mahadi et al., 2022). To test whether this approach leads to final results confined to identifiable components, we repeat the noise-free estimation from Fig. 1 across varying regularization strengths $\lambda$. In contrast to CORNN, FORCE learning did not suppress the non-identifiable components, even when $\lambda$ was scaled to very large values that decreased the accuracy (Fig. 3**A-B**). As learning converged to correct predictions within the top spectral components, only identifiable parameters continued to receive updates, whereas projections onto the lowest spectral components retained their norms (Fig. 3**B**, contrast FORCE vs. CORNN) and remained highly correlated with their initialization (Fig. 3**C**). Hence, FORCE learning not only underperformed compared to CORNN, but also retained non-identifiable parameters from initializations that were not constrained by the observed training samples.

*Low-rank regularization can lead to incorrect estimation under partial observations:* So far, we assumed that all neurons in a network were observed and the network itself had high-dimensional chaotic dynamics. Next, we study a realistic scenario, in which only a fraction of neurons are observed and dynamics stem from a low-dimensional structure. To this end, we reanalyzed the experiment of (Qian et al., 2024, Figure 5) (Figure S3**D-F**), which has shown that FORCE fails to recover the correct dRNN dynamics. Following recent advances (Valente et al., 2022; Dubreuil

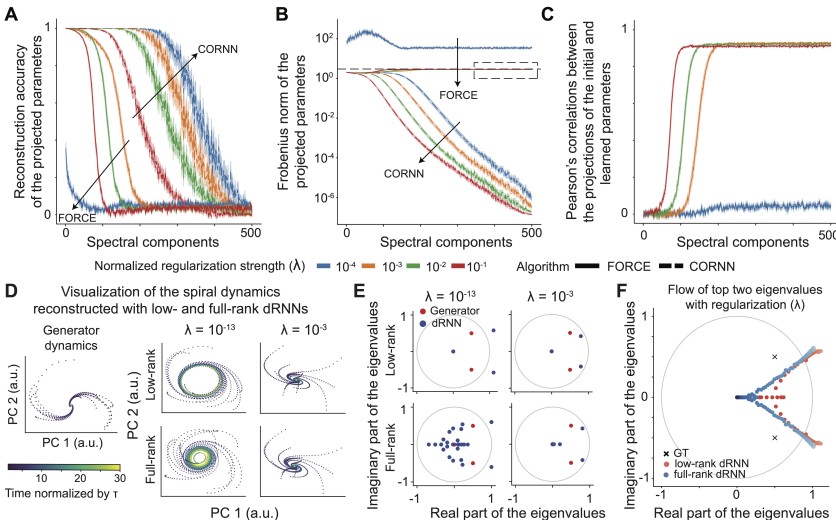

Figure 3: **Common estimators lack built-in guarantees for staying within the identifiable subspace. A-C** Tests of FORCE learning with chaotic RNNs. **A** Reconstruction accuracy of the projected parameters. Arrows distinguish results from dRNNs trained with FORCE ((Sussillo & Abbott, 2009; Perich et al., 2021)) and CORNN ((Dinc et al., 2023)). **B** Frobenius norm of the parameters projected along the spectral components. The boxed region highlights the convergence of FORCE's Frobenius norms to the initialization values (horizontal dashed black line). **C** Pearson correlations between the projections of the initial and FORCE-trained parameters. For regularization levels resulting in learning, FORCE does not update the majority of parameters beyond their initial values. **D-F** We reanalyze the experiment from (Qian et al., 2024, Figure 5b), where a generator RNN with only two non-zero (oscillatory, decaying) eigenvalues was reconstructed with a dRNN under partial observation. **D** Top two principal component projections of neural activity for 15 distinct initializations across ground truth generator RNN, rank-2 dRNN, and full-rank dRNN. $\ell_2$ regularization enabled recovery of the decaying spiral. **E** Corresponding eigenvalues of the generator RNN and the reconstructed dRNNs. **F** Flow of the two largest magnitude eigenvalues as a function of regularization strength. Parameters: see Appendix S3.

et al., 2022; Beiran et al., 2021), we next trained dRNNs by minimizing the $\ell_2$ loss function defined on the single step prediction errors, with and without rank constraints.

When dRNNs were trained with negligible weight regularization ($\lambda \approx 10^{-13}$), both low-rank and full-rank dRNNs spuriously generated limit cycles (Fig. 3**D**). Examining the eigenvalues confirmed this failure (Fig. 3**E**): neither model matched the ground truth in this regime consistent with the observations of (Qian et al., 2024). Nevertheless, introducing an $\ell_2$ penalty on the weights corrected this behavior, allowing both low-rank and full-rank models to recover the spiraling dynamics. Tracking the two dominant eigenvalues under increasing regularization strength (Fig. 3**F**) further revealed that low-rank (but not full-rank) dRNNs rapidly suppressed the oscillatory modes by collapsing them into a non-oscillatory form.

These results demonstrate that low-rank constraints alone do not resolve incorrect estimation and can bias the learned dynamics toward oversimplified solutions. (See Fig. S5 for another experiment, where observing about 10% of all neurons could mitigate the identifiability concerns.) A corollary of Theorem 1 presented in Appendix S2.1.5 suggests the identifiability issues in dRNNs persist under low-rank assumptions. While a more detailed theoretical study remains an important direction for future work, the present findings highlight that identifiability limitations imposed by the (lack of) richness of the dataset cannot be circumvented simply by enforcing low-rank structure on the dRNN, and weight regularization is a necessary component despite its omission from earlier case studies (Qian et al., 2024).

## 2.6 A BLUEPRINT FOR TRAINING ONLY THE IDENTIFIABLE PARAMETERS IN DRNNS

So far, we studied three common estimators used for dRNN training. We observed that FORCE training, the most commonly used method, introduced spurious parameters to the learned RNNs,

whereas CORNN and low-rank RNN training both required weight regularization to succeed in suppressing the non-identifiable components in estimated parameters. Now, we combine these insights into a blueprint for identifiable training of dRNNs:

**Theorem 2** (Blueprint for estimating identifiable dRNN parameters). *Consider a dRNN following Eq. 1 whose parameters $\theta \in \mathbb{R}^{N \times N_{\text{tot}}}$ is estimated by gradient descent of a differentiable loss $\mathcal{L}(\theta)$. Let $X = \sum_{r=1}^{R} \sqrt{T} \sigma_r u^{(r)} v^{(r)T}$ be the singular value decomposition of the observation matrix $X$ with $\text{rank}(X) = R$, $u^{(r)} \in \mathbb{R}^T$, and $v^{(r)} \in \mathbb{R}^{N_{\text{tot}}}$. Define $P_K = \sum_{r=1}^{K} v^{(r)} v^{(r)T}$, i.e., the projection matrix to the top $K$ spectral components of the Gram matrix. Assume that the gradient satisfies $[\nabla \mathcal{L}(\theta) v^{(r)}]_a = O(\sigma_r^n)$ for every entry $a = 1, \dots, N$, modes $r = 1, \dots, K$, any $\theta \in \Theta$, and some positive integer $n$. If $\theta^{(s)} P_K = \theta^{(s)}$ at iteration $s$ of the learning, then for any $\lambda$ satisfying $\lambda \gg \sigma_{K+1}^2$, and for any step size $\alpha > 0$, the update*

$$\theta^{(s+1)} = \theta^{(s)} - \alpha \nabla L(\theta) \left( \frac{1}{T} X^T X + \lambda I \right)^{-1},\tag{5}$$

*is a descent direction that satisfies $\theta^{(s+1)} P_K = \theta^{(s+1)} + O(\sigma_{K+1}^n / \lambda)$. (Proof in Appendix S2.2.2.)*

Theorem 2 explains and generalizes our observations above. CORNN's learning rule has the form in Eq. 5 with $\lambda$ playing the role of weight regularization. FORCE updates can be written as Eq. 5 in the large $T$ limit, but $\theta^{(0)} P_K = \theta^{(0)}$ condition is not satisfied. For a general estimator, Eq. 5 simply suggests parameters updates aligned with the top $K$ components chosen effectively by $\lambda$.

Building on this construction, we test how different estimators behave in practice when trained on finite, noisy datasets. These datasets are generated using RNNs trained on various behavioral tasks (refer to Appendix S2.4 for full methodological details). Figure S6 compares four approaches: CORNN, second-order cross-entropy minimization, first-order cross-entropy (Adam), and a standard $\ell_2$ loss. Each method minimizes single-step prediction errors, but differs in optimization and loss formulation. All algorithms, when regularized properly, predicted correctly the linear combinations of parameters aligned with the top spectrum of the Gram matrix. Fig. S7 studies the effects of spatiotemporally correlated noise, whereas Fig. S8 studies the effect of mismatches in time constants $\tau$ (a form of model mismatch). In both cases, more trials are needed for accurate estimation.

### 2.7 IDENTIFIABLE COMPONENTS INDUCE PRESERVED DYNAMICS ACROSS DRNNS

The next theorem formalizes the notion that parameter differences confined to the non-identifiable directions make the same dynamics predictions in some neural activity subspace (Fig. 4**A**):

**Theorem 3** (Preserved dynamics in identifiable neural activity subspaces). *Let $S_{\text{id}}(R) = \text{span}\{v_1, \dots, v_R\}$ be the identifiable neural activity subspace spanned by the top $R$ spectral eigenvectors of the Gram matrix (or $S_{\text{id}}$ in short), and assume that for a noiseless, task-performing RNN with dynamics in Eq. 1, the activities satisfy $r[t] \in S_{\text{id}}(R)$ for all $t$. Let $\tilde{\theta}$ be identifiable with $\tilde{\theta} P_{\text{id}} = \tilde{\theta}$, where $P_{\text{id}}$ projects onto $S_{\text{id}}$. Then, any parameterization $\theta = \tilde{\theta} + \Delta \theta$ with $\Delta \theta P_{\text{id}} = 0$ but $\Delta \theta \neq 0$ yields identical dynamics $\dot{r}[t]$ for all $r[t] \in S_{\text{id}}$, but not necessarily when $r[t] \notin S_{\text{id}}$. (Proof in Appendix S2.3.)*

Practically, this theorem suggests that to constrain dynamics in a $K$-dimensional activity-input subspace, only $K$ (noiseless) training samples are sufficient. We illustrate this on a simple RNN with two neurons implementing a limit cycle in Fig. 4**B**. Here, only two (moderately noisy) samples were sufficient to estimate the dynamics predictions correctly on the full two-dimensional activity plane.

*A case study of low-dimensional parameter subspaces driving RNN dynamics:* Another important and widely discussed aspect of task-trained RNNs is solution degeneracy, referring to the existence of many different parameter configurations that achieve similar task performance (Huang et al., 2025; Cao & Yamins, 2024). Such degeneracy can arise from distinct computational strategies that solve the same task in qualitatively different ways (Kurtkaya et al., 2025), which is fundamentally different from a potential redundancy created by non-identifiable components of RNNs that effectively use the same solution, shared by the same identifiable parameters. The latter possibility highlights the need to distinguish which components of the parameter space are truly task-relevant and identifiable. With the assumption that neural activities are dominated by task-relevant dynamics as RNNs are solving behavioral tasks, Theorem 3 suggests one powerful tool to recover these

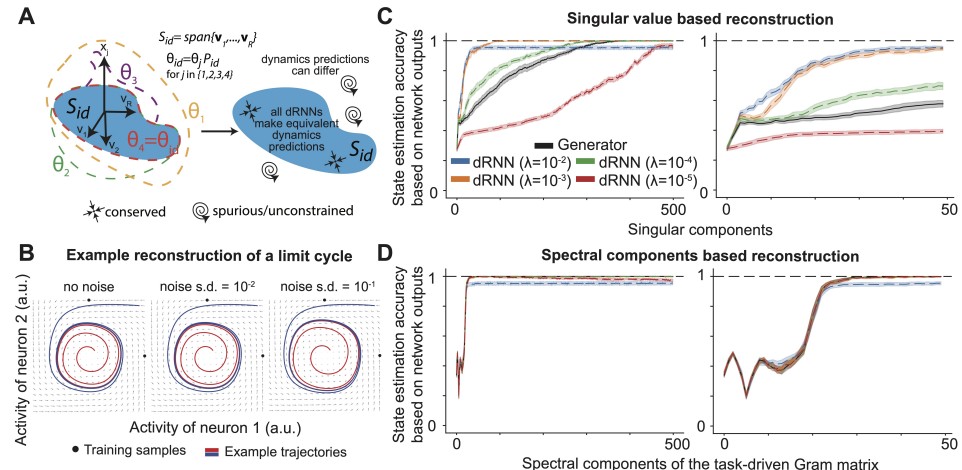

Figure 4: **Top spectral components encode parameter subspaces driving preserved dynamics.**
**A** An illustration of Theorem 3; the linear subspace spanned by the training samples defines the preserved subspaces where dRNNs make the same predictions of neural dynamics. **B** We reconstructed activities of an RNN ($N = 2$ neurons) implementing a limit cycle. **C-D** We next analyzed generator RNNs performing 3-bit flip flop tasks. **C** We reconstructed the weight matrix in each dRNN using the top $K$ singular components and computed the corresponding state estimation accuracy from the dRNN output. Plots show the output accuracies as a function of full (left) or close-up (right) spectrum. **D** Same as in **C**, but for projections onto top $K$ spectral components of a generalized task-driven Gram matrix. Parameters: see S3

parameters: Design a task-driven Gram matrix by collecting RNN activities across large number (*e.g.*, thousands) of trials, whose top eigenvectors would recover task-driven dynamics if they are low-dimensional. We test this idea in Fig. 4**C-D** with RNNs trained to perform 3-bit flip flop tasks, which are known to learn low-dimensional dynamics (Sussillo & Barak, 2013). Here, the network receives three separate binary input streams, each of which can flip or hold the value of an independent memory bit. This requires the RNN to maintain one of $2^3 = 8$ possible internal states. The networks state is output through three linear readouts.

We found that singular value decomposition did not lead to a low-dimensional parameter set responsible for the task-training (Fig. 4**C**). Hence, learned parameters were in no ways low-dimensional by nature. However, parameters projected to the top $\sim 10$–20 spectral components subserved the dynamics responsible for task operation. These parameters were also accurately reconstructed by dRNNs trained on the neural activities of the generator RNNs (Figs. S9), which also accurately solved the task (Fig. 4**C-D**). This suggests that task-relevant information in these networks was embedded in a restricted subset of spectral modes rather than being distributed across all parameters, and this subset was possible to extract from preserved dynamics across trials following Theorem 3.

### 2.8 REVEALING NON-IDENTIFIABLE COMPONENTS WITH TARGETED INTERVENTIONS

Parameters whose dynamical predictions are not represented in the training data cannot be recovered. We quantify this for dRNN training in terms of the zero-eigenvalue modes of the Gram matrix. Their resolution is only achievable with deliberate experimental interventions, which we next illustrate.

*Intervention experiment:* We focus on dRNNs trained to replicate the neural activities of generator RNNs trained on the 3-bit flip-flop task (Fig. 5). First, we use the generator RNNs to create an observational dataset over 5 distinct trials and compute the Gram matrix. We then construct an intervention dataset by sampling new training samples in four ways: (i) additional trials of the RNN under normal operation (blue in Fig. 5), (ii) projections restricted to the bottom spectral eigenvectors (green), (iii) projections restricted to the top spectral eigenvectors (orange), or (iv) random projections along the spectral components of the Gram matrix (red). Finally, we train dRNNs on the different combined datasets, and analyze reconstruction accuracies of the RNN parameters.

*Intervention results showcase the empirical utility of the Gram matrix spectrum:* Intervention strategy determines the performance of the dRNNs (Fig. 5**A-B**). When 500 intervention samples are

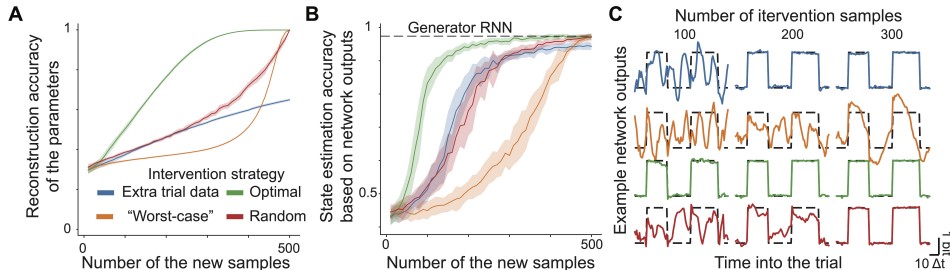

Figure 5: **Targeted interventions can expand the non-identifiable parameters in dRNNs.** We trained generator RNNs to perform the 3-bit flip-flop task and collected training samples. **A** Reconstruction accuracy of $\theta$ as a function of the number of new (intervention or extra) samples added to the training dataset. **B** State estimation accuracy, measured as the agreement between target flip-flop states and those predicted by dRNN outputs. In **A-B**, solid lines indicate the mean and shaded regions the s.d. across 20 RNNs. **C** Example outputs from trained RNNs for a single trial spanning 100 time points; dashed black lines are the ideal outputs. Parameters: Same as in Fig. 3**C-D**. Intervention strategies are color-coded and described in the main text.

available, all three Gram-matrix-based strategies (but not the one involving extra observational samples; blue in Fig. 5**A**) achieve near-perfect accuracies (Fig. S10). Interventions along the top eigenvectors provided little to no benefit ("worst-case"), whereas selecting the bottom eigenvectors ("optimal") accelerated recovery relative to random choices. Increasing the number of interventions increasingly aligned dRNN outputs with the ground truth flip-flop states (Fig. 5**C**). Even though the optimal strategy (green) does not involve samples encountered during task-relevant operation, dRNNs trained with these samples out-performed those trained on more task-relevant activities.

## 3 DISCUSSION AND CONCLUSION

In this work, we presented a set of theoretical results for assessing the reliability of dRNNs trained on observed neural trajectories. By connecting parameter identifiability to dynamical predictions made by dRNNs, we provide both theoretical guarantees and practical guidance for using these models as digital twins of neural circuits (Perich & Rajan, 2020).

Each theorem formalizes a key insight: Theorem 1 suggests that non-identifiable components cannot be resolved from limited data. Theorem 2 quantifies the differences between existing training methods and opens up new ways to design new estimators. Theorem 3 delineates boundaries where dynamical predictions can be trusted, and hints at why dRNNs in Eq. 1 have been empirically powerful: Their training is data-efficient, with each noiseless (or few noisy) sample constraining dynamics on the whole subspace. These insights led to an intervention strategy for enriching the training datasets, which opens avenues for experimental designs that systematically expand reliable prediction spaces, advancing our understanding of neural computation.

The practical value of parameter identifiability emerges when considering experimental validation costs. Suppose a dRNN predicts two attractors underlying different behaviors, each requiring months of single-cell optogenetics to test (Liu et al., 2024). Which should be prioritized? Our results provide a simple rule: trust predictions in subspaces with larger spectral components. If an attractor exists in $S_{\mathrm{id}}(R)$ for small $R$ and survives cross-validated regularization, it is presumably constrained by observed data (assuming Eq. 1 reasonably approximates neuronal processes). While empirical validation remains the gold standard, our theorems suggest an internally consistent method to rule out experiments based on unconstrained predictions.

Broadly, several empirical concerns affect data-driven models, and dRNNs are no exception. We showed that nonstandard noise and model mismatches demand more trials, while recording more neurons is needed to minimize biases from unobserved influences (Brinkman et al., 2018). These concerns have led to rather pessimistic theoretical assessments of the utility of data-constrained models (Das & Fiete, 2020; Qian et al., 2024). However, recent technological (Kim & Schnitzer, 2022; Manley et al., 2024) and computational (Linderman et al., 2017) advances have enabled successful causal predictions (Walker et al., 2019; Liu et al., 2024). Thus, these challenges are becoming surmountable. Our work suggests that one just needs to understand which predictions to trust.

## LIMITATIONS

While our work establishes a general theoretical framework for identifiability in dynamical recurrent neural networks, several limitations remain that should be acknowledged and that point to concrete directions for future research.

First, there is likely a simple but important connection to Takens' theorem in dynamical systems theory (Takens, 2006), which posits that the attractor of a dynamical system can be reconstructed from time-delay embeddings of a generic observable. We did not explore this direction here, but it is plausible that introducing delayed embeddings into our framework could further strengthen the identifiability results and provide a complementary perspective to our Gram-based analysis.

Second, while we studied low-rank RNNs and influences of unobserved neurons, these analyses were intended primarily to complement our central results on dRNNs. A more complete theory in these domains remains to be developed and represent natural and important extensions of our work.

Third, following established practice in the field (Das & Fiete, 2020; Qian et al., 2024) and for clarity of presentation, our paper is intentionally limited to theory and controlled synthetic experiments. While dRNNs have been applied to real neural recordings many times (Perich et al., 2021; Valente et al., 2022), we chose not to pursue such applications here. Beyond the practical issue of dataset access and additional complications associated with (somewhat nonstandard (Rajan et al., 2016; Perich et al., 2021; Valente et al., 2022)) preprocessing of neural activities, we believe little is to be gained scientifically from training one more RNN on these datasets without causal perturbations that can only be performed in experimental settings.

Finally, consistent with this view, Theorems 1 and 2 are best illustrated in simulated datasets where the ground truth is known. On the other hand, two key empirical applications of our theory remain practically untested and will likely remain so until single-cell level interventions become mainstream and instant. Testing Theorem 3 and the proposed interventions requires not just observational data but direct empirical evaluations at the level of individual neurons, which may take years to develop (Vinograd et al., 2024; Liu et al., 2024). We hope that future work will use our framework to rapidly discard inconsistent hypotheses (*e.g.*, perturbation predictions that result from non-identifiable components) and to design closed-loop intervention experiments that directly test Theorem 3. Such experiments would provide a stringent evaluation of our theory and clarify how identifiability constraints limit inference from real neural recordings.

Finally, we acknowledge the use of large language models for copyediting and grammar corrections, as well as simplification of jargon in several places of our writing.

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

## S1 EXTENDED RELATED WORKS

In the main text, we summarized prior work on RNNs as models of neural activity, as well as the general identifiability of RNNs and nonlinear systems.

**Dynamical models of neural activity.** A central premise of computational neuroscience is that computational models that reproduce neural activity will provide biological insight. However, recovering synaptic connectivity or precise mechanisms from dynamics alone is generally ill-posed (Das & Fiete, 2020; Brinkman et al., 2018), and even functional properties such as attractors can be unreliable when inferred from observational data alone (Qian et al., 2024; Göring et al., 2024).

Despite these limitations, predictive models have generated potentially meaningful results. RNNs trained on neural trajectories have been shown to capture features such as population-level gating (Finkelstein et al., 2021), inter-area communication motifs (Perich & Rajan, 2020), and low-dimensional attractor dynamics (Valente et al., 2022). Several of these predictions have been refined and confirmed through causal perturbations (Daie et al., 2021; Liu et al., 2024; Vinograd et al., 2024; Walker et al., 2019), demonstrating that data-driven models can sometimes generate testable mechanistic hypotheses. In an effort to preserve biological interpretability, some studies have trained "data-constrained" RNNs with a one-to-one mapping between model units and recorded neurons (Perich & Rajan, 2020; Perich et al., 2021; Dinc et al., 2023). This approach aims to avoid confounds introduced by hidden units and to estimate functional connectivity directly. However, even in these restricted settings, little has been studied about the identifiability of parameters, leaving open the question of whether different underlying models can equally explain the same data.

**Identifiability in nonlinear systems.** The broader control and systems literature provides a foundation for understanding when models can be uniquely determined from observed behavior. Classical realization theory shows that any finite-dimensional system's external behavior can be represented by a minimal, unique system if it is both controllable and observable (Sussmann, 1976). In this framework, two systems are indistinguishable if they generate the same outputs for all inputs, and minimal models are live in the quotienting the parameter space of the original model with this equivalence relation. Complementary results come from dynamical systems theory. Takens' embedding theorem (Takens, 2006) guarantees that, given a sufficiently large embedding dimension, the dynamics of a system can be reconstructed from time-delayed measurements of even a single observable (Schmid, 2010). This provides theoretical justification for reconstructing dynamics from partial observations, as is common in neuroscience. Yet in practice, neural data often violate these assumptions. Activations are highly redundant and typically lie in a low-dimensional subspace (Dubreuil et al., 2020; Perich et al., 2025), undermining identifiability.

**Identifiability of dynamical systems.** The question of whether models are uniquely determined by data, i.e., whether they are *identifiable*, has long been studied in control theory. Classical realization theory results show that any external behavior generated by a finite-dimensional system can be represented by a "minimal" and unique system, which must be controllable and observable (Sussmann, 1976). Such a minimal model can often be found by restricting the parameters to the quotient space of the original model space and the equivalence relation of indistinguishability, or, equivalently, by reducing the state space to the manifold occupied by the lower-dimensional underlying system (Crouch, 1979; Brockett, 2005). For neural networks specifically, identifiability has been examined under specific conditions (Sussmann, 1992; Poznyak et al., 2001; Albertini & Sontag, 1993). This analysis excluded "degenerate situations", such as those with parameter dependencies, nonobservability, and underlying low-dimensionality–all of which occur in real-world neural data (Dubreuil et al., 2020; Perich et al., 2025). Recent studies have highlighted how RNN dynamics are only partially constrained by partial input–output observations (Rajan et al., 2010; Kepple et al., 2022), leading to parameter ambiguity. Such studies have proposed frameworks to measure, understand, and intervene on solution degeneracy in task-trained RNNs (Huang et al., 2025).

**Identifiability in neural networks.** Identifiability in neural networks has been studied for decades, though usually under restrictive assumptions. For recurrent architectures with linear or smooth nonlinear activations (such as tanh), input–output mappings can constrain parameters up to permutation symmetries, except in degenerate situations caused by dependencies, nonobservability, or noncontrollability (Sussmann, 1992; Poznyak et al., 2001; Albertini & Sontag, 1993; Albertini

et al., 1994; Sontag, 2013). Real neural data, however, are precisely such degenerate cases: redundancy and low dimensionality leave entire parameter directions unconstrained.

Recent work has formalized these issues in both recurrent and feedforward networks. For example, distinct connectivity matrices in piecewise-linear RNNs can produce identical steady states (Biswas & Fitzgerald, 2022), and equivalence classes of minimal, identifiable systems have been defined for restricted classes of RNNs (Al-Falou & Trummer, 2003). Parallel efforts have analyzed parameter symmetries in feedforward networks, especially with ReLU nonlinearities (Bui Thi Mai & Lampert, 2020; Bona-Pellissier et al., 2023).

**Solution degeneracy in task-trained RNNs.** Within neuroscience and machine learning, non-identifiability is often discussed under the broader notion of solution degeneracy. Input-driven constraints shape RNN dynamics, but leave ambiguity (Rajan et al., 2010), and partial observability creates challenges for learning and inference (Kepple et al., 2022). More recently, (Huang et al., 2025) introduced a framework to quantify and control solution degeneracy in task-trained RNNs, showing that variability across solutions depends on model capacity and task complexity. Their results highlight the need for interventions to disambiguate latent mechanisms, as multiple parameterizations can fit the same task, using different mechanisms.

## S2 METHODS

### S2.1 A FRAMEWORK FOR ASSESSING PARAMETER IDENTIFIABILITY

In dynamical system models, the prediction depends not only on the parameters $\theta$, but also on the current state of the system and any external inputs. Moreover, unlike the traditional estimation problem, changing dynamical system parameters using data from a particular time point affects the future states, *i.e.*, output of one estimation becomes input of another one. Here, we first formally define a notion of identifiability concerning the parameters of dynamical systems and then present our main result on dRNNs defined by Eq. 1. Then, we conclude with extensions to another common RNN architecture and low-rank RNNs, and considerations of partial observations.

### S2.1.1 CONDITIONAL IDENTIFIABILITY IN DYNAMICAL SYSTEM MODELS

Intuitively, identifiability is about whether you can uniquely determine the parameters of a model from the observed data. If a model is identifiable, then, given enough data, there is only one set of parameters that could produce that data. If a model is non-identifiable, then there are multiple different sets of parameters that could produce the same observations. We start by citing a formal definition of this problem:

**Definition S1** (Identifiability (Lehmann & Casella, 2006)). *Let $\mathcal{P} = \{P_\theta : \theta \in \Theta\}$ be a model, or family of parameterized probability distributions, with parameter space $\Theta$. $\mathcal{P}$ is identifiable if and only if the mapping $\theta \mapsto P_\theta$ is injective, i.e., if*

$$P_{\theta_1} \hat{=} P_{\theta_2} \quad \Rightarrow \quad \theta_1 = \theta_2 \quad \text{for all } \theta_1, \theta_2 \in \Theta, \tag{S1}$$

*where $\hat{=}$ means equal in distribution.*

Identifiability of $\theta$ can often be achieved under certain "identification conditions." For instance, for a family of distributions $P_\theta$ that satisfy the condition $P_\theta \hat{=} P_{-\theta} \hat{=} P_{|\theta|}$, one may enforce $\theta \geq 0$ as an identification condition. In Definition S1, the probability distribution $P_\theta$ is defined over the observation space $\mathcal{Y}$, from which we often collect samples $\{Y_1, \ldots, Y_T\}$ with $Y_i \in \mathcal{Y}$ for $i = 1, \ldots, T$. $Y_i$ can be scalar or a vector (or something else), depending on the problem of interest. For our purposes, $Y_i \in \mathbb{R}^N$ is the vector containing the neural activities corresponding to a particular time $t = i\Delta t$, where $\Delta t$ is a time step used for discretization.

A key distinction with dynamical system models is that the parameter $\theta^*$ that generates a neural trajectory requires the observation of an auxiliary variable, $X_i \in \mathbb{R}^{N_{\text{tot}}}$ where $N_{\text{tot}} = N + N_{\text{in}}$. Moreover, while one could consider the pair, $(X_i, Y_i) \in \mathcal{X} \times \mathcal{Y}$ as a viable sample, the target value $Y_i$ constitutes part of the auxiliary variable $X_{i+1}$ by design. To see why, see Eq. 1 and recall that $x(t)$ is defined as the concatenated vector $[r(t), u(t)]$, whereas $y(t)$ corresponds to $r(t + \Delta t)$. Thus, we cannot directly apply Definition S1 to our problem of interest.

The solution to this dilemma comes from two distinct observations. One, our goal is to find a set of parameters such that all $\theta$ make the same predictions on the *whole* neural trajectory, not just at the end where RNN outputs are traditionally taken from. Thus, we enforce that the model correctly predicts all time steps accurately, in which scenario the data generation process can be written as a single-step prediction error on observed neural-inputs $X_i \in \mathcal{X}$:

$$Y_i \sim P(Y_i|X_i; \theta), \tag{S2}$$

where we observe the pairs $(X_i, Y_i)$ for $i = 1, \ldots, T$. Here, $P(Y_i|X_i; \theta)$ refers to the conditional distribution of $Y_i$ given $X_i$, parameterized by the deterministic parameter values $\theta$. As a second observation, Definition S1 can be extended to a case where $P_\theta$ is replaced with this conditional distribution:

**Definition S2** (Conditional Identifiability). *Let $\mathcal{P} = \{P(\cdot|\cdot; \theta) : \theta \in \Theta\}$ be a statistical model with parameter space $\Theta$. Let the ground truth data generation process follow the distribution $Y \sim P(Y|X; \theta^*)$ for some unknown $\theta^*$, where $(X, Y) \in \mathcal{X} \times \mathcal{Y}$ refers to the observable samples. We say that $\mathcal{P}$ is conditionally identifiable if the mapping $\theta \mapsto P(\cdot|\cdot; \theta)$ is one-to-one for all possible values of $X$:*

$$\forall(X, Y) \in \mathcal{X} \times \mathcal{Y} \quad P(Y|X; \theta_1) \hat{=} P(Y|X; \theta_2) \implies \theta_1 = \theta_2. \tag{S3}$$

The key distinction of this definition is that we do not enforce $X, Y$ to be observable from the full Euclidean space. Hence, we replace the joint distribution in Eq. S1 with a conditional one in Eq. S3, which then can be practically operationalized using the observed samples. Specifically, in Definition 1, we use a practical version of Definition S2 for studying parameter identifiability in dRNNs, where $\forall(X, Y) \in \mathcal{X} \times \mathcal{Y}$ is replaced with $(X_i, Y_i)$ for the samples $i = 1, \ldots, T$.

Before we conclude this formal discussion, we briefly note that in the statistics literature, conditional identifiability sometimes refers to identifiability under additional identification conditions. In contrast, here we use the term to denote identifiability with respect to conditional probability distributions.

### S2.1.2 Modeling noisy neural activities with dRNNs

In the main text, when introducing the dRNNs in Eq. 1, we omitted a third type of noise: the observation noise. Here, we show that this was without a loss of generality.

In a general case, one might (correctly) argue that a realistic experimental setup should include an observation noise term. Specifically, the neural trajectories that one aims to reproduce may be incorrectly observed, *i.e.*, instead of the true $r(t)$, one might observe a noisy version $\tilde{r}(t)$ such that

$$\tilde{r}(t) = r(t) + \sigma(t), \tag{S4}$$

where $\sigma(t)$ is some unknown observation error term associated with the observation. Fortunately, for the dRNNs in Eq. 1, there is no reason to explicitly incorporate this observation noise into the data generation process as it is already accounted by other two noise terms. We formalize this with a remark:

**Remark** (Observation error). *Assume that $r(t)$ is observed incorrectly following Eq. S4. Assume that $r(t)$ follows Eq. 1 with the pair $\{\epsilon_{\mathrm{in}}, \epsilon_{\mathrm{conv}}\}$. Then, $\tilde{r}(t)$ evolves via Eq. 1 corrupted by the modified terms $\hat{\epsilon}_{\mathrm{conv}} = \epsilon_{\mathrm{conv}} + \tau\dot{\sigma}(t) + \sigma(t)$ and $\hat{\epsilon}_{\mathrm{in}} = \epsilon_{\mathrm{in}} - W^{\mathrm{rec}}\sigma(t)$.*

*Proof.* The proof follows by taking the derivative $\tau\dot{\tilde{r}}(t)$ and plugging $\dot{r}(t)$ from Eq. 1 into the right hand side:

$$\begin{aligned} \tau\dot{\tilde{r}}(t) = \tau\dot{r}(t) + \tau\dot{\sigma}(t) &= -r(t) + \phi(W^{\mathrm{rec}}r(t) + W^{\mathrm{in}}u(t) + \epsilon_{\mathrm{in}}(t)) + \epsilon_{\mathrm{conv}}(t) + \tau\dot{\sigma}(t), \\ &= -\tilde{r}(t) + \phi(W^{\mathrm{rec}}\tilde{r}(t) + W^{\mathrm{in}}u(t) + \epsilon_{\mathrm{in}}(t) - W^{\mathrm{rec}}\sigma(t)) + \epsilon_{\mathrm{conv}}(t) + \sigma(t) + \tau\dot{\sigma}(t). \end{aligned} \tag{S5}$$

Then, defining $\hat{\epsilon}_{\mathrm{conv}} = \epsilon_{\mathrm{conv}} + \tau\dot{\sigma}(t) + \sigma(t)$ and $\hat{\epsilon}_{\mathrm{in}} = \epsilon_{\mathrm{in}} - W^{\mathrm{rec}}\sigma(t)$ concludes the proof. $\square$

Intuitively, even if the observed firing rates $\tilde{r}(t)$ differ slightly from the ground truth $r(t)$, they follow the same time evolution equations up to a re-definition of the noise terms. Thus, as long as the empirical estimation procedure for $\theta$ under the data generation model given in Eq. 1 is robust to input and conversion noise terms, no explicit modeling of the observation noise is necessary.

### S2.1.3 PROOF OF THEOREM 1: IDENTIFIABILITY IN dRNNs

Before proving Theorem 1, we first start with a simple yet powerful lemma:

**Lemma S1.** *Consider a data generation model:*

$$d \sim \phi(z + \epsilon_{\text{in}}) + \epsilon_{\text{conv}}, \tag{S6}$$

*where $z$ is deterministic and $\epsilon_{\text{in/conv}}$ are random variables. Assume $\phi(\cdot)$ is a strictly increasing monotonic non-linearity, and let the noise random variables $\epsilon_{\text{in}}$ and $\epsilon_{\text{conv}}$ be independent, with the latter having a non-vanishing characteristic function. Then, $P(d|z_1) = P(d|z_2)$ if and only if $z_1 = z_2$.*

*Proof.* The reverse direction of the if and only if statement is trivial, since $z_1 = z_2$ trivially implies $P(y|z_1) = P(y|z_2)$. For the forward direction, assuming $P(y|z_1) = P(y|z_2)$, we need to show that $z_1 = z_2$. We start by recalling that the characteristic function of a random variable $X$ is defined as $\varphi_X(\omega) = \mathbb{E}[e^{i\omega X}]$ for $\omega \in \mathbb{R}$. Since the probability distributions are equal, their characteristic functions are equal. Therefore:

$$\varphi_{y|z_1}(\omega) = \varphi_{y|z_2}(\omega) \quad \forall \omega \in \mathbb{R} \tag{S7}$$

Given the data generation model, for any fixed $z$:

$$\varphi_{y|z}(\omega) = \mathbb{E}[e^{i\omega y}|z] = \mathbb{E}[e^{i\omega(\phi(z+\epsilon_{\text{in}})+\epsilon_{\text{conv}})}] \tag{S8}$$

Since $\epsilon_{\text{in}}$ and $\epsilon_{\text{conv}}$ are independent:

$$\varphi_{y|z}(\omega) = \mathbb{E}[e^{i\omega\phi(z+\epsilon_{\text{in}})}] \cdot \mathbb{E}[e^{i\omega\epsilon_{\text{conv}}}] = \mathbb{E}[e^{i\omega\phi(z+\epsilon_{\text{in}})}] \cdot \varphi_{\epsilon_{\text{conv}}}(\omega) \tag{S9}$$

Therefore, from our assumption:

$$\mathbb{E}[e^{i\omega\phi(z_1+\epsilon_{\text{in}})}] \cdot \varphi_{\epsilon_{\text{conv}}}(\omega) = \mathbb{E}[e^{i\omega\phi(z_2+\epsilon_{\text{in}})}] \cdot \varphi_{\epsilon_{\text{conv}}}(\omega) \tag{S10}$$

Since $\varphi_{\epsilon_{\text{conv}}}(\omega)$ is non-vanishing (*i.e.*, $\varphi_{\epsilon_{\text{conv}}}(\omega) \neq 0$ for all $\omega$), we can divide both sides:

$$\mathbb{E}[e^{i\omega\phi(z_1+\epsilon_{\text{in}})}] = \mathbb{E}[e^{i\omega\phi(z_2+\epsilon_{\text{in}})}] \quad \forall \omega \tag{S11}$$

This equality of characteristic functions implies equality of distributions:

$$\phi(z_1 + \epsilon_{\text{in}}) \hat{=} \phi(z_2 + \epsilon_{\text{in}}) \tag{S12}$$

Since $\phi(\cdot)$ is strictly increasing and thus injective, we have:

$$z_1 + \epsilon_{\text{in}} \hat{=} z_2 + \epsilon_{\text{in}} \tag{S13}$$

This means the random variables $z_1 + \epsilon_{\text{in}}$ and $z_2 + \epsilon_{\text{in}}$ have the same distribution. Since $z_1$ and $z_2$ are deterministic constants and $\epsilon_{\text{in}}$ is the same random variable in both expressions, this is only possible if $z_1 = z_2$. Therefore, $P(y|z_1) = P(y|z_2)$ implies $z_1 = z_2$, completing the proof. $\square$

We now prove our main Theorem 1, which formalizes the parameter identifiability in dRNNs:

**Theorem** (Restatement of Theorem 1). *Consider an RNN defined by Eq. 1 with parameters $\theta^* \in \mathbb{R}^{N \times N_{\text{tot}}}$, where the noise random variables $\epsilon_{\text{in}}$ and $\epsilon_{\text{conv}}$ are independent, with the latter having a non-vanishing characteristic function. Consider an observation matrix $X \in \mathbb{R}^{T \times N_{\text{tot}}}$ defining the domain $\mathcal{X}$ and denote $P_{\mathcal{X}} \in \mathbb{R}^{N_{\text{tot}} \times N_{\text{tot}}}$ the projection matrix onto its row space. Then, any discretized RNN parametrized by $\theta$ such that:*

$$\theta P_{\mathcal{X}} = \theta^* P_{\mathcal{X}}, \tag{S14}$$

*gives the same conditional probability distribution on single-step predictions as the ground-truth RNN and vice versa. Then, $\theta^* P_{\mathcal{X}}$ (out of all RNN parameters) is identifiable if and only if the parameter space is restricted to $\Theta_{\mathcal{X}} = \{\theta \in \mathbb{R}^{N \times N_{\text{tot}}} : \theta = \theta P_{\mathcal{X}}\}$ (identification condition). In particular, an unrestricted $\theta^* \in \mathbb{R}^{N \times N_{\text{tot}}}$ is identifiable if and only if $P_{\mathcal{X}} = I$.*

*Proof.* The discretized RNN follows the set of equations $r_{t+1} = f(r_t) + \phi(\theta x_t + \epsilon_{\text{in,t}}) + \epsilon_{\text{conv,t}}$ where $x_t = [r_t, u_t]^T$. As a parallel to Lemma S1, define $d_t = r_{t+1} - f(r_t)$, which is an observed quantity.

First, we will prove that $\theta P_{\mathcal{X}} = \theta^* P_{\mathcal{X}}$ if and only if both parameters give identical conditional distributions on observable states. The forward direction is trivial. If $\theta P_{\mathcal{X}} = \theta^* P_{\mathcal{X}}$, then for any $x \in \text{row}(X)$, we have $x = P_{\mathcal{X}} x$, so $\theta x = \theta P_{\mathcal{X}} x = \theta^* P_{\mathcal{X}} x = \theta^* x$. Since the deterministic components are equal, Lemma S1 ensures $P_\theta(d_t|x_t) = P_{\theta^*}(d_t|x_t)$. For the reverse direction, suppose $P_\theta(d_t|x_t) = P_{\theta^*}(d_t|x_t)$ for all $x_t \in \text{row}(X)$. By Lemma S1, this implies $\theta x_t = \theta^* x_t$ for all such $x_t$. This is equivalent to $\theta P_{\mathcal{X}} = \theta^* P_{\mathcal{X}}$.

Second, we prove that $\theta^* P_{\mathcal{X}}$ is identifiable if and only if the parameter space is restricted to $\Theta_{\mathcal{X}} = \{\theta : \theta = \theta P_{\mathcal{X}}\}$. For the forward direction, suppose $\theta^* P_{\mathcal{X}}$ is identifiable without this restriction. Pick any $\Delta\theta \neq 0$ with $\Delta\theta P_{\mathcal{X}} = 0$. Then $\theta^* P_{\mathcal{X}} + \Delta\theta$ has the same projection: $(\theta^* P_{\mathcal{X}} + \Delta\theta) P_{\mathcal{X}} = \theta^* P_{\mathcal{X}}$. By the first statement above, $\theta^* P_{\mathcal{X}} + \Delta\theta$ gives the same conditional distributions as $\theta^* P_{\mathcal{X}}$. But $\theta^* P_{\mathcal{X}} + \Delta\theta \neq \theta^* P_{\mathcal{X}}$, contradicting identifiability unless $\Delta\theta = 0$ for any $\Delta\theta P_{\mathcal{X}} = 0$. This condition defines the restricted $\Theta_{\mathcal{X}}$. Now, consider the reverse direction. Suppose that $\theta_1 \in \Theta_{\mathcal{X}}$ gives the same distributions as $\theta_2$. By the first statement above, $\theta_1 P_{\mathcal{X}} = \theta_2 P_{\mathcal{X}}$, which (by the restriction of the parameter subspace) implies $\theta_1 = \theta_2$, *i.e.*, the parameter is unique. By inspection, $\theta^* P_{\mathcal{X}}$ gives the correct conditional distributions and is part of the restricted parameter family, hence $\theta_1 = \theta_2 = \theta^* P_{\mathcal{X}}$.

The final statement is a special case of the second statement proven above, concluding the proof. □

### S2.1.4 GENERALIZING TO A COMMON RNN ARCHITECTURE

Theorem 1 can be generalized to another commonly studied RNN architecture (Valente et al., 2022; Mastrogiuseppe & Ostojic, 2018; Sussillo & Abbott, 2009). First, we establish a direct link between these two architectures:

**Remark.** *Consider Eq. 1 without an input noise. After defining a coding variable as $z(t) = W^{\text{rec}} r(t) + W^{\text{in}} u(t)$ and re-defining the (pre-defined) inputs as $v(t) = u(t) + \tau \dot{u}(t)$, Eq. 1 transforms into:*

$$\tau \dot{z}(t) = -z(t) + W^{\text{rec}} \phi(z(t)) + W^{\text{in}} v(t) + \epsilon, \tag{S15}$$

*where $\epsilon = W^{\text{rec}} \epsilon_{\text{conv}}$ constitutes the (transformed) noise term.*

*Proof.* This remark has already been proven and published (Miller & Fumarola, 2012). Here, we simply retrace the arguments. Introducing $z(t) = W^{\text{rec}} r(t) + W^{\text{in}} u(t)$ into Eq. 1 and multiplying both sides with $W^{\text{rec}}$ leads to:

$$\tau \frac{\text{d}}{\text{d}t}(W^{\text{rec}} r(t)) = -W^{\text{rec}} r(t) + W^{\text{rec}} \phi(z(t)) + W^{\text{rec}} \epsilon_{\text{conv}}(t). \tag{S16}$$

We can rewrite the left hand side as:

$$\tau \frac{\text{d}}{\text{d}t}(W^{\text{rec}} r(t)) = \tau \frac{\text{d}}{\text{d}t}(W^{\text{rec}} r(t) + W^{\text{in}} u(t)) - \tau W^{\text{in}} \dot{u}(t) = \tau \dot{z}(t) - \tau W^{\text{in}} \dot{u}(t) \tag{S17}$$

Similarly, adding and subtracting $W^{\text{in}} u(t)$ for the right hand side leads to final solution:

$$\tau \dot{z}(t) = -z(t) + W^{\text{rec}} \phi(z(t)) + W^{\text{in}} [\tau \dot{u}(t) + u(t)] + W^{\text{rec}} \epsilon_{\text{conv}}(t). \tag{S18}$$

This reproduces Eq. S15, concluding the proof. □

This equation governs another commonly used RNNs for data-constrained training (Perich et al., 2021). In these models, we refer to the new variable, $z(t) \in \mathbb{R}^N$, as the "currents" and $v(t) \in \mathbb{R}^{N_{\text{in}}}$ shortly as (transformed) "inputs". Though the two equations are equivalent (up to a transformation of the input and redefinition of the state variables), Eq. S15 has been more regularly used in the field of computational neuroscience. We now provide an informal, yet necessary, discussion on why we choose the RNN architecture in Eq. 1 over Eq. S15 as the more suitable candidate for constraining neural trajectories by drawing analogies to the neural biology. This discussion may also explain the more frequent use of Eq. S15 over Eq. 1 in computational neuroscience to date.

One can consider $z(t)$, the state variable of Eq S15, as a current injected into an artificial neuron's soma, and $\phi(z(t))$ as the membrane potential and/or smoothed action potentials. Then, one can consider $r(t)$, the state variables of Eq 1 following the equations $\tau \dot{r}(t) = -r(t) + \phi(z(t)) + \epsilon_{\text{conv}}$, as the firing rates, smoothed averages of the neural spikes resulting from the injected currents. Since earliest experimental efforts often involved injecting currents into squid axons (Hodgkin & Huxley, 1952), it is not surprising that a computational view focusing on the currents as state variables (*i.e.*, Eq. S15) could have seen more intuitive over the years. On the other hand, recent experimental approaches allow brain-wide large-scale access to simultaneous firing rates up to millions of neurons (Urai et al., 2022; Kim & Schnitzer, 2022; Manley et al., 2024; Stringer et al., 2019). Hence, with the recent approaches aiming to reproduce these neural activities (Perich et al., 2021; Valente et al., 2022; Duncker & Sahani, 2021; Cohen et al., 2020; Finkelstein et al., 2021; Dinc et al., 2023; Qian et al., 2024), the form in Eq. 1 that places the firing rates as the fundamental state variable would be more directly applicable. Specifically, since the firing rates are often considered as the observables from these recordings, we use this form for our analysis in this work, and assume that $r(t)$, not $z(t)$, are the observables.

As the next corollary shows, apart from being computationally equivalent, Theorem 1 designed in Eq. 1 extends also to the architecture given in Eq. S15:

**Corollary S1** (Parameter identifiability in an equivalent RNN formulation). *Consider an RNN defined by Eq. S15 with parameters $\theta^* \in \mathbb{R}^{N \times N_{\text{tot}}}$. Consider an observation matrix $X \in \mathbb{R}^{T \times N_{\text{tot}}}$ defining the domain $\mathcal{X}$, in which we define each sample as $x_t = [\phi(z_t), v_t]$, and denote $P_{\mathcal{X}} \in \mathbb{R}^{N_{\text{tot}} \times N_{\text{tot}}}$ the projection matrix onto its row space. Then, any discretized RNN parametrized by $\theta$ such that:*

$$\theta P_{\mathcal{X}} = \theta^* P_{\mathcal{X}}, \tag{S19}$$

*gives the same conditional probability distribution on single-step predictions as the ground-truth RNN and vice versa. Then, $\theta^* P_{\mathcal{X}}$ (out of all RNN parameters) is identifiable if and only if the parameter space is restricted to $\Theta_{\mathcal{X}} = \{\theta \in \mathbb{R}^{N \times N_{\text{tot}}} : \theta = \theta P_{\mathcal{X}}\}$ (identification condition). In particular, an unrestricted $\theta^* \in \mathbb{R}^{N \times N_{\text{tot}}}$ is identifiable if and only if $P_{\mathcal{X}} = I$.*

*Proof.* Defining $x(t) = [\phi(z(t)), v(t)]$ and $d(t) = \tau \dot{z}(t) + z(t)$, we arrive at a linear model for data generation for a given time $t$:

$$d \sim \theta^* x + \epsilon. \tag{S20}$$

This is a linear model, and as such the proof follows from well known results in linear regression literature (Åström & Eykhoff, 1971) or by simply restating the logic used to prove Theorem 1. $\square$

Following the linear relationship stated in the proof of Corollary S1, equivalent versions of Theorems 2 and 3 could also be stated similarly.

### S2.1.5 PARAMETER IDENTIFIABILITY IN LOW-RANK RNNS

One might think that intrinsically low-rank nature of the identifiable subspace, as predicted by Theorem 1, could be enforced directly by training low-rank RNNs constrained on neural trajectories (Valente et al., 2022). In this approach, the recurrent weight matrix is factorized as $W^{\text{rec}} = CD$ with $C \in \mathbb{R}^{N \times K}$, $D \in \mathbb{R}^{K \times N}$, and $K \ll N$, so that only $O(KN)$ parameters are learned (Beiran et al., 2021; Dubreuil et al., 2022; Valente et al., 2022). This parametrization appears to align with the expectation that only a low-rank subset of parameters is identifiable. However, recent theoretical results reveal a crucial complication: even rank-one RNNs can generate neural trajectories that span the full $N$-dimensional space of activities (Dinc et al., 2025). In that sense, enforcing low-rank structure on $W^{\text{rec}}$ does not guarantee that the observed activity itself is low-dimensional, nor would it be expected to reduce the identifiability requirements of the system. To fully constrain the parameters, the observation conditions remain just as strict as in the full-rank case.

To resolve the apparent contradiction between these assertions, we first state a lemma that suggests a low-rank weight matrix suggests a low-rank $\theta$ matrix as long as $N_{\text{in}} \ll N$:

**Lemma S2** (Low-rank combined parameters). *If $W^{rec} \in \mathbb{R}^{N \times N}$ has rank $K \ll N$, then $\theta = [W^{rec}, W^{in}] \in \mathbb{R}^{N \times N_{tot}}$ has rank at most $K + N_{in}$, which remains low-rank as long as $N_{in} \ll N$.*

*Proof.* By the rank inequality for concatenated matrices:

$$\text{rank}(\theta) = \text{rank}([W^{\text{rec}}, W^{\text{in}}]) \leq \text{rank}(W^{\text{rec}}) + \text{rank}(W^{\text{in}}) \leq K + \min(N, N_{\text{in}}) = K + N_{\text{in}} \quad \text{(S21)}$$

Since $K \ll N$ and $N_{\text{in}} \ll N$, we have $\text{rank}(\theta) \leq K + N_{\text{in}} \ll N$. □

With this lemma, we now state and prove an extension of Theorem 1 to low-rank RNNs:

**Corollary S2** (Non-identifiability in low-rank RNNs). *Consider the conditions stated in Theorem 1. Suppose $\theta \in \mathbb{R}^{N \times N_{tot}}$ is parameterized as $\theta = CD$ with $C \in \mathbb{R}^{N \times K}$, $D \in \mathbb{R}^{K \times N_{tot}}$, and $K \ll N$. Then any parameterization of the form $\theta' = C(D + \Delta D)$ with $\Delta D P_{\mathcal{X}} = 0$ produces the same conditional probabilities on single-step predictions.*

*Proof.* We first show that $\theta' P_{\mathcal{X}} = \theta P_{\mathcal{X}}$:

$$\theta' P_{\mathcal{X}} - \theta P_{\mathcal{X}} = C(D + \Delta D) P_{\mathcal{X}} - CD P_{\mathcal{X}} = C\Delta D P_{\mathcal{X}} = C \cdot 0 = 0 \quad \text{(S22)}$$

Therefore $\theta' P_{\mathcal{X}} = \theta P_{\mathcal{X}}$. By Lemma S1, both parameterizations give the same conditional probabilities. The rank is preserved since $\text{rank}(C(D + \Delta D)) \leq \min(N, K) = K$. □

Corollary S2 shows that low-rank parameterizations do not resolve the fundamental ambiguity resulting from a finite observation domain $\mathcal{X}$: perturbations of the form $C(D + \Delta D)$ with $\Delta D P_{\mathcal{X}} = 0$ and $\Delta D \neq 0$ leave the dataset unchanged while preserving the network rank. On the other hand, unlike Theorem 1, the reverse logical direction is no longer true, *i.e.*, $C$ and $D$ cannot be uniquely identified as the rank factorization of $\theta$ is not unique. This is a general identifiability concern for the latent variables of low-rank RNNs (Dinc et al., 2025) and not necessarily specific to our formulation. In fact, Theorem 1 covers the joint parameters and suggests that $CD P_{\mathcal{X}}$ would be identifiable.

### S2.1.6 IDENTIFIABILITY UNDER PARTIAL OBSERVATIONS

A major challenge in training dRNNs is the presence of unobserved neurons, which is either omitted in practice (Das & Fiete, 2020) or used to argue caution against their use (Qian et al., 2024). On the other hand, data constrained models are used in practice to extract insights (Perich et al., 2021), predict individual neural activities under interventions (Walker et al., 2019), or guide causal intervention experiments (Liu et al., 2024; Vinograd et al., 2024).

To gain intuition, consider a set of observed $r(t)$ and unobserved $r_{\text{unobs}}(t)$, neurons. Then, for a given RNN, the dynamics of the observed neurons can be written as

$$\tau \dot{r}(t) = -r(t) + \tanh(W^{\text{rec}} r(t) + W^{\text{in}} u(t) + i(t) + \epsilon_{\text{in}}) + \epsilon_{\text{conv}}, \quad \text{(S23)}$$

where $i(t)$ denotes the influence of unobserved neural activities. This influence can be viewed as spatiotemporally correlated noise, but unlike random fluctuations, its structure is often highly aligned with the signal itself. This raises the question of whether parameter estimation remains possible at all, even under the guarantees of Theorem 1 and its empirical extensions to noisy conditions (Fig. 2).

Several works have been devoted solely to this problem (Qian et al., 2024; Brinkman et al., 2018), and our formulation here does not eliminate this issue whatsoever. We leave a theoretical discussion of this topic for future work. Instead, in Figure S5, we studied the effects of unobserved influences due to partial observations in dRNN training. We trained large-scale generator RNNs ($N = 10{,}000$) on the delayed cue discrimination task. From these, we observed only a subset of the neurons, whose activities were reconstructed with dRNNs. The top spectral components could not be recovered reliably under $\sim 1\%$ subsampling, but recovery improved rapidly once $\sim 10\%$ of neurons were observed. This level of recovery becomes realistic as imaging technologies are continuously improving (Kim & Schnitzer, 2022; Manley et al., 2024).

### S2.2 ESTIMATION OF dRNN PARAMETERS

Our second set of results presented in the main text, primarily in Figs. 2 and 3, concern the practical estimation of dRNN parameters under noisy dynamics and various training methods. Here, we first discuss the link between the Gram matrix $G_{\mathcal{X}}$ defined in Eq. 4 and the practical estimability of the parameters. Then, we show that multiplying the gradient updates with a regularized Gram matrix inverse would facilitate estimation of identifiable parameters. We show that the resulting Theorem 2 explains why FORCE learning potentially leads to non-identifiable parameters, the benefit of using a convex loss function.

### S2.2.1 PRACTICAL RELEVANCE OF THE GRAM MATRIX

Theorem 1 ties the identifiability of parameters to the projection matrix $P_{\mathcal{X}}$, where $X$ is an observation matrix defining the conditioning domain $\mathcal{X}$. Specifically, let $X = \sum_{r=1}^{R} \sqrt{T}\sigma_r u^{(r)} v^{(r)T}$ be the singular value decomposition of the observation matrix $X$ with $\text{rank}(X) = R$, $u^{(r)} \in \mathbb{R}^T$, and $v^{(r)} \in \mathbb{R}^{N_{\text{tot}}}$. By definition, both vectors have unit norm and satisfy orthogonality within each group, $e.g.$, $u^{(i)T} u^{(j)} = \delta_{ij}$ with $\delta_{ij}$ being Kronecker delta. Using this, the projection matrix can be found as:

$$P_{\mathcal{X}} = X^T (XX^T)^+ X,$$

$$= \left( \sum_{i=1}^{R} \sqrt{T}\sigma_i v^{(i)} u^{(i)T} \right) \left( \sum_{j=1}^{R} T\sigma_j^2 u^{(j)} u^{(j)T} \right)^{-1} \left( \sum_{k=1}^{R} \sqrt{T}\sigma_k u^{(k)} v^{(k)T} \right), \quad \text{(S24)}$$

$$= \sum_{i:\, \sigma_i \neq 0} v^{(i)} v^{(i)T}.$$

The Gram matrix defined in Eq. 4 can be found following:

$$G_{\mathcal{X}} = \frac{1}{T} X^T X = \sum_{i=1}^{R} \sigma_i^2 v^{(i)} v^{(i)T}. \quad \text{(S25)}$$

Comparing the two equations shows that the projection matrix can be practically estimated by considering the Gram matrix $G_{\mathcal{X}}$ and using the modes for which $\sigma_i \neq 0$, as $v^{(i)}$ constitute both the Gram eigenvectors and the orthonormal basis of the projection matrix. That being said, why do we want to use Gram matrix in practice?

The main motivation for using the Gram matrix, as opposed to computing $P_{\mathcal{X}} = X^T(XX^T)^+ X$ directly, is the fact that $X$ is almost always full-rank in a noisy scenario (Fig. 2) and thus the resulting $P_{\mathcal{X}} = I$ ends up being identity for many practical cases. Gram matrix, on the other hand, provides an intuitive explanation for when a mode is more likely to be constrained by many samples (putatively signal-constrained modes) vs which ones by few (putatively noise-dominated modes). We illustrate the intuition in Fig. 2**A**.

### S2.2.2 A BLUEPRINT FOR IDENTIFIABLE ESTIMATION OF DRNN PARAMETERS

We now prove Theorem 2, which states that the gradient updates when multiplied with a regularized Gram matrix leads to parameters that are confined within the top spectral components:

**Theorem** (Restatement of Theorem 2). *Consider a dRNN following Eq. 1 whose parameters $\theta \in \mathbb{R}^{N \times N_{\text{tot}}}$ is estimated by gradient descent of a differentiable loss $\mathcal{L}(\theta)$. Let $X = \sum_{r=1}^{R} \sqrt{T}\sigma_r u^{(r)} v^{(r)T}$ be the singular value decomposition of the observation matrix $X$ with $rank(X) = R$, $u^{(r)} \in \mathbb{R}^T$, and $v^{(r)} \in \mathbb{R}^{N_{\text{tot}}}$. Define $P_K = \sum_{r=1}^{K} v^{(r)} v^{(r)T}$, i.e., the projection matrix to the top $K$ spectral components of the Gram matrix. Assume that the gradient satisfies $[\nabla \mathcal{L}(\theta) v^{(r)}]_a = O(\sigma_r^n)$ for every entry $a = 1, \dots, N$, modes $r = 1, \dots, K$, any $\theta \in \Theta$, and some positive integer $n$. If $\theta^{(s)} P_K = \theta^{(s)}$ at iteration $s$ of the learning, then for any $\lambda$ satisfying $\lambda \gg \sigma_{K+1}^2$, and for any step size $\alpha > 0$, the update*

$$\theta^{(s+1)} = \theta^{(s)} - \alpha \nabla L(\theta) \left( \frac{1}{T} X^T X + \lambda I \right)^{-1}, \quad \text{(S26)}$$

*is a descent direction that satisfies $\theta^{(s+1)} P_K = \theta^{(s+1)} + O(\sigma_{K+1}^n / \lambda)$.*

*Proof.* We first prove that the updates will maintain the spectral confinement in the next parameter estimate $\theta^{(s+1)}$ and then show that the updates are still confined to the descent directions. As shown above, the gram matrix can be written as:

$$G_{\mathcal{X}} = \sum_{r=1}^{R} \sigma_r^2 v^{(r)} v^{(r)T}. \quad \text{(S27)}$$

Here, $\{v^{(r)}\}_{r=1}^{R}$ constitutes an orthonormal basis, where we set $\sigma_r = 0$ for $r > R$. Then we have

$$(G_{\mathcal{X}} + \lambda I)^{-1} = \sum_{r=1}^{N_{\text{tot}}} \frac{1}{\sigma_r^2 + \lambda} \, v^{(r)} v^{(r)T}. \tag{S28}$$

It is instructive to define $P_K = \sum_{r=1}^{R} v^{(r)} v^{(r)T}$ and $Q_K := I - P_K$. By assumptions of the theorem, we have $\theta^{(s)} P_K = \theta^{(s)}$, and $\theta^{(s)} Q_K = 0$. With this in mind, the update rule gives

$$\theta^{(s+1)} = \theta^{(s)} - \alpha \nabla \mathcal{L}(\theta^{(s)})(G_{\mathcal{X}} + \lambda I)^{-1}, \tag{S29}$$

Multiplying both sides on the right by $Q_K$ and using $\theta^{(s)} Q_K = 0$, we obtain

$$\theta^{(s+1)} Q_K = -\alpha \nabla \mathcal{L}(\theta^{(s)})(G_{\mathcal{X}} + \lambda I)^{-1} Q_K. \tag{S30}$$

Using the spectral decomposition of $(G_{\mathcal{X}} + \lambda I)^{-1}$, we have

$$(G_{\mathcal{X}} + \lambda I)^{-1} Q_K = \sum_{r=K+1}^{N_{\text{tot}}} \frac{1}{\sigma_r^2 + \lambda} v^{(r)} v^{(r)T}. \tag{S31}$$

Therefore,

$$\theta^{(s+1)} Q_K = -\alpha \sum_{r=K+1}^{N_{\text{tot}}} \frac{1}{\sigma_r^2 + \lambda} \nabla \mathcal{L}(\theta^{(s)}) v^{(r)} v^{(r)T}. \tag{S32}$$

By the assumptions of the theorem, each entry of the product satisfies

$$\left[ \nabla \mathcal{L}(\theta^{(s)}) v^{(r)} \right]_a = O(\sigma_r^n). \tag{S33}$$

Noting that $\sigma_r \geq \sigma_{r+1}$, this leads to the bound:

$$\nabla \mathcal{L}(\theta^{(s)}) v^{(r)} = O(\sigma_{K+1}^n) \quad \text{for all } r \geq K+1. \tag{S34}$$

Because $\lambda \gg \sigma_{K+1}^2$, we also have

$$\frac{1}{\sigma_r^2 + \lambda} = O\left(\frac{1}{\lambda}\right) \quad \text{for all } r \geq K+1. \tag{S35}$$

Combining these two bounds, we obtain

$$\theta^{(s+1)} Q_K = O\left(\frac{\sigma_{K+1}^n}{\lambda}\right), \tag{S36}$$

and therefore

$$\theta^{(s+1)} P_K = \theta^{(s+1)} - \theta^{(s+1)} Q_K = \theta^{(s+1)} + O\left(\frac{\sigma_{K+1}^n}{\lambda}\right), \tag{S37}$$

which establishes the spectral confinement of the update. We next show that the update is a descent direction. First, define

$$\Delta \theta := -\nabla \mathcal{L}(\theta^{(s)})(G_{\mathcal{X}} + \lambda I)^{-1}. \tag{S38}$$

Using the Frobenius inner product $\langle A, B \rangle = \text{Tr}(A^T B)$,

$$\langle \nabla \mathcal{L}(\theta^{(s)}), \Delta \theta \rangle = -\text{Tr}\left((\nabla \mathcal{L}(\theta^{(s)}))^T \nabla \mathcal{L}(\theta^{(s)})(G_{\mathcal{X}} + \lambda I)^{-1}\right). \tag{S39}$$

Since $G_{\mathcal{X}} + \lambda I$ is positive definite, so is its inverse. Hence

$$\text{Tr}\left((\nabla \mathcal{L}(\theta^{(s)}))^T \nabla \mathcal{L}(\theta^{(s)})(G_{\mathcal{X}} + \lambda I)^{-1}\right) = \|\nabla \mathcal{L}(\theta^{(s)})(G_{\mathcal{X}} + \lambda I)^{-1/2}\|_F^2 \geq 0, \tag{S40}$$

with equality only if $\nabla \mathcal{L}(\theta^{(s)}) = 0$. Therefore, the update direction is indeed a descent direction. $\square$

### S2.2.3 CONNECTIONS TO FORCE SOLVERS

FORCE solver is primarily developed for training chaotically initialized recurrent neural networks using recursive least-squares to train behaviorally relevant tasks, *i.e.*, the input-output maps, (Sussillo & Abbott, 2009). It was later applied to train dRNNs with slight modifications (Perich et al., 2021). Here, we briefly summarize the training method and derive how $\lambda$ parameter corresponds to the regularization parameter (Sussillo & Abbott, 2009).

We first formalize the recursive least-squares (RLS) update used in FORCE and assume $N_{\text{in}} = 0$ for simplicity. Consider the problem of learning a linear readout of the form

$$z(t) = w^T r(t), \tag{S41}$$

where $r(t)$ is the firing-rate vector, $w$ is the weight vector to be learned, and $f(t)$ is the target signal. The least-squares loss is

$$\mathcal{L} = \sum_t (z(t) - f(t))^2 = \sum_t (f(t) - w^T r(t))^2 = \|Aw - f\|_2^2, \tag{S42}$$

where the data matrix $A$ and the target vector $f$ are defined entrywise as $A_{ij} = r_j(i)$ and $f_i = f(i)$. The standard least-squares solution with an $\ell_2$ regularization term is

$$w_{\text{LS}} = (A^T A + \lambda I)^{-1} A^T f. \tag{S43}$$

To obtain recursive updates, we treat the data matrix $A$ and the vector $f$ as functions of time. Define

$$A(t) = [\, r^T(1) \; r^T(2) \; \ldots \; r^T(t) \,], \tag{S44}$$

so that $A(t)$ is a $t \times N$ matrix. Let

$$P^{-1}(t) = A^T(t)A(t) + \lambda I = \sum_{i=1}^t r(i)r^T(i) + \lambda I, \qquad s(t) = A^T(t)f = \sum_{i=1}^t r(i)f(i), \tag{S45}$$

such that the solution can be written as

$$w_{\text{RLS}}(t) = P(t)s(t). \tag{S46}$$

Both $s(t)$ and $P(t)$ admit recursive definitions:

$$s(t) = s(t-1) + f(t)\, r(t), \tag{S47a}$$

$$P(t) = \left(P^{-1}(t-1) + r(t)r^T(t)\right)^{-1}. \tag{S47b}$$

The update for $s(t)$ is immediate. The recursion for $P(t)$ follows from the Sherman-Morrison identity,

$$(A + uv^T)^{-1} = A^{-1} - \frac{A^{-1}uv^T A^{-1}}{1 + v^T A^{-1}u}. \tag{S48}$$

Since $\lambda I$ ensures invertibility at $t = 0$, the identity applies at every step. Using it, we obtain

$$P(t) = P(t-1) - \frac{P(t-1)\, r(t)r^T(t)\, P(t-1)}{1 + r^T(t)\, P(t-1)\, r(t)}, \tag{S49}$$

initialized with $P(0) = \lambda^{-1}I$. As $t$ increases, $P(t) = \left(\sum_i r(i)r^T(i) + \lambda I\right)^{-1}$ becomes the shrinkage-regularized inverse covariance estimator (Ledoit & Wolf, 2004). Multiplying the update for $P(t)$ by $r(t)$ yields

$$P(t)r(t) = \frac{P(t-1)r(t)}{1 + r^T(t)P(t-1)r(t)}. \tag{S50}$$

Using $s(t) = s(t-1) + f(t)r(t)$ and Eq. S50, the RLS weight update becomes

$$w_{\text{RLS}}(t) = P(t)s(t) \tag{S51a}$$

$$= w_{\text{RLS}}(t-1) - P(t)r(t)\left(r^T(t)\, w_{\text{RLS}}(t-1) - f(t)\right). \tag{S51b}$$

Defining the one-step prediction error

$$e_-(t) = r^T(t)\, w_{\text{RLS}}(t-1) - f(t), \tag{S52}$$

we can summarize the RLS updates as

$$P(t) = P(t-1) - \frac{P(t-1)r(t)r^T(t)P(t-1)}{1 + r^T(t)P(t-1)r(t)}, \tag{S53a}$$

$$w_{\text{RLS}}(t) = w_{\text{RLS}}(t-1) - e_-(t)\, P(t)r(t). \tag{S53b}$$

Following (Perich et al., 2021; Dinc et al., 2023), we used a modified version of this formulation to train the recurrent weights directly via FORCE. Specifically, defining $\hat{r}(t)$ as the model predictions and $r(t)$ as the ground truth neural activity to be matched by dRNN, the prediction error is now defined on the target neural activities:

$$e(t) = (\hat{r}(t) - r(t))/\alpha, \tag{S54}$$

the updates for the recurrent weights and inverse covariance become

$$P(t) = P(t-1) - \frac{P(t-1)\hat{r}(t-1)\hat{r}^T(t-1)P(t-1)}{1 + \hat{r}^T(t-1)P(t-1)\hat{r}(t-1)}, \tag{S55a}$$

$$W^{\text{rec}}(t) = W^{\text{rec}}(t-1) - e(t)(P(t)\hat{r}(t-1))^T. \tag{S55b}$$

Here $e(t)$ is an $N$-dimensional error vector, and $P(0) = \lambda^{-1}I$ sets the regularization level. The key modification here is that $W^{\text{rec}}$ is not initialized to zero, rather it is often initialized from a random distribution to enable spontaneous activity generation in the network. However, this initialization is precisely what leads to the non-identifiable component estimation as we show in Fig. 3 and cannot be mitigated by non-zero $\lambda$ values, *e.g.*, the same way that an offline least-squares would.

### S2.2.4 Convex optimization for parameter identifiability

Identifiability becomes a practically challenging problem when coupled with non-convex training objectives. Gradient-based training of dRNNs under such losses may converge to suboptimal solutions or local minima, further complicating parameter interpretability. Recent work has shown that convex reformulations of the learning problem can offer a powerful alternative. This formulation starts with designing a non-linear regression problem (Dinc et al., 2023). Specifically, given a trajectory of firing rates $r(0), r(1), \ldots, r(T)$ and corresponding inputs $u(0), u(1), \ldots, u(T)$, one constructs a regression problem by computing the discretized target:

$$d(t) = \frac{r(t+1) - (1-\alpha)r(t)}{\alpha} \approx \hat{d}(t) = \phi(\theta x(t) + \epsilon_{\text{in}}) + \epsilon_{\text{conv}}. \tag{S56}$$

This transforms the temporal dynamics problem into a standard supervised learning task: predict $\hat{d}(t) \approx d(t)$ from $x(t)$, which can be used to predict $\hat{r}(t+1) \approx r(t+1)$. Notably, minimizing the $\ell_2$ prediction error on $d(t)$ is equivalent to minimizing the $\ell_2$ prediction error on $r(t+1)$ following:

$$\mathcal{L}_2(\theta; r(t), \hat{r}(t)) = \frac{1}{T}\sum_{t=1}^{T}\sum_{i=1}^{N}(r_i(t+1) - \hat{r}_i(t+1))^2 = \frac{1}{T}\sum_{t=1}^{T}\sum_{i=1}^{N}(r_i(t+1) - (1-\alpha)r(t) - \alpha\hat{d}_i(t))^2,$$

$$= \frac{\alpha^2}{T}\sum_{t=1}^{T}\sum_{i=1}^{N}(d_i(t) - \hat{d}_i(t))^2 \propto \mathcal{L}_2(\theta; d(t), \hat{d}(t)). \tag{S57}$$

The key advance of CORNN is to replace the $\ell_2$ loss function on the prediction errors with a weighted cross-entropy, which is a convex loss function and has a global minimum:

$$\mathcal{L}_{\text{CORNN}}(\theta) = \frac{1}{T}\sum_{t=1}^{T}\sum_{i=1}^{n_{\text{rec}}} c_{t,i}\text{CE}\left(\frac{1+\hat{d}_{t,i}}{2}, \frac{1+d_{t,i}}{2}\right) + \sum_{i=1}^{N}\sum_{j=1}^{N_{\text{tot}}} \frac{\lambda}{2}\theta_{ij}^2, \tag{S58}$$

where $\text{CE}(a, b) = -b\log(a) - (1-b)\log(1-a)$ and $c_{t,i} = 1 - d_i^2(t)$. The loss function is minimized following a parameter update rule:

$$\theta^{(s+1)} = \theta^{(s)} + \left[E^{(s)T}X - \lambda\theta^{(s)}\right]\left(\frac{1}{T}X^TX + \lambda I\right)^{-1}. \tag{S59}$$

where the prediction error is defined as $E_i(t) = \frac{d_i(t) - \hat{d}_i(t)}{1 - d_i^2(t)}$. Since this is a convex optimizer, one can initialize $\theta^{(s)} = 0$ and run the minimization steps. In practice, (Dinc et al., 2023) uses an approximate initialization using a least-squraes solution:

$$\theta_{\text{initial}} = \frac{1}{T} \phi^{-1}(D)^T X \left[ \frac{1}{T} X^T X + \lambda I \right]^{-1}. \tag{S60}$$

In original publication, Dinc et al. (2023) has shown that CORNN is able to recover ground truth parameters $\theta^*$ from simulated neural trajectories, which arises from two key properties: First, CORNN is a convex method, guaranteeing convergence to a global optimum, or to a set of globally optimal parameters in the presence of redundancy. Second, in the low-error regime (close to global minima), minimizing the CORNN objective closely approximates least-squares estimation of the relation $\phi^{-1}(d) \sim \theta^* x$. However, the connections to the identifiability problem were not previously explored. Here, both the initialization in Eq. S60 and the update rules in Eq. S59 respect the blueprint rules setup in Theorem 2. Thus, as also shown in the main text, CORNN leads to parameter estimates confined within the identifiable subspace defined by the Gram matrix $G_{\mathcal{X}} = \frac{1}{T} X^T X$ and gated by a regularization parameter $\lambda$. Interestingly, however, the convex formulation enabled the regularized Gram matrix to coincide with the Hessian (near the solution), making the addition in Theorem 2 not a restrictive process but a helpful addition to the minimization.

## S2.3 PRESERVED DYNAMICS IN DRNNS

**Theorem** (Restatement of theorem 3). *Let $S_{\text{id}}(R) = \text{span}\{v_1, \ldots, v_R\}$ be the identifiable neural activity subspace spanned by the top $R$ spectral eigenvectors of the Gram matrix (or $S_{\text{id}}$ in short), and assume that for a noiseless, task-performing RNN with dynamics in Eq. 1, the activities satisfy $r[t] \in S_{\text{id}}(R)$ for all $t$. Let $\tilde{\theta}$ be identifiable with $\tilde{\theta} P_{\text{id}} = \tilde{\theta}$, where $P_{\text{id}}$ projects onto $S_{\text{id}}$. Then, any parameterization $\theta = \tilde{\theta} + \Delta\theta$ with $\Delta\theta P_{\text{id}} = 0$ but $\Delta\theta \neq 0$ yields identical dynamics $\dot{r}[t]$ for all $r[t] \in S_{\text{id}}$, but not necessarily when $r[t] \notin S_{\text{id}}$.*

*Proof.* The noiseless RNN dynamics are given by

$$\dot{r}(t) = -r(t) + \phi(\theta r(t)), \tag{S61}$$

with $\phi$ applied elementwise. Let $\tilde{\theta}$ be an identifiable parameterization such that $\tilde{\theta} P_{\text{id}} = \tilde{\theta}$, where $P_{\text{id}}$ projects onto $S_{\text{id}}$. Consider now $\theta = \tilde{\theta} + \Delta\theta$ with $\Delta\theta P_{\text{id}} = 0$ and $\Delta\theta \neq 0$. If $r(t) \in S_{\text{id}}$, then $r(t) = P_{\text{id}} r(t)$, and hence

$$\theta r(t) = \tilde{\theta} P_{\text{id}} r(t) + \Delta\theta P_{\text{id}} r(t) = \tilde{\theta} r(t). \tag{S62}$$

It follows that

$$\dot{r}(t) = -r(t) + \phi(\theta r(t)) = -r(t) + \phi(\tilde{\theta} r(t)), \tag{S63}$$

so the dynamics under $\theta$ and $\tilde{\theta}$ coincide for all $r(t) \in S_{\text{id}}$. If $r(t) \notin S_{\text{id}}$, then $(I - P_{\text{id}}) r(t) \neq 0$, and in general

$$\theta r(t) = \tilde{\theta} P_{\text{id}} r(t) + \Delta\theta (I - P_{\text{id}}) r(t) \neq \tilde{\theta} r(t), \tag{S64}$$

so the dynamics need not coincide. This proves the claim. $\square$

## S2.4 EXPERIMENT DETAILS FOR REPRODUCIBILITY

Here, we provide details of our experiments to ensure reproducibility. Additional details can be found in the code shared in the supplementary materials.

### S2.4.1 RECURRENT NEURAL NETWORKS

We use a biologically motivated and interpretable class of RNNs (Perich et al., 2021; Dinc et al., 2025). Since we focus on the discrete version of the RNNs, we utilize the Euler discretization described in Equation S67. In this section, we specify our implementation choices: how we initialize the weight matrices $W^{\text{rec}}$, $W^{\text{in}}$, and $W^{\text{out}}$, the distributions we sample for noise terms $\epsilon_{\text{in}}$ and $\epsilon_{\text{conv}}$, and other implementation details.

For reference, we construct RNN dynamics as follows:

$$\tau \dot{r}(t) = -r(t) + \phi(W^{\text{rec}} r(t) + W^{\text{in}} u(t) + \epsilon_{\text{in}}(t)) + \epsilon_{\text{conv}}(t) \tag{S65}$$

$$\hat{o}(t) = \psi(W^{\text{out}} r(t)) \tag{S66}$$

where $\tau \in \mathbb{R}$ represents the neuronal time constant, $r(t) \in \mathbb{R}^N$ the neural activities, $\dot{r}(t) \in \mathbb{R}^N$ their temporal derivatives, $u(t) \in \mathbb{R}^{N_{\text{in}}}$ the input signals, and $\hat{o}(t) \in \mathbb{R}^{N_{\text{out}}}$ the network outputs. In our experiments, we set $\phi(\cdot) = \tanh(\cdot)$ and $\psi(\cdot)$ as identity, $\tanh$, or sigmoid depending on the task, and use discretization parameter $\alpha$, which is calculated as the ratio of sampling interval $\Delta t$ to time constant $\tau$. Note that while the output weights $W^{\text{out}} \in \mathbb{R}^{N \times N_{\text{out}}}$ are used when training task-performing RNNs to generate ground-truth neural trajectories (as described in the following section), they are not involved in the parameter recovery process.

In our RNN implementation, we use Kaiming and Xavier initializations (He et al., 2015; Glorot & Bengio, 2010) with uniform and normal distributions for the weight parameters $W^{\text{in}}$, $W^{\text{rec}}$, and $W^{\text{out}}$. For the input noise $\epsilon_{\text{in}}$ and conversion noise $\epsilon_{\text{conv}}$, we implement Gaussian, Laplace, and Poisson distributions. However, we use the Poisson distribution predominantly in our experiments. During firing rate updates, since conversion noise $\epsilon_{\text{conv}}$ can cause values to exceed the bounds $[-1, 1]$, we clip the firing rates after each update: $r(t) = 1 - 10^{-6}$ when $r(t) \geq 1$ and $r(t) = -1 + 10^{-6}$ when $r(t) \leq -1$. Initial firing rates $r(t = 0)$ are sampled from Gaussian or uniform distributions depending on the experiment. To ensure reproducibility, we set fixed random seeds for Python's random, NumPy, and PyTorch random number generators.

In practice, neural activity data is discretized in time. Hence, we introduce discrete RNN models resulting from the Euler discretization of Eq. 1:

$$r[s+1] = (1 - \alpha) r[s] + \alpha \phi(z[s]) + \epsilon_{\text{conv}}, \tag{S67}$$

where we perform the discretization via $r[s] = r(s \cdot \Delta t)$, where we denote the discretized time scale as $\alpha = \Delta t / \tau$ and $s \in \mathbb{N}$ refers to the discretized time.

### S2.4.2 Obtaining ground truth neural trajectories

In our parameter recovery experiments, we use two different methodologies for generating ground truth neural trajectories. First, we use chaotic networks where we initialize parameters randomly and iterate without any supervision. Second, we train RNNs on one of three tasks (described in the following section: 3-bit flip-flop, delayed cue discrimination, delayed match-to-sample) and then examine parameter recovery in the presence of task-induced structure.

**Chaotic networks:** We use randomly connected recurrent neural networks to generate chaotic dynamics without any task-specific constraints. These networks consist of $N$ recurrently connected units with weights sampled from a Gaussian distribution $\mathcal{N}(\mu = 0, \sigma = 2/N)$, ensuring the network operates in a chaotic regime. The networks receive no external input during trajectory generation ($u(t) = 0$) and evolve according to their internal dynamics alone. Initial firing rates are sampled uniformly from $[-1, 1]$, and the system is iterated using the standard RNN update equation with $\tanh$ nonlinearity and step size $\alpha = 0.1$. These chaotic networks produce rich, complex temporal patterns that exhibit sensitive dependence on initial conditions while remaining bounded within the activation function's range. By studying parameter recovery from such unconstrained dynamics, we can assess identifiability in its most general form—without the structural biases imposed by task optimization.

**Trained networks:** In all training tasks, we train neural networks using input-output supervision, allowing networks to learn internal dynamics specific to each task. During initialization, we use Xavier initialization with uniform distribution as implemented in PyTorch.

Task-specific configurations vary as follows: for bias terms, we include learnable biases in the input and output linear layers of DCD and DMTS tasks but exclude biases in 3-bit flip-flop experiments. For output nonlinearities, we set $\theta(\cdot)$ as identity in 3-bit flip-flop, sigmoid in DMTS, and tanh in DCD tasks. Initial firing rates are sampled from $\mathcal{N}(0, \sqrt[4]{N})$ in 3-bit flip-flop, from $\tanh$ applied to $\mathcal{N}(0, 1)$ in DMTS, and from $\tanh$ applied to $\mathcal{N}(0, 0.1)$ in DCD. Since each task has different input-output requirements, the input dimension $N_{\text{in}}$ equals 3 in 3-bit flip-flop, 1 in DMTS, and 1 in DCD.

For all task training, we use Mean Squared Error (MSE) loss. The optimizers vary by task: we use Adam optimizer for 3-bit flip-flop and DMTS, while employing SGD for DCD. Additionally, we employ the ReduceLROnPlateau learning rate scheduler (with factor $0.5$ and patience equal to the number of epochs) specifically in 3-bit flip-flop experiments. Table 1 summarizes the key training hyperparameters for each task.

Table 1: Training hyperparameters to obtain generator networks for each task

| Hyperparameter | 3-bit flip-flop | DCD | DMTS |
|---|---|---|---|
| Network size ($N$) | $100, 500, 1000$ | $500$ | $1000$ |
| Input dimension ($N_{\text{in}}$) | $3$ | $1$ | $1$ |
| Output dimension ($N_{\text{out}}$) | $3$ | $1$ | $1$ |
| Number of epochs | $20000$ | $5000$ | $5000$ |
| Batch size | $50$ | $10$ | $10$ |
| Learning rate | $10^{-4}$ | $10^{-3}$ | $10^{-4}$ |
| Optimizer | Adam | SGD | Adam |
| LR scheduler | ReduceLROnPlateau | None | None |
| $\alpha$ (discretization) | $0.5$ | $0.5$ | $0.5$ |
| $\Delta t$ (ms) | $5 \times 10^{-3}$ | $5 \times 10^{-3}$ | $5 \times 10^{-3}$ |
| $\tau$ (ms) | $10 \times 10^{-3}$ | $10 \times 10^{-3}$ | $10 \times 10^{-3}$ |
| Input noise ($\epsilon_{\text{in}}$) | $0$ | $10^{-3}$ | $0$ |
| Output nonlinearity ($\theta$) | identity | tanh | sigmoid |
| Number of seeds | $20$ | $20$ | $20$ |

### S2.4.3 NOISE GENERATION PROCESSES

In most experiments, we sample noise independently at each timestep. However, for specific experiments examining the effects of realistic noise correlations, we implement spatially and temporally correlated noise.

**Standard (uncorrelated) noise:** By default, both input noise $\epsilon_{\text{in}}$ and conversion noise $\epsilon_{\text{conv}}$ are sampled independently at each timestep from the specified distributions (Gaussian, Laplace, or Poisson) with no spatial or temporal correlations.

**Correlated noise (experiment-specific):** In selected experiments, we generate spatially and temporally correlated input noise $\epsilon_{\text{in}}$ to model realistic neural recordings where nearby neurons and adjacent timepoints exhibit correlated fluctuations. First, we sample uncorrelated noise from $\mathcal{N}(0, \sigma)$ with dimensions $T \times N$, where $T$ is the number of timesteps and $N$ is the number of neurons. To introduce spatial and temporal correlations, we construct a 2D Gaussian kernel:

$$K(x, y) = \exp\left(-\frac{x^2}{2\sigma_T^2} - \frac{y^2}{2\sigma_N^2}\right) \tag{S68}$$

where $\sigma_T$ controls temporal correlation strength and $\sigma_N$ controls spatial (across-neuron) correlation strength. The kernel is normalized such that $\sum K(x, y) = 1$. We then convolve the uncorrelated noise with this kernel:

$$\epsilon_{\text{in}}^{\text{corr}} = K * \epsilon_{\text{in}}^{\text{uncorr}} \tag{S69}$$

where $*$ denotes 2D convolution with 'nearest' boundary conditions. Finally, we rescale the correlated noise to maintain the desired standard deviation $\sigma$:

$$\epsilon_{\text{in}} = \epsilon_{\text{in}}^{\text{corr}} \cdot \frac{\sigma}{\text{std}(\epsilon_{\text{in}}^{\text{corr}})} \tag{S70}$$

In these experiments, we use $\sigma_T = 3$ timesteps for temporal correlation and $\sigma_N = 50$ neurons for spatial correlation, with kernel size $30 \times 30$. Conversion noise $\epsilon_{\text{conv}}$ remains uncorrelated even in these experiments.

### S2.4.4 DESCRIPTION OF THE TASKS

Here, we clarify the implementation details and structure of three neuroscience-inspired tasks. First, we explain the 3-bit flip-flop task, where the network must maintain and selectively update multiple internal memory states. Second, we describe the delayed cue discrimination (DCD) task, where the network must classify an input signal and give an output after a delay period. Third, we explain our final task, delayed match-to-sample (DMTS), where the network must compare two sequential inputs and determine whether they match.

**3-bit flip-flop:** This task consists of three independent input channels where the values are $u_i(t) \in \{+1, 0, -1\}$ for $i \in \{1, 2, 3\}$. When a channel receives a positive or negative input signal, the network must output the corresponding value in that channel until a new non-zero signal arrives in the same channel. Importantly, inputs are presented randomly across trials, and after each presentation, the input signal returns to zero until the next random signal arrives. Therefore, the RNN must simultaneously maintain information from all three channels while producing the correct output signals.

Formally, the input dynamics are defined as:

$$u_i(t) = \begin{cases} \pm 1 & \text{if } B_i(t) \sim \text{Bernoulli}(0.05) = 1 \\ 0 & \text{otherwise} \end{cases} \tag{S71}$$

where $B_i(t)$ is a Bernoulli trial for channel $i$ at time $t$, and when $B_i(t) = 1$, the sign is chosen uniformly at random.

The network output follows a flip-flop dynamic where each channel starts at zero and latches to the most recent non-zero input:

$$o_i(t + 1) = \begin{cases} u_i(t) & \text{if } u_i(t) \neq 0 \\ o_i(t) & \text{otherwise} \end{cases} \quad \text{with } o_i(0) = 0 \tag{S72}$$

**Delayed cue discrimination (DCD):** The delayed cue discrimination task is more complex than 3-bit flip-flop as it requires both classification and delayed response. This task consists of three main intervals: input interval $T_{\text{in}}$, delay interval $T_{\text{delay}}$, and response interval $T_{\text{resp}}$. During the input interval, a cue of $\pm 1$ is presented in a single input channel. Throughout this period, the RNN must latch the information but should not produce any output, opposite to 3-bit flip-flop. After the input interval ends, the input becomes $0$ and the RNN must continue to maintain the output at $0$ during the delay interval. During the response interval, the RNN must produce a classification output based on the cue: if the cue was $+1$, the output should be $+1$; if the cue was $-1$, the output should be $-1$.

Formally, the input signal can be formalized as follows:

$$u(t) = \begin{cases} \pm 1 & \text{if } t \in T_{\text{in}} \\ 0 & \text{otherwise} \end{cases} \tag{S73}$$

The expected output is described as:

$$\hat{o}(t) = \begin{cases} +1 & \text{if } u_{\text{in}} = +1 \text{ and } t \in T_{\text{resp}} \\ -1 & \text{if } u_{\text{in}} = -1 \text{ and } t \in T_{\text{resp}} \\ 0 & \text{otherwise} \end{cases} \tag{S74}$$

where $u_{\text{in}}$ denotes the input value during $T_{\text{in}}$.

**Delayed match-to-sample (DMTS):** The third task is delayed match-to-sample. While sharing similarities with the delayed cue-discrimination task (delayed response, single input channel, and input classification), DMTS requires the network to compare two sequential inputs and respond accordingly. This task includes four distinct intervals: input interval $T_{\text{in}}$, delay interval $T_{\text{delay}}$, match interval $T_{\text{match}}$, and response interval $T_{\text{resp}}$. Similar to delayed cue-discrimination, the RNN should only produce the corresponding output during the response interval. Throughout the input interval, an input of $\pm 1$ is presented; afterward, during the delay period, the signal becomes $0$. After the delay period ends, another input of $\pm 1$ is presented during the matching interval. In the response interval, if the input and matching signals match, the RNN must give a positive response ($+1$); otherwise, the RNN should give a negative response ($-1$).

More formally, we can describe the input signal as follows:

$$u(t) = \begin{cases} \pm 1 & \text{if } t \in T_{\text{in}} \cup T_{\text{match}} \\ 0 & \text{otherwise} \end{cases} \tag{S75}$$

The ground truth output is described as:

$$\hat{o}(t) = \begin{cases} +1 & \text{if } u_{\text{in}} = u_{\text{match}} \text{ and } t \in T_{\text{resp}} \\ -1 & \text{if } u_{\text{in}} \neq u_{\text{match}} \text{ and } t \in T_{\text{resp}} \\ 0 & \text{otherwise} \end{cases} \tag{S76}$$

where $u_{\text{in}}$ denotes the input value during $T_{\text{in}}$ and $u_{\text{match}}$ denotes the input value during $T_{\text{match}}$.

### S2.4.5  FITTING RNN PARAMETERS TO REPRODUCE NEURAL TRAJECTORIES

After obtaining ground truth neural trajectories from chaotic or trained networks, we fit new RNN parameters to reproduce these observed dynamics. Rather than using computationally expensive backpropagation through time (BPTT), we employ a single-step prediction approach that frames parameter estimation as a feedforward regression problem.

**Single-step prediction framework:**

**Optimization methods:** We employ three primary approaches for parameter estimation:

*1. CORNN algorithm:* Our primary method uses the CORNN algorithm (Dinc et al., 2023), which employs an iterative update scheme with fixed point initialization computed via ridge regression on $z[s] = \text{arctanh}(d[s])$. We implement three loss variants: weighted loss (dividing prediction errors by $1 - d^2$ to account for tanh saturation), standard L2 loss, and derivative-weighted loss (multiplying by $1 - \hat{d}^2$). The algorithm includes outlier detection based on a threshold parameter (typically $0.2$ for trained networks, $1.0$ for chaotic networks). Convergence criteria: (1) $\sqrt{N} \cdot \sqrt{\text{mean}((\theta^{k+1} - \theta^k)^2)} < 10^{-5}$ after at least 10 iterations, or (2) maximum iterations reached (100-2000 depending on experiment complexity).

*2. FORCE learning:* In selected experiments with chaotic networks, we implement recursive least squares (RLS) FORCE learning (Sussillo & Abbott, 2009). FORCE updates parameters online using rank-one updates to the inverse covariance matrix, minimizing either current errors (pre-nonlinearity) or firing rate errors (post-nonlinearity). We use regularization parameters $\lambda = 100$ for recurrent weights and run the algorithm for up to 1000 iterations.

*3. Gradient-based optimization:* For comparison in selected experiments, we use PyTorch-based gradient descent with Adam optimizer (learning rate $10^{-3}$, up to $10^4$ iterations). Parameters are optionally initialized from the fixed point solution. This approach uses either Binary Cross-Entropy (BCE) loss or Mean Squared Error (MSE) loss, with L2 regularization applied through weight decay.

**Regularization:** The L2 regularization parameter $\lambda$ ranges from $10^{-23}$ to $10^{-1}$ depending on the experiment, with typical values around $10^{-15}$ to $10^{-13}$ for chaotic networks and $10^{-13}$ to $10^{-5}$ for trained networks. In CORNN, regularization is scaled by the number of data points $T$.

**Experimental variations:** We perform parameter recovery on both chaotic RNNs and trained networks performing the three tasks (3-bit flip-flop, DCD, and DMTS). For experiments with external inputs (trained task networks), we concatenate $u[s]$ with firing rates in $x[s]$; for chaotic networks without inputs, we set $u[s] = 0$.

## S3    FIGURE PARAMETERS

**Fig. 1:**    For **C-E**, ground truth weights for the generator RNNs (noiseless and chaotic) were drawn from $W_{ij}^{\text{rec}} \sim \mathcal{N}(0, \frac{g^2}{N})$ with $g = 2$. RNNs had no outside inputs. For **C-D**, we used $N = 1000$ neurons and a discretization parameter of $\alpha = 0.1$. dRNNs were trained using the single step prediction paradigm and a (convex) weighted logistic loss function ($\lambda = 10^{-15}$) (Dinc et al., 2023). For **E**, $N = 500$ and $\lambda$ varies, otherwise the same parameters are used.

**Fig. 2:**    Same as in Fig. 1**E**, but with $\lambda = 10^{-13}$, $T = 2500$. For **C** and **F**: solid lines show mean, shaded regions show s.e.m. over 20 randomly initialized RNNs. For **B, D, E**: each line or dot represents a single reconstruction experiment.

**Fig. 3:**    **A-C**, Same conditions as in Fig. 1**E**, with trajectory length fixed to $T = 2000$. We initialize the parameters of dRNNs trained with FORCE learning using a zero-mean Gaussian distribution with $g = 3$ and train for 100 epochs on the full samples. Reported $\lambda$ values are used directly for CORNN and multiplied by $10^5$ for FORCE. Lines: mean; shaded regions: s.e.m. over 20 randomly initialized RNNs. For **D-F**, generator RNN had $N = 500$ neurons, but only 25 were observed for dRNN training. Low- or full-rank dRNN parameters were trained by minimizing the $\ell_2$ loss on the single-step prediction errors via the ADAM optimizer. Dataset taken from (Qian et al., 2024, Figure 5b): $\alpha = 0.01$, $T = 30000$ training samples, with noise injected at each step ($\epsilon_{\text{in}} \sim \mathcal{N}(0, 1)$), and an extra observation noise ($\mathcal{N}(0, 1)$) was added to the measured neural activities without any feedback to the dynamics.

**Fig. 4**    For **B**, $\alpha = 0.05$ and $\lambda = 10^{-13}$. Zero-mean Gaussian input noise is added to two samples, s.d. values shown in panels. For **C-D**, dRNNs were trained using CORNN ($N = 500$, $\lambda = 10^{-6}$, $\epsilon_{\text{in}} \sim \mathcal{N}(0, 10^{-4})$, $\epsilon_{\text{conv}} \sim 0.1\text{Poisson}(10^{-2})$, $\alpha = 0.5$) with 500 base and varying numbers of extra samples. Outputs for the dRNNs are computed using the generator RNNs' output weights.

## S4  SUPPLEMENTARY FIGURES

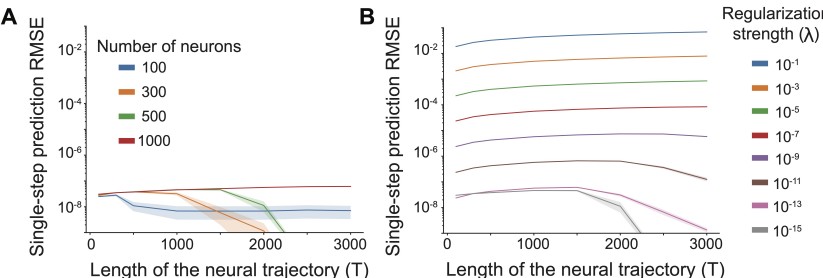

Figure S1: **Single-step prediction root-mean-squared errors corresponding to Figure 1C-E. A** Effect of the number of observed neurons on single-step prediction RMSE across trajectory lengths $T$ with negligible regularization ($\lambda = 10^{-15}$). **B** Effect of regularization strength $\lambda$ on single-step prediction RMSE across trajectory lengths $T$. These results complement Figure 1**C-E** by showing the direct single-step prediction errors for RNNs trained to reproduce neural trajectories sampled from chaotic RNNs. A reasonable tolerance level for the single-step prediction RMSE is $O(10^{-8})$, since squared error is minimized during training and machine precision is $\sim 10^{-16}$. Parameters: For **A**, same as in Fig. 1**C-D** but with varying $N$. For **B**, same as in Fig. 1**E** but with additional $\lambda$ values.

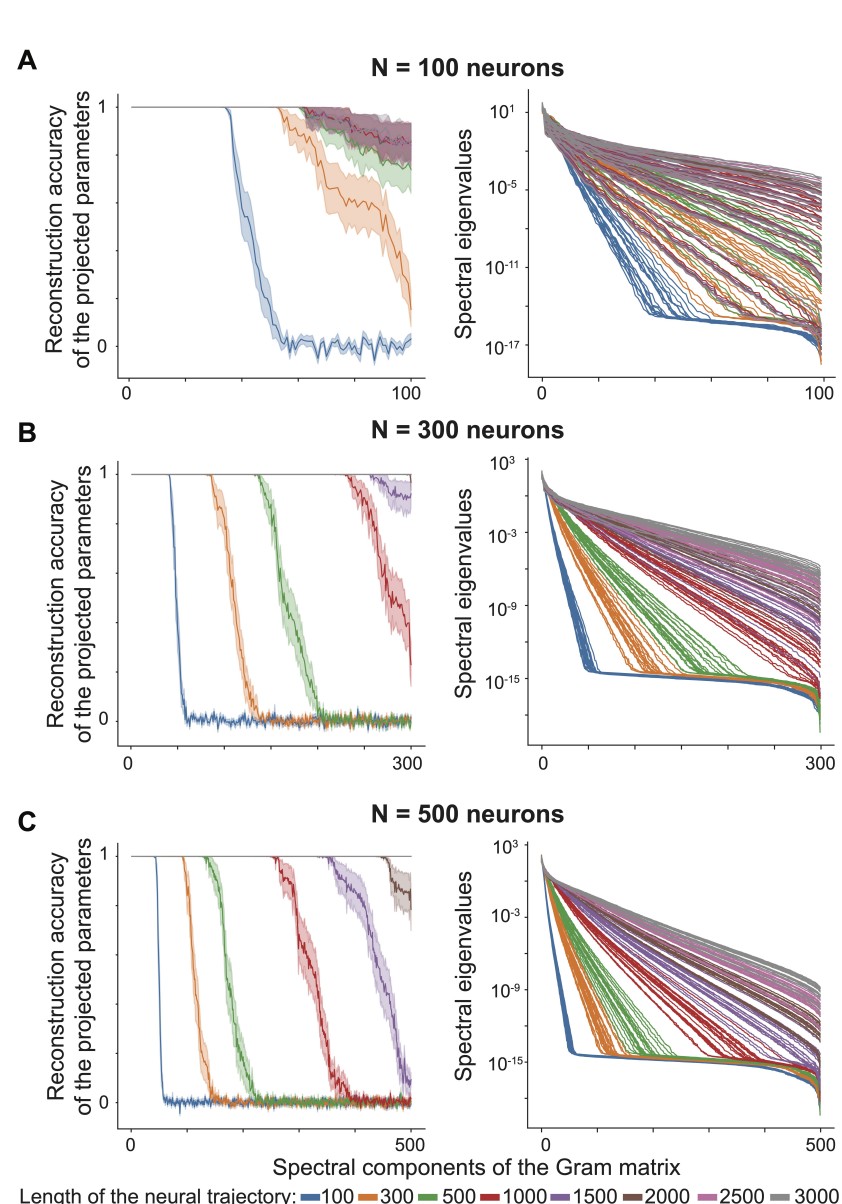

Figure S2: **Empirical verification of the neural uncertainty principle with networks of varying sizes.** We performed the same analysis as in Fig. 1**C-D** using the same experimental parameters, except RNNs had $N = 100$ (**A**), $N = 300$ (**B**), and $N = 500$ (**C**) neurons.

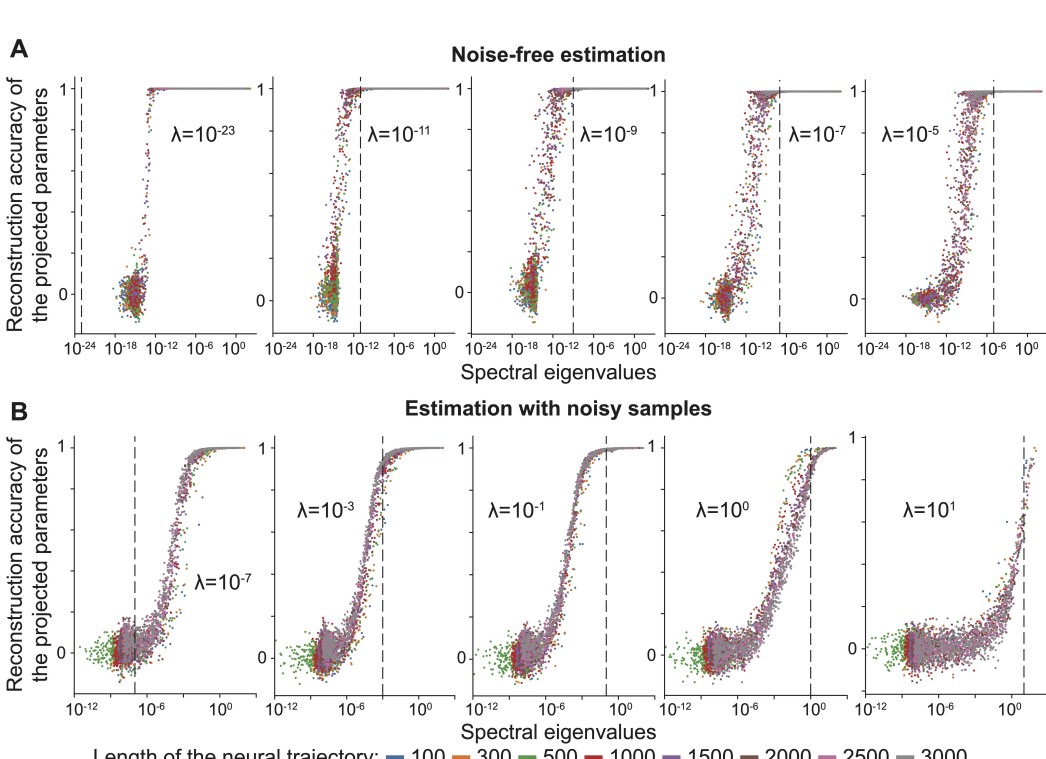

Figure S3: **Regularization levels control which spectral components are used for parameter estimation.** We performed the same analysis as in Fig. 1**E** for varying $\lambda$ values for noiseless (**A**) and noisy (**B**) evolutions of the RNN. For the noisy case, we picked $\epsilon_{\mathrm{in}} \sim \mathcal{N}(0, 10^{-6})$ and $\epsilon_{\mathrm{conv}} \sim \mathrm{Laplace}(10^{-3})$, in which $x$ in $\mathrm{Laplace}(x)$ refers to the scale parameter of the Laplace distribution.

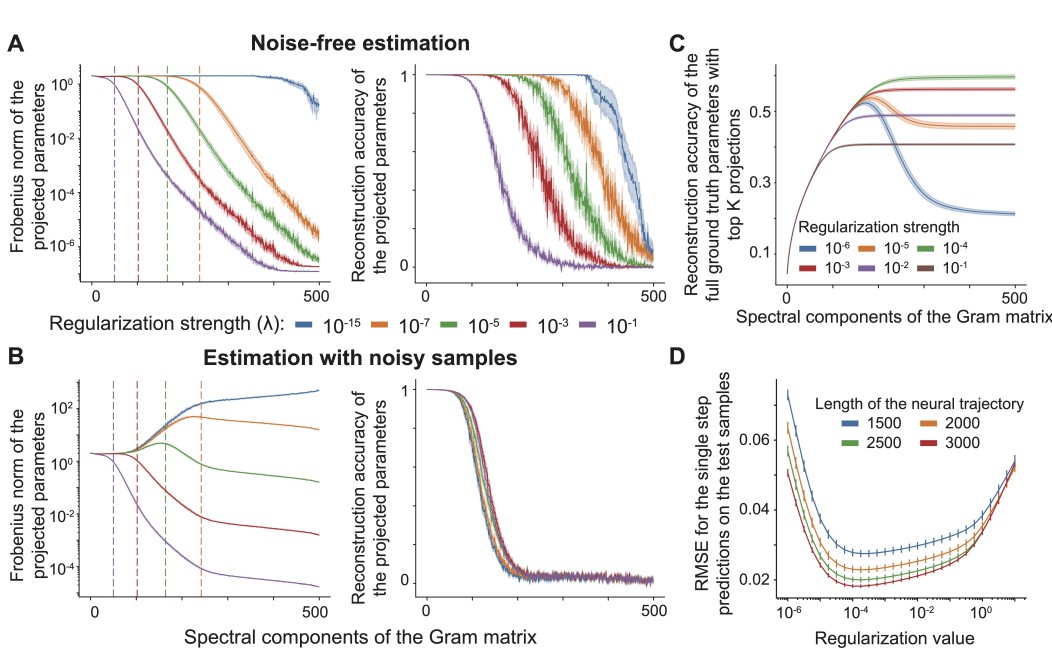

Figure S4: **Noise amplifies non-identifiable parameter estimates, weight regularization miti-
gates this inflation.** Similar to Fig. S3 and using the same experimental parameters, we examined
the Frobenius norm of parameter components projected onto the spectral components of the Gram
matrix (*left*) and the reconstruction accuracy measured as the correlations between ground truth and
predicted projections (*right*) across varying $\lambda$ values for noiseless (**A**) and noisy (**B**) RNN evolu-
tions. Without regularization, noise consistently amplified the estimation of non-identifiable param-
eter values, which were not constrained by the observed training data to begin with. **C** Accuracy
of reconstructing $\theta^*$ with $\hat{\theta}P_K$, in which $P_K$ is the projection matrix constructed using the top K
spectral components for noisy dynamics. If regularization level is not strong enough, additional
components contributing from the lower spectrum of the Gram matrix can decrease the reconstruc-
tion accuracy. **D** The regularization strength can be estimated using cross-validation on the contin-
uously divided train and validation datasets. Here, we used $T$ many training samples and sampled
an additional 100 samples, on which we computed the single-step prediction errors. The optimal
regularization strength ($\lambda = 10^{-4}$) that led to best reconstruction in panel **C** roughly corresponded
to the lowest prediction errors. Parameters: For **A-B**, similar to Fig. S3, we picked the noise values
as $\epsilon_{in} \sim \mathcal{N}(0, 10^{-6})$ and $\epsilon_{conv} \sim \text{Laplace}(10^{-3})$. For **C**, we used $N = 500$, $\epsilon_{in} \sim \mathcal{N}(0, 10^{-2})$ and
$\epsilon_{conv} \sim \mathcal{N}(0, 10^{-3})$, $T = 3000$, otherwise the same parameters as in Fig. 1**E**. For **D**, we used the
same parameters as in **C** but varied $T$. Solid lines: means. Error bars: s.e.m. over 20 runs.

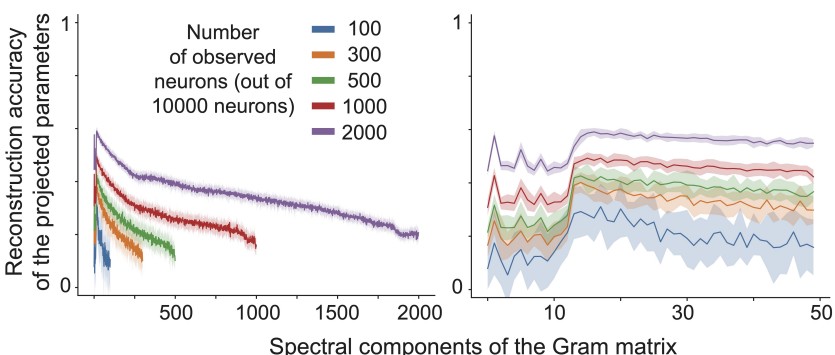

Figure S5: **Sparsely observed neurons exhibit unobserved influences that bias the top spectral components of the estimated Gram matrix.** A realistic assumption, supported by recent theories of neural computation (Dinc et al., 2025) and empirical work in task-trained low-rank RNNs (Valente et al., 2022; Beiran et al., 2021; Mastrogiuseppe & Ostojic, 2018; Schuessler et al., 2020; Dubreuil et al., 2022), is that the same latent variables underlie the dynamics of both observed and unobserved neurons. If this is the case, then the top spectral components of the Gram matrix are presumably dominated by the projections of latent computations and may become secluded and biased by the missing activity. In contrast, the lower spectral components may still be reconstructed. To test this hypothesis empirically, we trained large-scale generator RNNs with $N = 10,000$ neurons on the delayed cue discrimination tasks from Fig. S7 and performed inference using only partially observed neural populations. Estimation accuracies (Pearson's $r$) of the projected parameters between dRNNs and generator RNNs were plotted as a function of the spectral components of the Gram matrix computed from the training samples. The right panel corresponds to the close-up plot. As the number of observed neurons increased to about $10\%$ of the population, top $\sim 10 - 15$ spectral components, initially non-identifiable, became identifiable again. Parameters: Same as in Fig. S7, but with $N = 10000$, only independent noise injections, and about 300 trials, *i.e.*, comparable to a single imaging session (Ebrahimi et al., 2022).

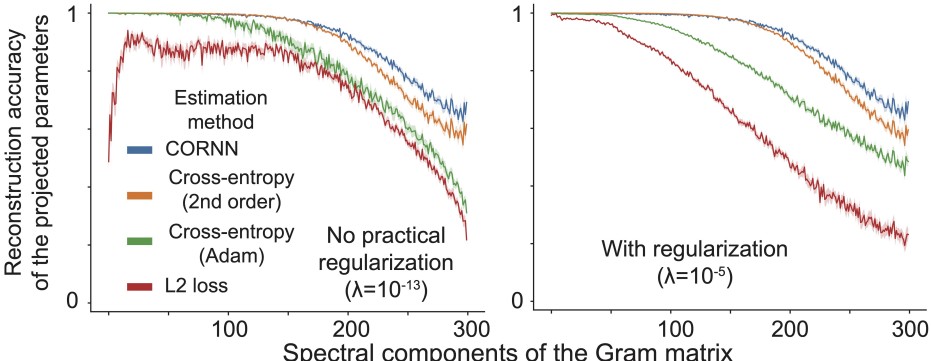

Figure S6: **Spectrally constrained estimators accurately recover top components despite radically different loss functions and optimization paths.** We reconstructed parameters of RNNs ($N = 300$) trained on a delayed match-to-sample (DMTS) task using four distinct optimization strategies, each minimizing single-step prediction errors. *Left:* Estimators with negligible regularization ($\lambda = 10^{-13}$). All except the first-order minimization of an $\ell_2$ loss led to the expected accuracy curves decreasing monotonically along the spectral components. $\ell_2$ loss is non-convex and notoriously difficult to converge to an appropriate local minimum in the absence of proper regularization. *Right:* With regularization ($\lambda = 10^{-5}$), all estimators produced reconstruction accuracies that decreased systematically along the spectral components of the Gram matrix computed from the training samples. Parameters: $T_{\text{in}} = 30$ ms, $T_{\text{delay}} = 80$ ms, $T_{\text{resp}} = 50$ ms, $\Delta t = 5$ ms, and $\alpha = 0.5$. RNNs were injected with random noise at every time step with $\epsilon_{\text{in}} \sim \mathcal{N}(0, 10^{-4})$ and $\epsilon_{\text{conv}} \sim 0.1 \, \text{Poisson}(10^{-3})$. Training samples included $B = 40$ trials, each with 38 time points, totaling $T = 1520$.

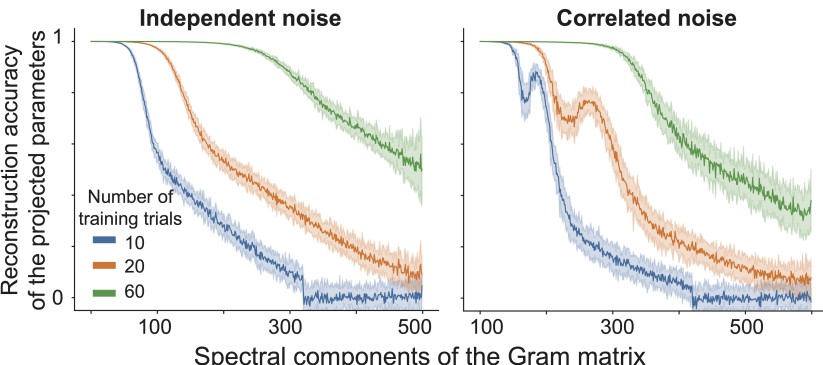

Figure S7: **Spatiotemporally correlated noise induces structure that complicates estimation, but can be mitigated with increased number of trials.** We trained generator RNNs ($N = 500$) on the delayed cue discrimination task with injected random, spatiotemporally independent (*left*) vs correlated (*right*) noise. Estimation accuracies (Pearson's $r$) of the projected parameters between dRNNs and generator RNNs were plotted as a function of the spectral components of the Gram matrix computed from the training samples. While training with spatiotemporally correlated noise required more trials to be accurate, it eventually converged to the structure predicted by Theorem 1. Parameters: $T_{\text{in}} = 30$ ms, $T_{\text{delay}} = 80$ ms, $T_{\text{resp}} = 50$ ms, $\Delta t = 10$ ms, and $\alpha = 0.5$. RNNs were injected with (random or correlated, see **Methods** for details) noise at every time step with $\epsilon_{\text{in}} \sim \mathcal{N}(0, 10^{-4})$ and $\epsilon_{\text{conv}} \sim 0.1\,\text{Poisson}(10^{-3})$. Each training trial contained 32 time points. dRNNs were trained ($\lambda = 10^{-7}$) with varying numbers of observed neurons and trials, as indicated in the figure legends.

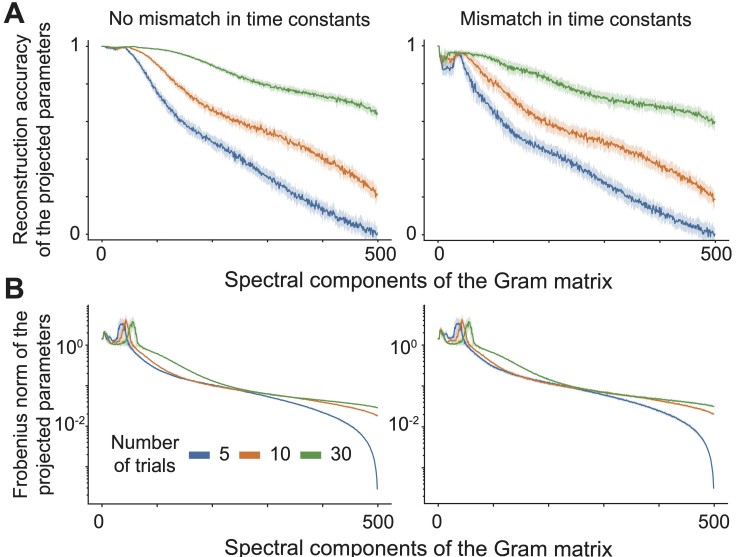

Figure S8: **Even with mismatched time constants, dRNN training tracks the spectrum of the Gram matrix.** We trained dRNNs to reproduce RNNs performing the 3-bit flip-flop tasks from Fig. 4. In the mismatch condition, time constants were sampled as $\alpha \sim \mathcal{N}(0.5, 0.05^2)$ instead of being fixed at $\alpha = 0.5$. **A** Reconstruction accuracy vs spectral components of the Gram matrix. **B** Parameter norms vs spectral components. Parameters: $N = 500$, $\lambda = 10^{-2}$, $\epsilon_{\text{in}} \sim \mathcal{N}(0, 10^{-6})$, $\epsilon_{\text{conv}} \sim 0.1\text{Poisson}(10^{-1})$. Each trial contained 100 time points.

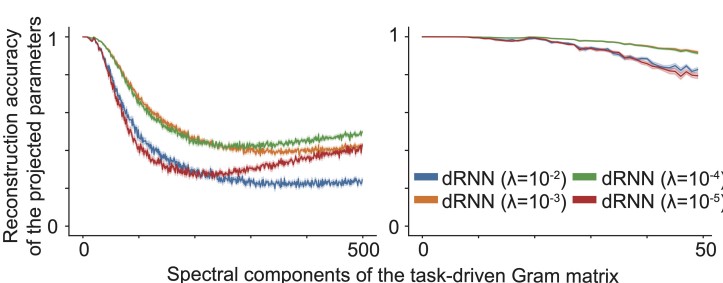

Figure S9: **dRNNs accurately reconstruct the top spectral components encoding task-driven dynamics.** Plots of reconstruction accuracies for the projected parameters obtained from experiments in Fig. 3**C-D** vs the spectral components of the task-driven Gram matrix computed as in Fig. 3**D**. Left plot shows the full spectrum, right plot shows the first 50 components. Top $10 - 20$ spectral components, which are shown to include task-relevant parameters in Fig. 3**D**, are accurately estimated from the ground truth RNNs.

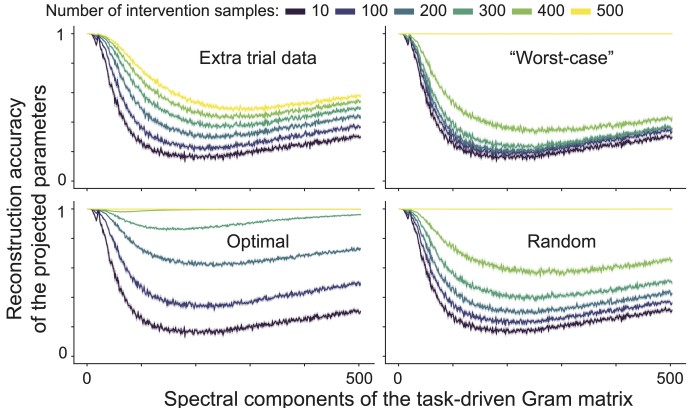

Figure S10: **Task-irrelevant parameters vary widely across dRNNs trained on task-driven neural activities.** We reanalyzed the parameter degeneracy in dRNNs trained in Figs. 3 and 4. In order to quantify the contributions of parameter subspaces to the task-relevant dynamics, we computed a generalized Gram matrix from 1000 trials of each network performing the 3-bit flip-flop task. The top spectral components correspond to the parameter subspaces driving the task-relevant dynamics (also see Fig. 3**D**). Using these spectral components, we evaluated reconstruction accuracies of the projected parameter dimensions as a function of the number of intervention samples used during reconstruction. Compared with Fig. 4**C** (where $\sim 100$ intervention samples led to high output accuracies for the optimal strategy; and $\sim 200$ for the random and "extra-trial-data" strategies), dRNNs that reliably solved the 3-bit flip-flop task exhibited substantial variability in spectral components beyond the top $10 - 20$. As expected from Theorem 1, all three Gram-matrix–based strategies (*i.e.*, all except "extra-trial-data") eventually achieved perfect reconstruction.

