# OpenReview forum: "A theory of parameter identifiability in data-constrained recurrent neural networks"
_ICLR.cc/2026/Conference — ICLR 2026 Conference Withdrawn Submission_

### Official Review · Reviewer_2GLr · 2025-10-27

**Soundness:** 3
**Presentation:** 3
**Contribution:** 2
**Rating:** 6
**Confidence:** 4

**Summary:**

This paper seeks to characterize when the parameters of an RNN can be reliably recovered from neural data, with applications to data-constrained modeling of neural dynamics. To do so, they establish conditions under which parameter subspaces are constrained by data, and when they might not be, supported by theoretical results and numerics. In particular, they argue that parameter combinations aligned with top components of the Gram matrix of observed neural activity are identifiable, whereas those associated with near-zero eigenmodes are not. They then show that FORCE learning fails to constrain parameters to identifiable subspaces, and then argue that non-identifiable parameters are optimally recovered with perturbations aligned with modes of the Gram matrix associated with small eigenvalues.

**Strengths:**

The writing is clear and organized. The numerics in this paper and appendix are quite extensive, touching on many of the more realistic conditions that neuroscientists may be concerned with (correlated noise, unobserved neurons, etc.). The observations made about FORCE necessarily learning non-identifiable parameters should be of interest to those working on or fitting data-constrained models.

**Weaknesses:**

Many of the claims feel overstated. Overall, it feels there is a large gap between what is actually proven in the theorems/propositions, and what is then concluded about identifiability of models from data as a result. For example, Theorem 1, along with most of the following results, characterize parameter identifiability in terms of a projection matrix derived from the concatenated observed neural activity/external inputs, yet this framing strictly holds only for the noiseless dynamics setting, with matched student/teacher architecture and no unobserved influences. The fact that the identifiability criteria of Theorem 1 is identical to that of noiseless LTI systems/linear RNNs speaks to how restrictive of a setting this is. Throughout this paper, stated conditions relating to parameter identifiability seem to be primarily of the necessary kind, whereas the abstract/title/discussion would lead one to think the contents also substantially cover realistic sufficient conditions for identifiability.

I acknowledge that extensions of theorems to more realistic settings are considered, but they still feel underexplored. For example, Proposition S1 regarding recovery under partial observations seems like merely a restatement of Theorem 1, with an addendum to restrict to the case where recovery of parameters associated with unobserved neurons can be safely ignored. I don't see how that result meaningfully characterizes how partial observations can corrupt identifiability.

I am still leaning towards an accept due to the extensive empirics, which I think would be useful in itself to the neuroscience community, but feel the paper would be much stronger if the theory spoke to the empirics better.

Finally, I find it strange that this paper develops a framework around parameter subspaces defined by the top eigenvectors of $X^\top X$, but makes no reference to PCA. Isn't this identical to (or at least closely related to) projecting parameters onto the top PCs of the activity+external inputs?

Other comments:
1. A minor point: throughout, $P \in \mathbb{R}^{N_X \times N_X}$ is stated to project to the column space of $X \in \mathbb{R}^{TM \times N_X}$, but by shape, this must be referring to the row space projector (projects to the subspace of $\mathbb{R}^{N_X}$ spanned by the rows of $X$).
2. The dynamics noise scale used in the empirics pertaining to estimation with noise---the more relevant/realistic case---feels absurdly small. For example, in Fig. S3, $\epsilon_{in} \sim \mathcal{N}(0,10^{-6})$. Since this noise is precisely the noise that corrupts the input $\theta x(t)$ to the nonlinearity, which is the relevant part of the dynamics invoked for Theorem 1, it would seem this would be the stress point that should be the most tested.

**Questions:**

1. Can the learning of non-identifiable parameter combinations by certain algorithms like FORCE be rectified by simply post-hoc projecting learned parameters to the identifiable subspace, as estimated by the empirical Gram matrix?

2. Very minor: In Fig. 2, presumably, the top modes of the Gram matrix change/wiggle as longer neural trajectories are observed, beyond just overall increases in rank. Consequently, the parameter subspaces evaluated need not be exactly comparable across curves in A, B, and C, no?

3. How is accuracy of parameter recovery defined, e.g. in Fig. 2 and Fig. 3? I'm assuming this is something like 1 - (relative error in frobenius norm), but this should be stated somewhere.

4. Regarding effects of partial observations on identifiability: the paragraph starting at line 1345 seems to state that recovery of parameters associated to top spectral components is poor for partially observed systems, even under very long recording sessions, yet the following paragraph seems to contradict this paragraph in spirit, and instead spins an optimistic tone. A very minor point, but I think this disconnect should be clarified, as I feel this discussion point will be of interest to the neuroscience community.

---

> ### Author Response · Authors · 2025-11-22
>
> We thank the referee for their careful reading and thoughtful feedback, and for their support of our work. We address each point below.
>
> **Claims feel overstated; gap between theorems and conclusions.** We appreciate this comment and revised the manuscript to more carefully align claims with what is proven. We now clearly distinguish necessary versus sufficient conditions throughout and adjusted the framing in the abstract, introduction, and discussion accordingly. This includes softening language around identifiability and explicitly stating the scope of the theoretical setting we study.
>
> **Theorem 1 holds only in a restrictive setting.** The referee is correct that the earlier presentation focused on the noiseless setting. In the revision, we extend Theorem 1 to include noise, clarify the assumptions under which the identifiability conditions apply, and add citations showing that this setting is widely used in systems neuroscience. The introduction now explains clearly why this simplified setting is both experimentally relevant and a logical starting point for identifiability analysis. We also highlight that although the criterion resembles that of LTI systems, this connection has not been made in the nonlinear dRNN literature, and establishing it enables our later results (Theorems 2 and 3).
>
> **Necessity vs sufficiency.** We now state explicitly which results provide sufficient conditions for identifiability in this simplified setting (revised Theorem 1), and which results characterize only necessary constraints. The abstract and discussion were revised accordingly so expectations match the scope of the analysis.
>
> **Partial observation results appear weak or overstated.** We agree that the previous version did not meaningfully illuminate how partial observations corrupt identifiability. We removed the theoretical corollary entirely, as it was too preliminary for the main paper. Instead, we now focus on the empirical insight (Fig. 3) showing that low-rank structure alone does not resolve identifiability issues under subsampling. A full theoretical treatment would require a separate manuscript, and we state this explicitly in the Limitations section.
>
> **Theory should speak to the empirics more clearly.** We understand where this point comes from. We added a new Figure 2 to show how Theorem 1 and the Gram-matrix spectrum connect to practical estimability under noisy conditions. The revised Results section highlights how Theorems 2 and 3 directly explain the empirical findings, including estimator behavior, regularization effects, and constraints on dynamical predictions. These connections are now emphasized more clearly in the structure of the section.
>
> **Relation to PCA.** We now note explicitly that the Gram-matrix eigenvectors coincide with PCA directions only when the concatenated data matrix is mean-centered. Since the matrix we use is not necessarily centered, the two decompositions are not always equivalent. This clarification was added directly after the definition of the Gram matrix as: “In the case of a mean-centered observation matrix X, its spectral decomposition corresponds to the principal component analysis regularly performed in neural datasets.”
>
> **Minor correction: row-space vs. column-space projector.** Good catch! We corrected this throughout the manuscript. The projector indeed maps onto the row space of the observation matrix.
>
> **Empirical tests with higher noise levels.** We thank the referee for flagging this. Figure 2 now includes results across a range of noise scales, and the conclusions remain unchanged.
>
> **Question 1.** The empirical Gram matrix is rarely singular under noise. In practice, letting regularization guide the estimator, as in CORNN, performs substantially better (Fig. 3). Post-hoc projection could mitigate part of the issue, but would not generally be optimal.
>
> **Question 2.** Correct. Comparisons across panels in Fig. 2 are based on estimation accuracy, not direct comparison of the subspaces themselves.
>
> **Question 3.** We clarified the definition in the caption of Figs. 1 and 2. We report both the errors in reconstructing the relative relationship between parameters (correlation) and the agreement between reconstructed norms, following standard metrics from prior work.
>
> **Question 4.** The referee is correct. We removed the long-term-recordings discussion, as it was not central to our main conclusions and requires more (empirical) evidence that we cannot do justice to. We retain the empirical result showing that with roughly ten percent of neurons observed, the top spectral components can still be accurately reconstructed.
>
> **Closing remarks.** We appreciate the referee’s careful assessment. The revisions substantially strengthened the alignment between theory and empirical results, clarified the scope, and improved the overall presentation. We look forward to further discussions and to the evaluation of our revised manuscript.

---

### Official Review · Reviewer_Txye · 2025-10-29

**Soundness:** 2
**Presentation:** 2
**Contribution:** 2
**Rating:** 2
**Confidence:** 4

**Summary:**

This work studies the identifiability of parameters of RNNs from observations of their inputs and dynamics, with motivation from the use of these models in neuroscience. Their main theorem states the fact that components of the parameters outside the subspace spanned by the observations - which is equivalent to the union of eigenspaces with non-zero eigenvalue of the empirical observability Gramian - are not identifiable.

**Strengths:**

As the authors clearly explain, this paper addresses an important topic. The main theoretical results of the paper are intuitive, though perhaps unsurprising. The range of experiments presented is interesting and mostly convincing.

**Weaknesses:**

The authors present their work in very sweeping terms, but as a reader their results didn't live up to the expectations set by the title and abstract. First, I do not find the main result (Thm. 1) to be particularly surprising; see my comment below about the study of LTI systems in control theory. Second, I am confused by the fact that the authors defer to appendix the extension of their main results to the partially-observed case. That seems central to the point they're trying to make, so though I guess it's a pretty deflationary result it's odd not to mention it clearly in the main text. I'd suggest bringing this into the main text, and moderating the tone of the paper overall. I also have a series of more specific concerns, questions, and suggestions, which I list under **Questions**.

**Questions:**

- There's a long line of work in control theory on problems of identifiability in system identification for linear dynamical systems, see for instance recent works by [Simchowitz et al. (2018)](https://proceedings.mlr.press/v75/simchowitz18a.html) or [Geadah et al. (2024)](https://ieeexplore.ieee.org/abstract/document/10886179) and references they cite. It would be useful to make contact with this literature, as my impression is that some analog of their Thm. 1 is folklore there. Also, the discussion of Gram matrices coincides with the classical observability Gramian; it'd be useful to make this connection.

- It'd be interesting to extend these results to nonlinearities that are not strictly monotone. For instance, the paper you cite by Biswas and Fitzgerald focuses on some of the degeneracies that arise from using a threshold-linear function.

- Thm. 2 is (up to the error term) a consequence of the fact that the parameter updates induced by gradient descent lie within the span of the observed covariates. This seems well-known to me, so I think you might consider citing a reference, or at least commenting on the conceptual content.

- Related to this aspect of gradient-descent-type algorithms, I don't think it's surprising that FORCE would retain initialization-dependence, and thus non-identifiable components. Is your aim here primarily to show that CORNN compares favorably to FORCE in this regard?

- I think Cor. 2 could be made more mathematically precise; when is this quadratic approximation reliable? In the appendix you don't consider the remainder term.

- The authors discuss regularization at length. However, thinking naively about neuroscience experiments, it's unclear to me how one should choose a regularization parameter in a principled way, as the data is presumably almost always nonstationary. Can you elaborate on this? For instance, you show in Figure S10 an example where you can get good estimation by choosing a good $\ell_2$ penalty; how do you choose this?

- The results on interventions are interesting, but could you at least speculate on the informativeness of experimentally-feasible interventions? I suppose those would be limited by partial observation.

- I'm curious about the generality of the claim in Figure S5 - this correspondence between variance and task-relevance should depend on the nature of the solution found by the RNN, right? For example, some solutions based on feedforward amplification - like Karel Svoboda's group has recently suggested are at play in ALM, in [Daie et al. (2023)](https://www.biorxiv.org/content/10.1101/2023.08.04.552026v2.full) - would seem to violate this.

- There is a long discussion in the appendix (starting around line 1360) about long-term recordings, but it's not clear to me whether that will necessarily resolve any of the challenges documented in the paper, because on those timescales there can clearly be substantial changes due to plasticity and other factors. I'm confused about why you'd cite Driscoll et al. (2017) here and not comment the fact that that paper shows substantial drift in responses over time, which seems contrary to the idea that such long recordings would help constrain an RNN model with fixed weights.

---

> ### Author Response · Authors · 2025-11-22
>
> We thank the referee for their careful reading of the manuscript and candid feedback. We address each comment below.
>
> **Theorem 1 is not surprising and connections to LTI.** We understand why the result may feel natural once presented in this framing. Our contribution is to formalize this connection for nonlinear recurrent models used in systems neuroscience, where this perspective has not been articulated. This framing also enables the later results. We clarified this motivation in the introduction and made the links to classical control theory and linear dynamical systems explicit both in the introduction and right after Theorem 1.
>
> **Partial observation results should be in the main text.** The theoretical extension was too preliminary and risked distracting from the main contribution, so we removed it. In the revised manuscript, we revised the abstract to make it clear we will focus on a simplified setting that is nevertheless relevant for systems neuroscience in order to set the expectations about this aspect right. In the main text, we clearly state the empirical insight in Fig. 3: low-rank structure alone does not resolve identifiability issues under partial observations. A more complete theoretical treatment of this topic deserves its own manuscript, which we make explicit in our limitation section.
>
> **Extensions to non-monotonic nonlinearities.** We agree that this is an interesting direction. The Biswas and Fitzgerald study focuses on steady-state neural activities, which complements rather than overlaps with our dynamical setting. Our priority here was to establish the core framework. These nonlinearities form a natural extension, and we plan to explore them in follow-up work.
>
> **Concerns about Theorem 2.** Theorem 2 is not simply the observation that gradient updates lie in the span of the covariates. In practice, the Gram matrix seldom has strict zero modes, and Theorem 2 shows how regularization guides which non-zero modes are used. The key contribution is that the result unifies existing estimators, including FORCE and CORNN, under a single framework. This was not stated in the literature. We revised the text to emphasize this point and added further connections to linear regression and Tikhonov regularization at the end of Section 2.4.
>
> **FORCE retaining non-identifiable components.** While this behavior may appear natural once framed in this way, it has not been recognized in the dRNN literature. Our contribution is to show why methods like FORCE retain non-identifiable components and to make the origin of this dependence explicit. We clarify this interpretation in the revised manuscript.
>
> **Corollary 2** We removed Corollary 2 entirely, which was not central to our results and its removal made the presentation clearer.
>
> **Choosing regularization in practice.** This is a common challenge in systems neuroscience. In Fig. S4C-D, we illustrate circularly shifting the data for cross-validation, which is widely used when trial structure is not available and remains effective even when the data are temporally correlated. In this case, the regularization parameter chosen through cross-validation (Fig. S4C) matches the value that yields the best parameter reconstruction (Fig. S4D). In datasets with trial structure, cross-validation can be done in the standard across-trial manner.
>
> **Interventions and feasibility.** Experimentally feasible interventions have indeed been demonstrated (for example, Vinograd et al., 2024). Theorem 3 clarifies which directions can provide new information: directions aligned with the dominant Gram-matrix components will not reveal additional dynamical constraints, whereas directions outside that subspace can. We added a short discussion at the third paragraph of the conclusion describing this connection.
>
> **Task-relevant parameters.** This correspondence depends on how strongly task-relevant dynamics dominate the recorded activity. In Fig. 4C–D we construct the task-driven Gram matrix using 1000 trials, where the dominant dynamics are driven by the task. These panels also illustrate that even when SVD reconstruction is high dimensional, the parameters subserving the dynamics can still be low dimensional. This provides a concrete visualization of Theorem 3.
>
> **Long-term recordings and plasticity changes.** We agree with the referee that long-term recordings introduce plasticity that complicates constraining fixed-weight models. We removed this discussion from the manuscript.
>
> **Conclusion.** We appreciate the referee's comments and suggestions. They helped us substantially improve the clarity and focus of the manuscript. Our revisions aim to make the framing more transparent while preserving the central message: connecting parameter identifiability, practical estimability, and preserved dynamics in this simplified but commonly studied setting of data-constrained RNNs. We look forward to the reviewer’s further questions and to their evaluation of the revised manuscript.

---

> > ### Comment · Reviewer_Txye · 2025-11-23
> >
> > Thanks for the revision! I've read through the replies to my comments and those of the other referees, and have some remaining questions which I'd like to discuss with the authors before I consider raising my score.
> >
> > First, regarding Thm. 2:
> >
> > - The authors frame this result as "a blueprint for identifiable training of dRNNs", but (like Thm. 1) it boils down to the fact that the identifiable and non-identifiable components of the parameters are separated into subspaces. This is what allows the (approximate) projector to be defined. Thus, echoing my original review and 2GLr's comments, this framing as a "blueprint" feels like an overstatement. As I mentioned, various versions of subspace-confinement for gradient-based algorithms are folklore in the optimization literature. For example, this is spelled out in detail for linear models [here](https://projecteuclid.org/journals/annals-of-statistics/volume-50/issue-2/Surprises-in-high-dimensional-ridgeless-least-squares-interpolation/10.1214/21-AOS2133.full). There, without the preconditioner and starting from the origin, the iterates are always confined to the subspace spanned by the Gram matrix. Again, this subspace-decomposition based notion of identifiability applies under restrictive assumptions. Can you clarify what you mean by a "blueprint"?
> >
> > - The result is asymptotic in $1/\lambda$, and doesn't control deviations from the identifiable subspace over multiple steps.
> >
> > - From a technical perspective the theorem appears incomplete, because it doesn't restrict the maximum step size to ensure stability.
> >
> > Second, I have a related question regarding Thm. 3. Like Thms. 1 and 2, this is a consequence of decomposing the state space into an "identifiable" subspace and its orthogonal complement. Thus, given that $\Delta \theta$ is non-zero with kernel given by the identifiable subspace, the result seems immediate. One might also want to consider the restrictions required on $\theta$ and on the nonlinearity $\phi$ such that the $r \mapsto \phi(\theta r)$ maps the identifiable subspace on to itself; this closure condition would allow predictivity. In general, the dynamics in the two subspaces will interact. Can you clarify the application of Thm. 3 to the specific applications you have in mind? This might help address the concerns regarding practical applicability raised by the other referees.
> >
> > Third, I'm still worried about regularization. Can you clarify what you mean when you say that you "circularly [shift] the data for cross-validation"? If you train and test on consecutive blocks of time, then the training and test sets are not independent, right? I'm concerned about this because [it has been a problem in neural data analysis before](https://www.biorxiv.org/content/10.1101/2025.03.09.642245v1).
> >
> > I also have some questions about the revised Figure 3, as it makes me wonder whether regularization is sufficient to enable accurate parameter estimation in the sense of eigenvalues. The authors re-analyzed an experiment from Figure 5 of Qian et al. (2024). Let me try to summarize my understanding; I appreciate the authors' clarifications if I've missed something here or in the work they reference:
> >
> > Looking at Figure 5 of Qian et al., those authors argued that dRNNs trained on partial observations overestimate the magnitude of a pair of complex-conjugate outlier eigenvalues in the ground-truth weights, which leads to a spurious stable limit cycle instead of a decaying spiral. The training there is done using some (seemingly ad hoc!) selection of inference procedures: MAP with a Gaussian prior of vanishing variance (so min-norm interpolation), FORCE, CORNN, and vanilla gradient descent. Based on my reading, Qian et al. didn't omit regularization entirely, but they didn't systematically test different choices. Is that correct?
> >
> > Here, the authors show that with stronger regularization you can get a decaying spiral, rather than a spurious limit cycle. However, even with regularization chosen based on cross-validated prediction error (Figure S4) the amount of shrinkage applied to the eigenvalues still doesn't seem sufficient to get the decay rate correct. Can you get an accurate eigenvalue estimate for some value of the $\ell_2$ penalty? How does the amplitude decay vary as a function of the $\ell_2$ penalty? This is related to the broader question of what you're seeking when you train a dRNN, as asked by bFPj.
> >
> > A few small suggestions which I missed before (so it's up to you whether you want to consider them):
> > - [Wagenmaker et al. (NeurIPS 2024)](https://arxiv.org/abs/2412.02529) and [Beiran and Litwin-Kumar (Nat. Neuro 2025)](https://www.nature.com/articles/s41593-025-02080-4) (on bioRxiv since May 2024, now published) seem relevant to the question of optimal perturbations and observations.
> >
> > - What happens if you have mismatched nonlinearities? This would be a step towards a more realistic setting.

---

> ### Author Response · Authors · 2025-11-24
>
> Thank you for the careful reading of our revision and for the detailed follow-up questions. We appreciate that we are now aligned on the broad utility of the work and are discussing narrower technical points rather than major disagreements. As a general comment, we are fully committed to translating ideas without overstating novelty and making necessary adjustments.
>
> **Theorem 2 and the term blueprint.** We explicitly cited classical Tikhonov results on linear regression in our revision (lines 256-269), which explains CORNN's behavior near its global minimum. Our intention with blueprint was not to imply algorithmic novelty, but rather as a simple recipe that explains why FORCE and CORNN behave as they do, and how one could design estimators constrained to identifiable subspaces. We are happy to change/omit the word blueprint.
>
> **Regarding asymptotic nature and step size:** Correct! In practice, both FORCE and CORNN converge in roughly ten iterations. Since the regularization parameter changes logarithmically and the remainder term grows at most linearly, the theorem captures the regime that matters in applications (Fig. 3). We will add a sentence to clarify these points.
>
> **On Theorem 3** This is a helpful point, and we will add a sentence after the theorem. This response (and the cited paper about LLMs) clarified for us where the main disconnect in interpretation of our work came from. We are not suggesting that one should make claims based solely on dRNNs. That is not how dRNNs are conceptualized in Perich et al. 2020, nor how we or system neuroscience labs we interact with think of them. dRNNs are theoretical tools to guide causal experiments.
>
> As noted by the referee, the main point of Theorem 3 is modest, but it is useful. If an attractor or fixed point lies in the identifiable subspace, its existence is preserved even when non-identifiable components are perturbed. If r* satisfies $\dot r*=0$, then $\dot r* =0$ when $Δ\theta \neq 0$. Interactions between subspaces may affect local dynamics near the attractor, but they do not determine whether the attractor exists. This is the coarse level at which dRNN analyses are meant to be interpreted in practice.
>
> **Regarding applications** Perich et al. (2020, 2021) and recent experimental work by Vinograd et al. (2024) and Liu et al. (2024) use data constrained models to propose attractors that are later tested experimentally. These perturbation experiments take years to prepare, and many predicted attractors do not replicate causally. Based on conversations at SFN, experimentalists were especially excited about Theorem 3 for this reason. It helps them prioritize predictions that are genuinely supported by identifiable structure.
>
> **Regularization and circular cross-validation** For Fig. S4C and S4D, we trained on the first thousand data points and tested on the last hundred. In practice one would perform multiple splits by circularly shifting the data before selecting the first XYZ points as the training set. This is standard when analyzing continuous neural recordings without trials, e.g. when decoding location from place cells. Circular shift keeps correlations between train and test sets confined to a short window at the boundary. One could also discard a small boundary region in the test set, though with this procedure, the cross validation in Fig. S4 selects a regularization value very close to the optimal one for reconstruction. Also, most experiments do have trial structure with several seconds between trials, which exceeds neural autocorrelation times. In those settings train test splits are straightforward.
>
> **Re-analysis of Qian et al. 2024** The referee is correct. In Qian et al., the chosen regularization values for both L2 regression and CORNN were effectively inactive once training converged. Up until $\lambda \sim 10^{-2}$, CORNN behaved similarly in our experiments (and Qian et al. used $10^{-6}$, practically no regularization). Regarding exact recovery of decay rates, we agree that such fine-grained accuracy is not expected, we discussed this line of thought more broadly in our previous response. We will clarify both points.
>
> **Closing remarks** We will add the suggested references and include a note on mismatched nonlinearities in the limitations. We thank the referee for the questions, which helped us sharpen several arguments. We will implement all edits described above and are happy to respond to any further points. We hope the referee will see the impact this work has for systems neuroscience, a field where theoretical progress is slower and results can appear less significant than they are (e.g., due to the experimental costs they save).
>
> Finally, we emphasize that we conceptualize our work as a high impact theoretical/computational application to the field of neuroscience (which is a major area of submission in ICLR), as opposed to an application of NeuroAI and/or alignment between brain and artificial intelligence.

---

> > ### Comment · Reviewer_Txye · 2025-11-24
> >
> > Thanks for your prompt reply!
> >
> > Respectfully, I am not sure what in my comment led you to "appreciate that we are now aligned on the broad utility of the work and are discussing narrower technical points rather than major disagreements". I am trying to get to a place where I can confidently assess the specific technical contributions, as a prerequisite to figuring out whether the manuscript as a whole rises above the sum of these parts. That's why I said I wanted to discuss these points "before I consider raising my score". I'm sorry that I did not make this clear, but I still have broader concerns regarding the philosophy (qua bFPj) and practical applicability of the results, especially relative to the framing. I want to be very up front about this so as to not incorrectly set expectations of where I stand regarding scores.
> >
> > Now, regarding your replies to my questions:
> >
> > - My objection to the word "blueprint" is that it sets up the result as something which can directly guide practice. I'm not convinced that this result meets that goal, again because of the underlying fact that it's a subspace separation result. I'm therefore on board with dropping it.
> >
> > - I don't follow what you mean by "the regularization term changes logarithmically"; this is as a function of what parameter? Also, shouldn't convergence in 10 iterations or so also only work for a particular range of step sizes such that those iterates don't blow up or oscillate?
> >
> > - I do not find your appeal to conversations at SfN convincing, principally because I don't see how one can confidently conclude identifiability when you have few constraints on the underlying biophysics (an extreme form of model mismatch). This is particularly concerning to me in the context of hypothalamus - as in the two works from the Anderson lab you cite - where extrasynaptic signalling and other intrinsically-slow mechanisms are at play.
> >
> > - Thanks for clarifying exactly what you mean by circular cross-validation. This is in fact what I'm concerned about. Yes, autocorrelation times of single neurons in most areas are fast relative to seconds-long trials (but see some of the Anderson lab's recordings - they have some long timescales there!), but activity can be subject to longer-timescale effects of internal and external state, among other things. I'm not even necessarily convinced by leaving out blocks of trials, as there can be similar effects that are, depending on the area, salient.
> >
> > Zooming out from these particular concerns, you mention again the fact that exact eigenvalue recovery is not expected. But what then is the philosophical objective of a parameter-level identifiability theory? This is where I tend to agree with bFPj's comment about coarse-grained features of the dynamics.

---

> ### Author Response · Authors · 2025-11-24
>
> Thank you for the clarifications. The concise nature of responses here can lead to misinterpretation, so we will directly focus on addressing the remaining points carefully below.
>
> **Point 1 (terminology).** We will follow up with your suggestions about the word blueprint. Since our intention was not to claim algorithmic novelty, we will remove the term and simply refer to the result by its theorem number.
>
> **Point 2 (regularization and convergence).** What we meant is that regularization is chosen over a logarithmic sweep, which is the standard procedure in FORCE and CORNN. Empirically, both methods converge in roughly ten iterations across the settings we studied. While points about how to train these models are technically correct, Fig. 3 also shows the nontrivial predictions of Theorem 2 are satisfied for both of these approaches.
>
> **Points 3 and 4 (model mismatch and cross-validation).** We fully agree that model mismatch, slow biophysical mechanisms, and long-timescale state fluctuations limit what any data-constrained model can guarantee. Our goal is not to argue that such effects disappear or are irrelevant. Rather, the identifiability results help determine which predictions are constrained by the recorded activity and which depend on unconstrained directions *within the model class being used.* We just think there is clear value in clarifying parameter identifiability in this setting.
>
> **Regarding circular cross-validation:** In continuous recordings without trial structure, clean i.i.d. splits are not possible, so circular shifts are used as a practical method. In simulations where ground truth is known, Fig. S4 shows that this approach selects regularization values that yield the best recovery. This is primarily a pragmatic choice, and data-constrained models were trained with it.
>
> **Broader objective.** In applied settings, dRNNs are used to generate hypotheses about broad dynamical structures that can then be tested experimentally. The aim of our identifiability results is not to recover underlying biophysics or exact values of eigenvalues, it is also not clear how much current technology would allow it precisely for the reasons laid out by the referee. We simply wish to determine which predicted structures of a fitted dRNN are genuinely supported by identifiable components of the data and which lie in non-identifiable directions. This distinction informs how much confidence one should place in specific model-derived hypotheses before investing substantial experimental effort. Our contribution is to make this boundary precise for the model class under consideration. The underlying biophysics or the precise eigenvalues would then be revealed with the experiments that should follow the use of dRNNs.

---

> ### Comment · Reviewer_Txye · 2025-11-24
>
> Sorry, one question here regarding your comment yesterday, before I address your follow-up reply. In response to my question about Theorem 3, you write "(and the cited paper about LLMs)". Which paper are you referring to?

---

> ### Author Response · Authors · 2025-11-24
>
> Jonathan Kao's work cited in the earlier response: "Illusions of Alignment Between Large Language Models..."

---

> ### Comment · Reviewer_Txye · 2025-11-24
>
> Sorry, thanks for the clarification. I was confused because I didn't intend to cite that in reference to Thm. 3, and also because it's not really a paper about LLMs per se. I referenced it just as an illustration of where cross-validation can lead to spurious predictivity.

---

> ### Author Response · Authors · 2025-11-24
>
> No problem. The paper did feel out of place (**Important edit: especially since it argues the circular shifts we used in Fig. S4 are the standard methodology and the paper primarily raises concerns on fully random splits, which would be of course not appropriate. e.g., see "we strongly recommend evaluating whether these other relationships are robust under contiguous splits."**), but was helpful in determining the broader concept referee wanted to think of our work in. However, this work would actually be closer to our line of thought (and is another application of dRNNs apart from the Andersonlab papers): https://www.nature.com/articles/s41586-024-08433-6
>
> Please allow me to summarize these long responses down to the primary point of our philosophy, both for AC and other referees, so that we can focus on specific contributions of our work:
>
> Our work does not really make a major claim about theoretical properties of RNNs, how well they represent underlying dynamics, or how useful they are on their own. We start with the philosophy that empirical validation with causal perturbations is the gold standard for any mechanistic claim. dRNNs are one way many labs use to guide these perturbations (works by Matthew Perich, Jonathan Pillow, Kanaka Rajan, Scott Linderman, David Anderson, Mark Schnitzer, Karl Deisseroth, and many others), as with large scale, these perturbations can no longer be planned adhoc with simple analyses. Our work provides a useful guidance for how to do this in a more principled way than to do it without concerns about parameter identifiability. The latter has been the case to date, which is why we believe our work is important. As referee has identified, there are many other interesting questions here and they should be studied. The fact that our work engaged the referees at this level, who made several expert suggestions on the more theoretical side of the literature and came up with several interesting questions for future work (a feat that a good research paper should induce in the readers), is good evidence for us that there is value in this thinking and sharing it with the broader community. We would greatly appreciate a focus on these points in the remaining discussions and thank you for your continuing suggestions.

---

> ### Comment · Reviewer_Txye · 2025-11-27
>
> Thank you for your reply, and for the clarifications. I would like to clarify that I only cited the Kao et al. paper to point out that cross-validation with correlated data has cropped up as a problem before in neural data analyses; it doesn't frame how I think of this work. I am intimately familiar with the Pagan et al. paper you reference.
>
> I think we're actually close to being on the same page regarding the broader philosophy of this work; I agree with you that figuring out principled ways to guide perturbations is an important potential application for dRNNs. Where I think we disagree is in the assessment (a) of the importance of technical contributions to this paper and (b) of the utility of its contributions for experiment. So far, my views on those two issues haven't really shifted, though I appreciate your clarifications and modification of the wording, which help a lot with (b) relative to the initial version. Since this work is presented as a theory paper, I still put a lot of weight on (a), which seems to be consistent with comments by other reviewers. With that, I'd be curious to see what the other reviewers think at this point, as they have yet to chime in. If a consensus is reached with a different balance of weightings than the one I have in mind, I'd still be open to raising my score.

---

> > ### Author Response · Authors · 2025-11-27
> >
> > We are glad there is now convergence and would love to hear from other referees as well. To respond to the final point raised, we note that leaving aside our conceptual contributions and the theorems, we have two significant technical contributions that are admittedly not derived from complex mathematics but has strong implications for the field:
> >
> > 1. FORCE learning, the most widely used training algorithm in these works, leads to spuriously estimated parameters.
> >
> > 2. Low-rank regularization does not mitigate over-parametrization issues, i.e., low-rank RNNs can have spurious parameters.
> >
> > Apart from the fact that no dRNN work to date considered the importance of Gram matrix (and thus none of the theorems are folklore for neuroscientists), point 1 was also never truly raised. Point 2 was raised, but on the case of a specific and intuitive example (Qian et al., 2024; where eigenvalue changes leads to distinct mechanisms), but here we show this is a more general phenomenon not specific to particular attractor structures.
> >
> > We look forward to responses from other referees.

---

### Official Review · Reviewer_U37S · 2025-10-31

**Soundness:** 3
**Presentation:** 2
**Contribution:** 2
**Rating:** 4
**Confidence:** 3

**Summary:**

The authors consider the problem of fitting RNNs to data, something often done in neuroscience. They characterise identifiability in the cleanest case: within model class, one-step on fully observed data, with monotonic nonlinearities, and noiseless; and find an intuitive result - the weights are only identifiable in the span of the `input datapoints' (meaning the span of the set of concatenations of previous timestep's activity vectors and current timestep's input vector). They show some theoretical and empirical results regarding which estimators will and won't recover these identifiable parameters, and present empirical results on how FORCE learning does not set non-identifiable weights to zero. Finally, they show how to design interventions to enlarge the set of identifiable weights, and that, if the activity stays within the identifiable span, that the dynamics will generalise.

**Strengths:**

- The main theorem, theorem 1, was intuitive and interesting
- The writing was clear, barring some AI-like verbosity
- The question was interesting, and well-framed
- The comparison to another common training method, FORCE, was cool.
- I liked the analysis of low rank networks S2.

**Weaknesses:**

Overall, the framing of the paper got me very excited, but I found that technical concerns led me to see the contributions as smaller than I initially thought. I will list them here, and the authors can likely correct me on some of my mistaken understandings.

- First, since the RNN is trained on one-step prediction, and assuming complete observability, the problem becomes a zero-layer feed-forward neural network, or a general linear model [linear regression then nonlinear link function]. Then the result is simply: if the nonlinearity is monotonic, identifiability becomes the same as for linear regression, and that is identifiable on the span of the input data. (A) Emphasising this simplicity seems good? (B) surely this is already well-known? Googling identifiability of generalised linear models provides many results. In this particular setting it may be new, but it seems very related to existing ideas?

- Then I had some concerns with the unobserved data case. Firstly, unless I'm confused, it is wrongly signposted in the text (it's at the end of appendix B.2, not C as advertised?) Then, for some reason phi become arbitrary rather than monotonic? Finally, and more importantly, the result's framing seemed weird. It showed that, even if you know the parameters relating to the unobserved data, you keep the nonidentifiability of the parameters outside the span of the observed data. So far so good. It did not show that, "even if $P=I$, RNN parameters may remain non-identifiable due to the hidden influence of unobserved neurons or redundancy introduced by non-monotonic activation functions". It just showed, exactly as in the original observed case, there exists a class of non-identifiable parameters living outside the span of the observed data. By assuming that the unobserved parameters were correctly observed you remove all the interesting parts of the problem? And you don't discuss the role of the nonlinearity at all? So why should I draw the conclusion you suggest from the theorem?

- Next, I was concerned by the claims about l2 regularisation. In the simplest case (no noise), finding the estimator that fits the data with minimal l2 norm will clearly select the weights that have no projection in the nullspace of the data matrix, solving the identifiability problem. Yet the authors claim that this is insufficient. They justify this claim by showing that FORCE learning recovers non-zero non-identifiable weights - certainly an interesting result on a shortcoming of FORCE learning. But I don't see the link - they claim that FORCE learning effectively performs l2 regularisation, but I don't see how. I read through the original algorithm and couldn't see the link to the fact it is minimising error + l2 regularisation, and in fact I view the authors' results as evidence in the opposite direction. If it were minimising such a loss, it would not have these non-identifiable components upon convergence!

- Theorem 2 seemed solid [did not check this proof], showing that if you start in the identifiable weights and effectively regularise the weights you will stay there. I was surprised about the fact corollary 1 is a local claim, about the loss near a minima where second order taylor expansions are relevant. This is a severe restriction for a general loss, and should be acknowledged as such (for example, perhaps name the corollary local identifiability in nonlinear regression). My take-away was still that l2 regularisation saves the day.

- I found the discussion of noise confusing. The model is introduced as noisy, but all the analytics are not about that setting. The only discussion of noise is in 4.3, where suddenly the data have noise added to them. I did not get why the important quantity is the span of the noiseless component - that's only true if both your real dynamics and your fitting are applied to the noiseless data, which is not the case? (Unless the added noise is just observation noise, not the input or conversion noise introduced in section 2) I agree that estimating the rank of data with noise is interesting, and likely relevant, but (a) surely this stuff is well studied? (b) the link to the rest of the work is very unclear to me.

Overall, despite thinking this was an interesting question, I felt like the more interesting parts weren't tackled. In the noiseless case with full observability it comes down to a very simple result the same as in linear regression about the span of the data. Further, the writing made this simplicity hard to see. A large part of the surprises in this problem seem to come from unobserved parts of the model/noise, and model mismatch; none of these were robustly tackled. Add to this additional confusions listed above, and I'm afraid I am currently leaning reject.

Further, stylistically, there was just a lot of material, that made it hard to digest. Appendix B was basically a continuation of the paper (8 pages!), including B.3, one of my favourite bits. Some more digesting by the authors, and much punchier writing (it's very verbose), will likely help the presentation. But this is vague advice, so not something I can reasonably request changed in a rebuttal.

**Questions:**

Clear from the above I think.

---

> ### Author Response · Authors · 2025-11-22
>
> We appreciate the candid feedback provided by the referee. Your comments helped us clarify the framing of our results and improve the presentation. Below we respond to each concern in turn.
>
> **Concern 1.** We agree with the reviewer’s framing. The introduction now explicitly cites prior work on identifiability in linear dynamical systems, beginning with the sentence "In contrast, linear dynamical systems enjoy remarkably clean identifiability properties…" We also added a clarifying sentence at the end of Theorem 1 to make this connection clear.
>
> **Concern 2.** We appreciate this feedback. This part of the manuscript is an extension of the main framework rather than a central theoretical result. Full identifiability under unobserved influences remains an open challenge in the field, including in recent work focused specifically on this problem (Qian et al., 2024). Our main contributions here are empirical.
>
> 1. We show that low-rank regularization alone does not mitigate these issues, which contrasts with interpretations from earlier low-rank RNN studies (Valente et al., 2022).
>
> 2. We show that weight regularization guided by the Gram matrix remains important even with unobserved influences.
>
> To make these contributions clearer, we moved the empirical results into the main text (Fig. 3) and removed the corollary that was not central to the main message.
>
> **Concern 3.** The theoretical properties of a regularized loss and the practical optimization of that loss are distinct concepts. FORCE implements an L2-regularized objective through an online algorithm, but does not necessarily find the optimal value of the loss function. To make this link explicit, we provide a derivation of the FORCE update rule in Appendix S2.2.3 and highlight the connection in the main text with the statement "lambda corresponds to a weight regularization in the limit of large samples \citep{mahadi2022recursive}."
>
> **Concern 4.** The locality in the corollary referred only to minimizing single-step prediction errors. It stated that when the estimator reaches a global extremum, lambda functions effectively as a regularization parameter. We removed this discussion because it shifted attention away from the main conclusion that Theorem 2 unifies FORCE and CORNN under a single estimator framework. The central message remains that L2 regularization plays an important role, which we believe is a simple but important message since  theoretical studies sometimes do overlook this when discussing dRNN limitations.
>
> **Concern 5.** We agree that there was a missing intermediate step. The original text moved too quickly from model to observation noise. Since this level of detail is not essential for our main argument, we revised the manuscript and added a more intuitive explanation of how the Gram matrix quantifies sample size effects. This revision appears in Fig. 2 and the corresponding section.
>
>
> **Overall concerns.** To address this broader concern, we extended Theorem 1 to include noise. Some connections between Theorems 1, 2, and 3 were not sufficiently emphasized in the previous version of the manuscript, which likely contributed to the confusion. While several directions raised by the reviewer are indeed important, they fall outside the scope of the present work. Our focus is on establishing the connections between parameter identifiability, practical estimability, and preserved dynamics. These links were not known in the literature, and we view them as an important foundation for future work on model mismatch and partial observations. We make this explicit in the limitations section with the statement:
>
> "Second, while we studied low-rank RNNs and influences of unobserved neurons, these analyses were intended primarily to complement our central results on dRNNs. A more complete theory in these domains remains to be developed and represents a natural and important extension of our work."
>
> **Final stylistic concern.** We implemented all stylistic suggestions through targeted rewrites and new experiments where needed. We thank the referee for these constructive comments and look forward to their reevaluation of the manuscript.

---

### Official Review · Reviewer_bFPj · 2025-11-01

**Soundness:** 2
**Presentation:** 2
**Contribution:** 2
**Rating:** 2
**Confidence:** 2

**Summary:**

The manuscript presents a mostly theoretical treatment of the issue of parameters identifiability in the fits of recurrent neural networks (RNNs) to neural data. RNNs are increasingly fitted to experimentally measured neural data as a way to extract the relevant dynamics and potentially gain insights into the underlying mechanisms. Yet it is not fully understood to what extent the parameters of RNN are in principle identifiable based on finite, noisy data. The authors present several theoretical insights into this question and propose experimental approaches that might mitigate issues of identifiability.

**Strengths:**

The premise of the paper is clearly outlined. Several interesting connections to past work (even outside neuroscience) are presented and discussed.

The focus on relatively simple, tractable settings allows the authors to gain precise insights into which parameters of their models are identifiable and which are not (in the form a various theorems and corresponding proofs).

The work appears technically sound.

**Weaknesses:**

While the paper provides some interesting insights into which parameters of data constrained RNNs are identifiable or not, the practical relevance of these insights is not clear. One prominent application of data-constrained RNNs in neuroscience is to obtain smooth/denoised estimates of low-dimensional, latent dynamics from high-dimensional, noisy observations. For such applications, presumably it does not matter if multiple RNN weight matrices exist that can explain the dynamics equally well. Likewise, many mechanistic insights into the function of the fitted RNNs (like the topology of fixed points) are probably possible even if the RNN weight matrix cannot be identified uniquely. In fact, it is known that the same type of dynamics can be implemented even by different classes of RNNs (work on “Universality” by Sussillo and colleagues, 2019).

The authors should also clarify the connection of their work to past studies that have found a close relationship between dimensions of the weight matrix and dimensions of the dynamics. This relationship has been described in detail in low-rank networks (work by Ostojic et al) and even nominally high-rank RNNs have been found to be functionally low-rank (Krause et al, 2022), whereby a only a low-d subspace of the weight matrix is sufficient to explain the corresponding low-d dynamics. These lines of work seem closely related to those in this manuscript, which also finds that the subspace of the identifiable weights is closely linked to the subspace explored by the dynamics.

I found some of the sections of the paper are rather dense and difficult to read. It would help if the authors could at times provide more intuitions about the insights gained from their theorems.

**Questions:**

What types of insight are affected by the non-identifiability presented by the authors? If the goal of fitting an RNN to data is to infer latent dynamics, or generate hypotheses about the underlying topology of the dynamics, does it matter that the RNN parameters are not fully identifiable?

The causal interventions proposed by the authors to alleviate non-identifiability seem to be focused on characterizing the components of the weight matrix that are not identifiable, as they are not sufficiently constrained by the measured data. But why would it even be desirable to constrain dynamics that are not explored in the “natural” operation of a neural circuit?

---

> ### Author Response · Authors · 2025-11-22
>
> We appreciate referee's careful reading of our manuscript and constructive comments, which motivated us to more clearly articulate the importance of parameter identifiability and to refine several conceptual points. Below we address the concerns point by point.
>
> **Importance of parameter identifiability** We illustrate the importance of parameter identifiability with two representative cases:
>
> 1. Inter-area communication (Perich et al., 2021). A flagship application of dRNNs is inferring communication between brain regions. However, if parameters are not identifiable, then different parameter settings (equally consistent with the data) may imply contradictory or even nonexistent interactions. As we noted in the pre-rebuttal, this can lead to spurious conclusions about cross-area communication.
>
> 2.  Predictive reliability of RNN dynamics (Theorem 3). Theorem 3 shows that only the identifiable components of the parameters determine dynamical predictions in the well-supported activity subspace. Predictions outside this subspace depend on non-identifiable directions and therefore cannot be trusted. We emphasize this point in the revised conclusion (third paragraph).
>
> **Latent variables and low-rank RNNs?** The reviewer’s statements regarding latent-variable models and dRNNs do not fully reflect how these models were applied to empirical datasets so far (please refer to new introduction). Most dRNN studies, including Perich et al. (2021), do not study latent dynamics; the goal is to recover neural dynamics. Even when latent variables do arise, as in low-rank RNNs, identifiability remains essential. Different weight matrices can generate the same latent structure yet diverge in their predicted neural activities. Theorem 3 formalizes when such divergences matter: if two parameter sets agree on the identifiable subspace, any disagreement must lie in non-identifiable directions. Moreover, low-rank RNNs can “hallucinate’’ attractors/predictions not supported by data (Corollary S2), as shown in Fig. 3. We therefore view our framework as complementary to latent-variable analyses, since it identifies which dynamical predictions are genuinely data-constrained.
>
> **Universality in RNN solutions** We agree that universality (i.e., multiple RNNs implementing similar input-output functions) is an important topic in task-trained models. However, this concept is only tangentially relevant to our setting. Our focus is not on algorithmic universality but on single cell level dynamical predictions. Even if two RNNs implement the same input-output map, Theorem 3 shows that they may differ in predicted single-neuron trajectories unless their identifiable components match. These neuron-level predictions are important for empirical applications such as confirming the existence of attractors with perturbation experiments. To clarify this, we expanded the explanation of Theorem 3 in the main text and introduced Figs. 4A-B, which illustrate how preserved-dynamics subspaces arise and why they matter.
>
>
>
> **Density of results** To address this concern, we have removed several nice-to-have results from the manuscript and focused primarily on the main contributions. We have also reduced redundancy in writing, and performed several revisions to improve readability.
>
> **Question 1.** Please refer to Theorem 3, Corollary S2, and Fig. 3. Even if the goal of fitting a dRNN was to infer latent dynamics, parameter identifiability plays an important role as low-rank RNNs can still suffer from non-identifiability issues. Moreover, as soon as the goal becomes controlling neural populations or testing the existence of these attractors, the question of which neurons to interfere with significantly benefits from a consideration of parameter identifiability and associated predictions of preserved dynamics.
>
> **Question 2.** Good question! Identifying these weakly constrained directions is helpful for several reasons. First, it prevents over-interpreting dRNN predictions that are not supported by the observed data. Second, many experiments require probing dynamics in subspaces that are poorly sampled. For example, if a dRNN predicts a line attractor near the edge of the Gram spectrum, it would be unwise to commit to long single-cell perturbation experiments without first ensuring that this prediction is actually constrained by data. A practical approach is to excite groups of neurons to expand the identifiable subspace, refine predictions, and iterate. Finally, resolving these additional directions makes it possible to study generalization and abstraction related questions, i.e., to generate more reliable predictions for novel stimuli or tasks.
>
> Overall, we thank the referee for their time and thoughtful feedback. It helped us clarify our framing and make explicit how our results differ from the low-rank RNN literature. We look forward to any further questions and to the referee’s reevaluation of our work.

---

### Author Response · Authors · 2025-11-13
**A pre-rebuttal submission (part 1 of 2)**

Dear AC and Referees,

We would like to thank you for your efforts in reviewing our manuscript. We find the reviews to be constructive. Below, we briefly highlight the referee consensus, provide a summary of all major concerns, and then explain how we intend to address them in our revisions. We then conclude with our high level response to the reviews, along with a proposed revised title and abstract, which seems to be the major point of confusion.

This is not intended to address all concerns, but to show that we understand which direction of change is requested. **For now, we only ask you to confirm whether you would like to see a rebuttal based on these revisions, and be open to revising your recommendation accordingly.**

## Referee consensus

Multiple referees noted that our work has a clear theoretical motivation and includes extensive evaluations. The mathematical results were found to be intuitive, though at times some connections to existing work in control theory were missing. Referees raised concerns, however, about specific claims (particularly regarding L2 regularization in FORCE learning) and noted that the presentation had major flaws. There was consensus among referees that the work was motivated in a way that implied a major breakthrough in parameter identifiability for general models, though all our results were derived on a simplified dRNN training setting. Therefore, more modest framing at the outset would have better aligned expectations with the actual contributions, which we agreed.

## Broad concerns

1) The main criticism seems to be about presentation, specifically the mismatch between expectations set by the first few pages and the results themselves.

Response: We agree and have revised the title/abstract (see below) and will revise the text to better match our contributions, removing claims of a “complete theory.”

2) There are valid concerns about the simplicity of the theorem 1, which does have connections to existing results on dynamical systems theory and linear control. Referees had some very good suggestions to strengthen these connections.

Response: We will follow these suggestions. While these connections seem natural in hindsight, introducing the Gram matrix framework to dRNN analysis is novel and provides practical tools for the field.

3) There were concerns about this simplified setting, i.e., the type of RNN used and the assumptions about 1-1 matching between artificial and biological neurons.

Response: We will highlight these limitations for a general theory. However, these exact settings are widely used in neuroscience (e.g., see Perich et al. 2021). Our framework helps researchers interpret results from methods they already use.

4) Several technical questions were raised, including the regularization of FORCE, the noise regimes tested empirically, and the local nature of some results.

Response: We agree with and will clarify all these aspects. In short,

- FORCE uses an online algorithm to approximately minimize a regularized least-squares problem.
- We will add more experiments with noisy dynamics. We conducted a brief experiment similar to Fig. 2 with $10^{-1}$ variance in input noise; our findings remain unchanged.
- We clarify that Corollary 1’s connection to weighted least-squares is illustrative, not the main result. We will move it to the methods.

5) The treatment of realistic scenarios (partial observations, significant noise, model mismatch) was insufficient in the main text, despite these being crucial for practical applications. These extensions need to be brought forward and developed more thoroughly.

Response: We agree with these comments. Specifically, we will move key results on partial observations (currently Appendix B.2) and noise analysis (Section 4.3) to the main text, reducing methodological materials in Section 4 that can be shortened or moved to methods. As referees noted, the writing is quite verbose, which we will significantly shorten as well.

6) Practical relevance of parameter identifiability was not well motivated.

Response: Since our goal is to explain, not just predict, neural responses, incorrect estimation of unconstrained parameters can introduce spurious structure that does not exist. For instance, consider inter-area communications within multi-region dRNNs, which has three “brain” regions: A, B, C. If A -> C sends a negative current and B-> C sends a positive one, the net effect is cancelled and the dynamics predicted in C is equivalent to having received no current from either brain regions at all, but the interpretations are distinct and can be affected by unconstrained parameters.

---

> ### Author Response · Authors · 2025-11-13
> **Part 2 of 2**
>
> ## Our high-level response
>
> Since referees provided constructive guidance, we believe our revisions will address their concerns. Additionally, we believe several key aspects of our work were evaluated through a different lens than intended:
>
> 1) Our work comes as the first attempt at providing general insights into parameter identifiability/recoverability in dRNNs. While other fields may have studied similar problems, which we had agreed should be highlighted, this introduction on its own has merits.
> 2) While Theorem 1's connection to linear models may seem expected in hindsight, that is only true once presented clearly as in our work. No dRNN work has ever mentioned the projection (or the Gram) matrix.
> 3) Theorem 3, which establishes how non-identifiable parameters affect dynamics and predictions, received minimal discussion despite being crucial for understanding when RNN-based conclusions about neural circuits are reliable.
> 4) We provide an end-to-end treatment, identifying the problem (Theorem 1), proposing practical solutions (Theorem 2), and analyzing implications on dynamics (Theorem 3). We believe these core contributions stand on their own merit.
>
> **Revised title:** Parameter identifiability and preserved dynamics in data-constrained recurrent neural networks
>
> **Revised abstract:**
>
> Researchers routinely study the neural algorithms of the brain by training recurrent neural networks (RNNs) to reproduce observed neural activity. However, whether the biological insights gained from these overparameterized RNNs are actionable remains underexplored. In particular, it is unclear which RNN parameters are constrained by a given training set of neural trajectories. To bridge this gap, we focus on a simplified but experimentally relevant setting of dRNN training, characterize the identifiable parameter subspaces in data-constrained RNNs, and report five key findings: (i) RNNs contain vast unconstrained parameter regions due to low-dimensional training data; (ii) existing training methods can mistakenly attribute importance to non-identifiable parameters; (iii) we provide a blueprint for designing estimators that operate exclusively within identifiable parameter subspaces; (iv) despite parameter non-identifiability, activity subspaces with preserved dynamics exist across all trained RNNs; and (v) we propose targeted intervention experiments to optimally expand the identifiable parameter subspaces. Our results establish practical guidelines to overcome parameter non-identifiability issues when training data-constrained RNN models in systems neuroscience.

---

> > ### Comment · Reviewer_Txye · 2025-11-13
> > **Please of course submit a rebuttal**
> >
> > If the authors can provide a revision and rebuttal along these lines that convincingly addresses my concerns and those of the other referees, I would of course be open to revising my recommendation of rejection.

---

> > > ### Comment · Reviewer_U37S · 2025-11-17
> > > **Ditto**
> > >
> > > Ditto

---

> > > > ### Author Response · Authors · 2025-11-20
> > > >
> > > > Thank you for the replies. We will submit a rebuttal and a revision in the upcoming days.

---

### Author Response · Authors · 2025-11-22
**General response**

We would like to thank all referees for their time and thoughtful engagement with our manuscript. The recommendations were genuinely helpful. We particularly appreciated those concerning how to situate our results within the broader theory literature. In our reading of the reviews, many concerns arose from a valid misinterpretation of our main contributions and from an emphasis from the referee side on extensions that, while valuable, were not intended to be the central findings of the work. Our central contribution is the end-to-end framework for studying, evaluating, and operationalizing parameter identifiability in dRNNs. We agree many of our results may seem natural once this framework is introduced, which in our opinion speaks to the power of the framework, but they are certainly not known or appreciated in our field.

In this revised manuscript, we tightened the scope of the manuscript and set correct expectations by revising the title, abstract, and introduction to more accurately reflect the goals and limits of our framework. We also responded to technical comments with new experiments, or clarifications in the text. Below, we summarize the revisions and restate what we view as the core contributions of our work.

## Summary of revisions

1. We have reframed Theorem 1 to include noise, and made clear connections to GLMs and linear dynamical systems.

2. We added a new Figure 2, which motivates and illustrates the use of Gram matrix for the practical estimation scenarios.

3. We simplified Theorem 2 to its most crucial components and removed a corollary that was unnecessarily confusing for the readers.

4. We added a new appendix deriving the FORCE algorithm and restating the CORNN algorithm. This appendix is now directly connecting Theorem 2 to these models, showing how our findings generalize earlier work on training dRNNs.

5. We moved analysis on low-rank RNNs and partial observed populations to Fig. 3.

6. We added two panels to Figure 4, which illustrate the utility of Theorem 3.

7. We now moved additional results to either the main text or removed unnecessary clutter completely from the manuscript.

## Technical Contributions

1. **Characterizing identifiable parameters for data-constrained RNNs (Theorem 1; Fig. 1A-B).** We derive exact conditions under which dRNN parameters are identifiable from neural trajectories. While this setting is restricted, it is regularly used in practice.

2. **Gram-matrix for quantifying estimability (Fig. 1C-E).** We show that the eigenspectrum of the Gram matrix determines the estimable parameter directions. While similar ideas exist in control theory/dynamical systems, primarily for linear systems, this connection is novel in our line of research concerning nonlinear dRNNs.

3. **Effects of noisy dynamics (Fig. 2).** We analyze how noise in dRNN dynamics alters the Gram spectrum and shrinks estimable parameter subspaces. Experiments demonstrate that noisy dynamics primarily affect low-eigenvalue directions.

4. **A general estimator design for constraining estimation to identifiable subspaces, which unifies existing training algorithms under an umbrella blueprint (Theorem 2, Fig. 3).** We introduce a unified estimator blueprint that encompasses FORCE and CORNN, but also enables designing new estimators.

5. **Identification of subspaces with preserved-dynamics across dRNNs (Theorem 3; Fig. 4).** We prove that all RNNs sharing the same identifiable parameters exhibit identical dynamics and output trajectories on specific subspaces. This explains why dRNNs with different non-identifiable parameters can still produce consistent neural dynamics within reliably supported subspaces, but also warns about the use of dRNNs outside of their validity regime.

6. **A principled strategy to design experiments that expand identifiability (Fig. 5).**  We describe how perturbations, additional inputs, or selective recordings can systematically enlarge the identifiable subspace and increase interpretability of dRNN models.

## Conceptual Contributions

1. **A clear demonstration that RNNs possess large non-identifiable parameter regions.**  We show that even in a simplified, commonly studied, controlled setting, intrinsic dimensionality of the training data quantifies the identifiability of parameters.

2. **Explanation of why and when existing RNN training can introduce spurious structure.**  Because training unconstrained directions is ill-posed, standard estimators (including FORCE) can attribute functional meaning to non-identifiable components.

3. **A framework for when RNN-based neuroscience conclusions are worth testing experimentally.** By distinguishing identifiable vs. non-identifiable components, we specify which aspects of an RNN’s learned dynamics are grounded in data and which are artifacts of underconstrained training. In a world where single-cell intervention experiments are not cheap, this provides a theoretical method for ruling out hypotheses.

---

### Note · Authors · 2025-12-05

**Comment:**

We thank all reviewers for their constructive comments. After the policy changes, three out of four reviewers did not have a chance to respond to our rebuttal. In the end, we believe our work constitutes a major advance in our field, but may require a longer review process and direct expertise working with experimental datasets to be fully appreciated. We will be submitting to a future venue.

**Withdrawal Confirmation:**

I have read and agree with the venue's withdrawal policy on behalf of myself and my co-authors.